

# Human impacts and their interactions in the Baltic Sea region

Marcus Reckermann[1], Anders Omstedt[2], Tarmo Soomere[3,4], Juris Aigars[5], Naveed Akhtar[6], Magdalena Bełdowska[7], Jacek Bełdowski[8], Tom Cronin[9], Michał Czub[10], Margit Eero[11], Kari Hyytiäinen[12], Jukka-Pekka Jalkanen[13], Anders Kiessling[14], Erik Kjellström[15], Karol Kuliński[8], Xiaoli Guo Larsén[9], Michelle McCrackin[16], H. E. Markus Meier[17,18], Sonja Oberbeckmann[19], Kevin Parnell[3], Cristian Pons-Seres de Brauwer[9], Anneli Poska[20, 21], Jarkko Saarinen[22,23], Beata Szymczycha[8], Emma Undeman[16], Anders Wörman[24], Eduardo Zorita[6]

[1]International Baltic Earth Secretariat, Helmholtz-Zentrum Hereon, Max-Planck-Str. 1, 21502 Geesthacht, Germany.
[2]Departement of Marine Sciences, Box 461, 405 30 Göteborg, Sweden.
[3]Department of Cybernetics, School of Science, Tallinn University of Technology, Ehitajate tee 5, 19086 Tallinn, Estonia
[4]Estonian Academy of Sciences, Kohtu 6, 10130 Tallinn, Estonia.
[5]Latvian Institute of Aquatic Ecology, Voleru iela 4, 1007, Rīga, Latvia.
[6]Institute of Coastal Systems – Analysis and Modeling, Helmholtz-Zentrum Hereon, Max-Planck-Str. 1, 21502 Geesthacht, Germany.
[7]Institute of Oceanography, University of Gdansk, Pilsudskiego 46, 81-378 Gdynia, Poland.
[8]Institute of Oceanology, Polish Academy of Sciences, Marine Chemistry and Biochemistry Department, Powstańców Warszawy 55, 81-712 Sopot, Poland.
[9]DTU Wind Energy Department, Frederiksborgvej 399, Bygning 115, rum S40, 4000 Roskilde, Denmark.
[10]Institute of Oceanology, Polish Academy of Sciences, Marine Ecology Department, Powstańców Warszawy 55, 81-712 Sopot, Poland.
[11]Technical University of Denmark, National Institute of Aquatic Resources, Kemitorvet, Building 201, 2800 Kgs. Lyngby, Denmark.
[12]Department of Economics and Management, P.O. Box 27, 00014 University of Helsinki, Finland.
[13]Finnish Meteorological Institute, Erik Palménin aukio 1, 00560 Helsinki, Finland.
[14]Swedish University of Agricultural Sciences, Department of Animal Nutrition and Management; Aquaculture, Box 7024, 750 07 Uppsala, Sweden.
[15]Swedish Meteorological and Hydrological Institute, Rossby Centre, 601 76 Norrköping, Sweden.
[16]Baltic Sea Centre, Baltic Nest Institute, Stockholm University, 106 91 Stockholm, Sweden.
[17]Department of Physical Oceanography and Instrumentation, Leibniz Institute for Baltic Sea Research Warnemünde, Rostock, 18119, Germany.
[18]Research and Development Department, Swedish Meteorological and Hydrological Institute, Norrköping, 601 76, Sweden.
[19]Biological Oceanography, Environmental Microbiology, Leibniz-Institute for Baltic Sea Research, Seestrasse 15, 18119 Rostock, Germany.
[20]Department of Physical Geography and Ecosystem Science, Lund University, Sölvegatan 12, 223 62 Lund, Sweden.
[21]Department of Geology, Tallinn University of Technology, Ehitajate tee 5, 19086 Tallinn Estonia.
[22]Geography Research Unit, P.O.Box 8000, 90014 University of Oulu, Oulu, Finland.
[23]School of Tourism and Hospitality, University of Johannesburg, Johannesburg, South Africa.
[24]KTH Royal Institute of Technology, 100 44 Stockholm, Sweden.

*Correspondence to*: Marcus Reckermann (marcus.reckermann@hereon.de)





**Abstract.** Coastal environments, in particular heavily populated semi-enclosed marginal seas and coasts like the Baltic Sea region, are stongly affected by human activities. A multitude of human impacts, including climate change, affects the different compartments of the environment, and these effects interact with each other.

As part of the Baltic Earth Assessment Reports (BEAR), we present an inventory and discussion of different human-induced
factors and processes affecting the environment of the Baltic Sea region, and their interrelations. Some are naturally occurring and modified by human activities (i.e. climate change, coastal processes, hypoxia, acidification, submarine groundwater discharges, marine ecosystems, non-indigenous species, land use and land cover), some are completely human-induced (i.e. agriculture, aquaculture, fisheries, river regulations, offshore wind farms, shipping, chemical contamination, dumped warfare agents, marine litter and microplastics, tourism, coastal management), and they are all interrelated to different degrees.
We present a general description and analysis of the state of knowledge on these interrelations. Our main insight is that climate change has an overarching, integrating impact on all of the other factors and can be interpreted as a background effect, which has different implications for the other factors. Impacts on the environment and the human sphere can be roughly allocated to anthropogenic drivers such as food production, energy production, transport, industry and economy.

We conclude that a sound management and regulation of human activities must be implemented in order to use and keep the
environments and ecosystems of the Baltic Sea region sustainably in a good shape. This must balance the human needs, which exert tremendous pressures on the systems, as humans are the overwhelming driving force for almost all changes we see.

The findings from this inventory of available information and analysis of the different factors and their interactions in the Baltic Sea region can largely be transferred to other comparable marginal and coastal seas in the world.



## 1 Introduction

Anthropogenic climate change has been regarded as a major driver for environmental changes since the industrial revolution. The IPCC has been the leading worldwide expert body to assess and document the currently available knowledge (Agrawala, 1998). Regional climate change assessment reports have taken the IPCC example to the regional scale, e.g. for the Baltic and North Seas (BACC Author Team, 2008, BACC II Author Team, 2015, Quante and Coljn, 2016). However, a multitude of human-induced factors next to climate change affects the environments of coastal seas. For the Baltic Sea region, the following was concluded: „*When addressing climate impacts on, for example, foresty, agriculture, urban complexes, and the marine environment in the Baltic Sea basin, a broad perspective is needed which consider not only climate change but also other significant factors such as changes in emissions, demographic and economic changes, and change in land use.*" (von Storch et al., 2015). Furthermore, it was stated that climate change effects *"are not straightforward and are difficult to distinguish from other human factors such as atmospheric deposition, forest and wetland management, eutrophication and hydrological alterations"* (Humborg et al., 2015). In this paper, we examine a number of different human factors (see definition of terms below) affecting the coastal environment of the Baltic Sea region. We assess what is currently known about the impact of climate change on these factors and how they influence each other. In this respect, we define the coastal zone as the land areas, which have an impact on or are affected by the coastal sea, and the sea areas that are directly and substantially affected by the land in their immediate neighborhood, or have an impact on the land. The Baltic Sea is a marginal sea that is strongly influenced by its coasts so that coastal issues can be examined here like under a magnifying glass.

The notion that the state and function of physical and living environments depend on different interacting factors is old. Humboldt stated in 1807: *"In this great chain of causes and effects, no single fact can be considered in isolation. The general equilibrium obtaining in the midst of these disturbances and apparent disorder is the result of an infinite number of mechanical forces and chemical attractions which balance each other; and while each series of facts must be examined separately in order to recognize a specific law, the study of nature, which is the main problem of general physics, demands the gathering together of all the knowledge dealing with modifications of matter."* (Humboldt and Bonpland, 2009). This notion has since evolved into now generally accepted and used concepts, e.g. "ecosystem" (Tansley, 1935), "biosphere" and "biogeochemistry" (Lapo, 2001).

At least since the Neolithic Revolution, i.e. the transformation from a hunter-gatherer to a farmer subsistence, humans have shaped the earth´s surface, including the Baltic Sea region (Lavento, 2019). With industrialization and its related technological developments, e.g. the Haber Bosch process to make nitrogen useable for the production of fertilizers (Smil, 1999, Erisman et al., 2008) and the general application of standard hygienic procedures (Cavaillon and Chrétien, 2019), the human population has increased in such a way that it has become a dominant factor in shaping the environment (Lewis and Maslin, 2015). Consequently, a new geological epoch has been suggested and labeled as the "age of humans", the "anthropocene" (Crutzen, 2002).

Climate and other human-induced factors are regionally very different. The warming with its direct and indirect consequences (e.g. reduced sea ice, sea level rise, hydrological changes) affects all natural processes and may impact other factors. Here, we make an inventory of different human factors including climate change, and assess how they may interact, as far as we found evidence in the scientific literature. Some factors may be completely independent of the warming, which may directly or indirectly affect others. The list of factors treated in this paper cannot be complete and should be updated continuously.

Feedbacks within the complex regional Earth system (e.g. the atmosphere, land surfaces, water bodies, biosphere, biogeochemistry, geology) may be complicated and difficult to disentangle, more so when human impacts are involved (Gaillard, 2013, Gaillard et al., 2015). The different factors may affect each other, synchronously or cumulatively, creating negative or positive feedback effects. While a direct effect may be straightforward and easy to detect and explain, the indirect effects are mostly more difficult to uncover. Extreme precipitation events have meteorological causes, which may be connected



to changing climate. Still, the impacts of such events on the human environment, like flooding, damage or drying crops may be caused or exacerbated by human design (impervious surfaces or other land use changes such as mono-cultural agriculture). In some cases, the local climate itself can be impacted by human-induced changes in the environment (e.g. albedo changes

due to afforestation, desertification or contruction; Gaillard et al., 2015). Therefore, we face a complex and non-linear system of effects and feedbacks between climatic and non-climatic factors. Moreover, politically motivated management decisions, which have no or little natural scientific groundings, may have more substantial impacts than natural ones and may be even more unpredictable than them. Some projects have attempted to include human behavior into scenarios (e.g. BONUS BALTICAPP and others (Hasler et al., 2019), see also Arheimer et al., 2012; Zandersen et al., 2019; Bartosova et al., 2019;

Pihlainen et al., 2020).

Complex analyses and modeling exercises have been performed to characterize the interactions between the different factors (e.g. Crain et al., 2008, Liess et al., 2016, Robinson et al., 2018, Stelzenmüller et al., 2018, Gissi et al., 2021). The present paper intends to make a novel and straightforward inventory of factors and connections, covering the above aspects as far as possible; information on individual factors must be limited and just of overview character. This paper should serve as an easily

accessible introduction to the topic with an emphasis on the Baltic Sea region, but almost all aspects can be transferred to similar marginal sea regions of the world. We elaborate on the relationships of the different factors with respect to climate change on the one hand, and among the various factors on the other hand. As humans are heavily involved, socio-economic issues are strongly relevant for most factors and connections described here.

Since the early 1990s, the knowledge on the physical and biogeochemical environments of the Baltic Sea and their relationship

has been systematically assessed, initially by BALTEX and since 2013 by its successor Baltic Earth. This study is one of the thematic Baltic Earth Assessment Reports (BEARs), which comprises a series of review papers that summarize and assess the available published scientific knowledge on climatic, environmental and human-induced changes in the Baltic Sea region (including its catchment). The BEAR reports in this Special Issue of Earth System Dynamics reflect the Baltic Earth Grand Challenges and scientific topics of Baltic Earth (Baltic Earth, 2017).

While the other papers in this Special Issue deal with natural factors and their relation to climate change (salinity, biogeochemistry, natural hazards) and scenarios for future conditions in the Baltic Sea region, this paper addresses natural and anthropogenic factors in addition to climate change. We assess how they affect, or are affected by, climate change and how they interact. We believe that the findings elaborated in this assessment can largely be transferred to other marginal and coastal seas, which are comparably heavily used and affected by humans.

## 205 2 The region

The Baltic Sea region has been subject to dramatic environmental changes since the last glaciation (Borzenkova et al., 2015). Human activities have strongly affected the region since the withdrawal of the ice sheets. Fishers, gatherers and hunters inhabited the coasts of the early Baltic Sea already at 11,000 years BP, and neolithic cultures practiced crop cultivation and animal husbandry around 6,000 BP. Deforestation and changes in forest composition have been documented since around

4,000 BP (Gaillard et al., 2015). Over the centuries, the human impact on the environment extended to more detrimental effects such as pollution due to iron mining (Lavento, 2019). Currently, the Baltic Sea drainage basin covers about 20% of the European continent, with roughly 85 million people living in the catchment (HELCOM, 2018a). It can be roughly subdivided into a sparsely populated, mostly pristine north with natural coastal (rocky) landscapes, and a strongly transformed, agricultural landscape in the highly populated south, with mostly low sedimentary coasts and graded shorelines (Figure 1). Numerous

rivers enter the Baltic Sea, some of them with catchments covering more than one country, draining nutrients, sediments and pollutants from the surrounding land areas into the Baltic Sea (HELCOM, 2018b).



The Baltic Sea features some special conditions, which make it vulnerable to specific pressures. It is almost enclosed from the open ocean, the exchange through the narrow Danish Belts and Sounds is very restricted, and the tidal range is very small (Feistel et al., 2008, Leppäranta and Myrberg, 2009). In addition to that, the Baltic Sea has a complex bottom topography, with

deep basins separated by shallow sills, which hamper water exchanges. These factors together with the dense population and the prevalent agricultural lands in the southern catchment give rise to special biogeochemical conditions. Eutrophication has for many years been identified as a major threat to the Baltic Sea, which has resulted in the implementation of the HELCOM Baltic Sea Action Plan (HELCOM, 2007). The plan is being updated in 2021, with additional factors discussed, including climate change.

Various organisations are working on environmental issues of the Baltic Sea. HELCOM, the intergovernmental organization to protect the marine environment of the Baltic Sea, has for decades worked on describing the different drivers and stressors (e.g. HELCOM, 2018a) and has functioned as an interface between the science and political decision making communities. HELCOM´s importance on the firm and largely successful management of various environmental issues in the Baltic Sea cannot be underestimated. In close collaboration with Baltic Earth, the HELCOM expert network on climate change EN Clime

has produced a climate change fact sheet, including impacts on many environmental factors (HELCOM, 2021). BONUS (Kononen et al., 2014), the EU funding scheme for environmental Baltic Sea research (succeeded by the North-Baltic Sea common scheme BANOS), has funded many projects dealing with different anthropogenic factors and their interrelations (e.g. AMBER, BALTICAPP, BALTIC-C, ECOSUPPORT, INFLOW, INTEGRAL, MIRACLE, SHEBA and others, see BONUS (2021), and has also had a significant impact on management.


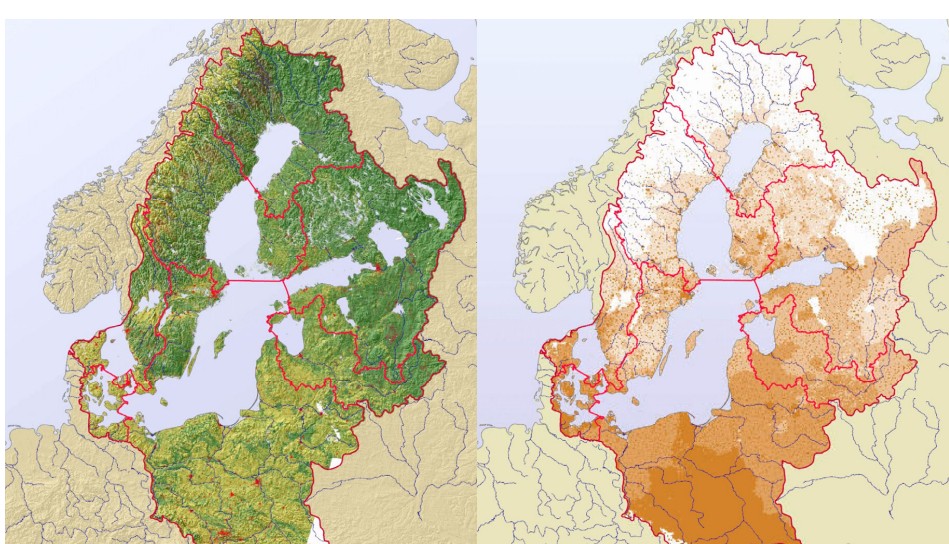

**Figure 1. The Baltic Sea drainage basin with land cover (left) and population distribution (right). Dark green: forests, light green: open and agricultural spaces, different shades of orange stand for population density. Red lines designate drainage subbasins. Maps from GRID-Arendal Baltic Environmental Atlas, http://maps.grida.no**


Natural variability in the region is very strong and the anthropogenic climate signal attributable to greenhouse gases is just beginning to emerge from the background noise (Bhend, 2015;, Barkhordarian et al., 2016). Parameters directly related to temperature, like ice extent or the seasonality of biota show the most robust signals in climate-related changes (BACC Author Team, 2008; BACC II Author Team, 2015). For many other climatic parameters, including precipitation, wind, runoff and





salinity, the signal is still small compared to the variability, precluding any conclusions about emerging trends (Christensen et al., 2021; Meier et al., 2021a).

## 3 Terminology and the DPSIR description

The terms *drivers*, *pressures*, *stressors*, *impacts, cumulative impacts* etc. are all commonly used in the literature, and they are generally not strictly defined. The term *cumulative* implies that effects add up to a final, stronger impact than each of the
individual drivers, either immediately or (more frequently) over a longer time. This may not be the case for all combination of drivers, as they may act additively, synergistically or antagonistically (Boldt et al., 2014). The terms *drivers*, *pressures*, *states*, *impacts* and *responses* are part of the DPSIR concept to assign a structure to the different factors and their links to environmental changes (EEA, 1999; Gari et al., 2015).

In Table 1, we use this concept to define the different factors we analyze in depth in Section 5. For each factor (except *Marine*
*Ecosystems* and *Coastal Management*), we apply the DPSIR concept, which facilitates the understanding of the underlying driving forces, pressures and impacts. In the following analysis in Section 5 and thereafter, we focus on the impacts.

**Table 1: Matrix of different factors using the DPSIR analysis (Drivers, Pressures, States, Impacts, Responses).**

| Factor | Driver | Pressure | State | Impact | Response |
|---|---|---|---|---|---|
| Climate change | Industrialization | GHG emissions | Current climate | Warming, modification of hydrological cycle, acidification, sea level rise | Climate change mitigation; reduction of GHG emissions; geoengineering |
| Coastal processes | Wind and waves, currents, human activities | Erosion, accretion, coastal infrastructure | Current coastal condition | Impacts on coastal ecosystems and human uses | Coastal management and infrastructure |
| Hypoxia | Food production | Nutrient releases | Current state of hypoxia | "Dead" zones, modification of redox state and biogeochemistry, impact on ecosystems | Reduction of nutrients, geoengineering |
| Acidification | Industrialization | GHG emissions, alkalinity | Current pH of sea water | Impacts on physiological functions of certain species, hampered calcification of shells | Reduction of GHG emissions; geoengineering |
| SGD | Hydrology, coastal aquifers | Modified concentrations of nutrients and chemical substances in coastal waters | Current groundwater release in some coastal areas | Impacts on nutrient and contaminant concentrations and related impacts on organisms and coastal ecosystems | Coastal management and infrastructure |
| Non-indigenous species | Globalization and transport, climate change | Ballast water/hull transport and other new parthways, e.g. though channels; new species invading indigenous ecosystems (migration northwards) | Current abundances of non-indigenous species in different regions | Modification of indigenous ecosystems, expulsion of indigenic species and ecosystem functions, species diversity potentially affected | Antifouling and ballast water regulations, climate change mitigation |
| Land cover and use | Food production, transport, industrialization | Monocultures, expulsion of indigenous ecosystems, modification of runoff substances, sealing of soils, fragmentation of ecosystems and landscapes | Current state of land use and cover | Eutrophication, inundations, adverse effects on species diversity | Land restorations |
| Agriculture/Nutrient loads | Food production | Excess nutrients entering the ecosystems | Current state of agricultural release of nutrients to the sea | Eutrophication, hypoxia | Reduction of nutrient release |
| Aquaculture | Food production | Release of excess nutrients and chemicals/pharmaceuticals to coastal waters, indirect excess fisheries for fish meal in other parts of the world | Current state of aquaculture | Eutrophication, pollution | Land based circulation systems, sustainable use of open systems |
| Fisheries | Food production | Removal of fish biomass and related ecosystem functions | Current state of fisheries and fish stocks | Decimation of commercial fish species; impacts on ecosystem functions and fisheries; cascading effects on other parts of the | Fishery regulations |
| River regulations | Transport, energy and water management | Modification of river flows, construction of channels and dams | Current state of river regulation and restauration | Modification of flow rates and nutrients, resp. nutrient ratios downstream; impacts on migrating fish and related fisheries | River restaurations |
| Offshore wind farms | Energy production, climate change | Extensive wind farms in coastal and open waters | Current development of offshore wind farms | Impacts on local ecosystems; fish and marine mammals; bird migration routes; noise during construction phase | Regulation of offshore industry |
| Shipping | Transport | Commercial shipping along international shipping routes, cruise and leisure shipping in coastal waters | Current state of shipping | Impacts on marine ecosystems through antifouling, ballast water, black/grey and waste water, scrubber water, underwater noise, various contaminants; impacts on air quality through exhaust and combustion products; GHG emissions | IMO regulations |
| Chemical contaminants | Transport, industrialization, chemical production, food production, various economic activities | Diffuse and point source emissions of organic contaminants and heavy metals to air, water and land, subsequent transport to the sea | Current concentrations of contaminants in waters, sediments and organisms | Impacts on physiological functions of various marine organisms; impacts on human health due to food chain bioaccumulation of some contaminants. | Different regulations/technical guidance on chemical production, use, waste handling; global treaties |
| Dumped millitary material | World War II | Dumping of unexploded warfare agents in various locations | Current state of corrosion of dumped warfare agents | Potentially harmful impacts on marine ecosystems, potential danger of poisoning and accumulation up the food chain up to humans | Various national and international efforts to retrieve the dumped objects as far as possible |
| Marine litter | Industrialization, chemical production, various economic activities | Diffuse release via rivers and other pathways to the sea; concentrations on specific locations (eddies, coastal streches, beaches) | Current concentrations and distribution patterns of marine litter and microplastics | Potentially harmful effects on the physiology of different organisms | Regulations in plastic production, distribution and use; efforts to retrieve larger fragments from sea water |
| Tourism | Human recreation | Coastal regions flooded with humans; cruise shipping, leisure shipping | Current state of tourism | Impacts on transport, waste management, coastal infrastructure in coastal regions; impacts on coastal ecosystems though fishing, angling, boating, bathing etc. | Efforts to implement a sustainable tourism |


We classify our parameters into two groups, loosely following Boldt et al., 2014, Table 4: Firstly, we consider natural environmental factors which would prevail on an Earth without humans, but which are strongly affected by human activities: Climate, land cover, sea level, coastal processes, nutrient loads, hypoxia, acidification, submarine groundwater discharge and non-indigenous species. Then we consider the human factors offshore wind farms, shipping, fisheries, chemical contaminants,
dumped ammunitions, marine litter and microplastics, agriculture, aquaculture, river regulations, tourism and coastal management. A clear separation is sometimes difficult (e.g. for land cover and land use) but this is not relevant in this context.



We define the term *environment* here in an integrative manner as the non-living (abiotic), physical environment, like wind, temperature, precipitation etc.; the living or directly affected environment, i.e. ecosystems and biogeochemical conditions; and the socio-economic environment, which is everything related to human activities including infrastructure at the coast or sea, or agriculture. Here, we define the term *climate change* to describe the human-induced changes to the climate. Indeed, the term *global change* (or *regional change*) would better depict the amalgam of current changes because climate change is but one human-induced factor, and it may not be the dominant one in many cases.

It is worth noting that there are other, deeper layers of drivers of environmental change. Human needs and the underlying values determine the levels of production, consumption and efforts consumers, industries and societies put into managing environmental problems. Human needs and aspirations are realized in consumption patterns, life styles, education and savings rate. At an aggregate level, they can be described in terms of social cohesion, speed of urbanization, and birth rate. These aspects of behaviour are ultimately driven by values, cultures, religion and habits. Socio-economic impacts can be included in the climate models by prescribing specific emissions storylines based on socio-economic developments and allow studies of alternative future states. The large spread in the different scenarios reflects the uncertainties in future development (Hyytiänen et al. 2021).

## 4 The impact matrix of interrelations

To allow a comprehensive and straightforward approach to the problem, we introduce a matrix in which we show at a glance the impacts of the various factors on each other. This matrix is the core of our analysis (Table 2). We try to give a qualitative (is there evidence for a connection or impact, or not) overview of the connections between the different factors. The text in Section 5 provides brief characterizations of the current knowledge of the factors and bullet lists of potential interrelations for each factor, following the matrix (Table 2). As far as possible, we try to present evidence, i.e. references from the scientific literature, for individual links between factors. However, we also speculate on potential links (question marks), which have not (or sparsely) been confirmed in the scientific literature but which may be plausible, not to rule out a connection, which has not yet been described. We regard the items emphasized by question marks as particularly interesting as they represent a potential connection for which there is no or insufficient evidence in the literature, but which cannot be excluded plausibly, and which may be worth considering further. These linkages are also briefly discussed in Section 6 as potential knowledge gaps. The different factors and their connection with others, as well as to climate, are discussed in the text.

**Table 2: The matrix of factors. + = evidence for a connection; - = no evidence for a connection; ? = no evidence but connection plausible (based on author´s judgment). The table is read from left to right, i.e. going to the right within the first line "Climate change" shows factors that climate change has an impact on (or not), etc.**





| impact by↓/on→ | Climate change | Coastal processes | Hypoxia | Acidification | SGD | Marine ecosystems | Non-inig. species | Land cover and use | Agriculture Nutr. loads | Aquaculture | Fisheries | River regulations | Offshore wind farms | Shipping | Chem. Contamin. | Dumped military | Marine litter | Tourism | Coastal management |
|---|---|---|---|---|---|---|---|---|---|---|---|---|---|---|---|---|---|---|---|
| Climate change | | + | + | + | ? | + | + | + | + | + | + | ? | + | + | + | ? | ? | + | + |
| Coastal processes | - | | ? | ? | + | ? | - | + | + | ? | ? | + | + | + | ? | ? | + | - | + |
| Hypoxia | - | - | | + | - | + | - | - | + | ? | + | - | - | - | + | ? | - | - | - |
| Acidification | - | - | - | | - | ? | - | - | - | ? | ? | - | - | - | ? | - | - | - | - |
| SGD | - | - | ? | ? | | ? | - | - | + | - | - | - | - | - | + | - | - | - | - |
| Marine ecosystems | - | - | + | + | - | | + | - | - | - | + | - | - | - | - | - | - | + | - |
| Non-inigenous species | - | - | - | - | - | + | | - | - | - | + | - | - | + | + | - | - | - | ? |
| Land cover and use | + | - | + | + | + | ? | - | | + | - | + | + | ? | - | + | - | - | + | + |
| Agriculture/Nutrient loads | + | - | + | + | + | + | - | + | | + | + | + | - | - | + | - | - | - | ? |
| Aquaculture | - | + | + | - | - | + | + | + | + | | ? | - | + | - | ? | - | ? | + | + |
| Fisheries | - | - | ? | - | - | + | ? | - | ? | ? | | - | + | ? | - | - | + | ? | + |
| River regulations | - | + | ? | + | ? | ? | - | + | + | ? | + | | - | - | ? | - | ? | - | + |
| Offshore wind farms | + | + | - | - | - | + | - | ? | - | + | + | - | | + | + | ? | ? | + | + |
| Shipping | + | + | - | + | - | + | + | - | + | ? | + | - | + | | + | - | + | + | + |
| Chemical contaminants | - | - | - | - | - | + | - | - | + | + | + | - | - | - | | - | - | - | - |
| Dumped military | - | - | - | - | - | ? | - | - | - | - | + | - | + | - | + | | - | - | ? |
| Marine litter | - | - | - | - | - | + | - | - | - | ? | + | - | - | - | ? | - | | + | ? |
| Tourism | + | - | - | - | - | ? | - | + | + | - | - | - | + | + | - | - | + | | + |
| Coastal management | - | + | - | - | ? | ? | ? | + | + | + | + | ? | + | + | - | + | ? | + | |

+   evidence for a connection
-   no evidence for a connection
?   no evidence but connection plausible (based on "expert" judgment)

## 5 Factors of regional change

In this section, we provide short overviews over the current state of knowledge of some factors, which affect the regional Earth system of the Baltic Sea region, followed by a bullet list describing how it may have an impact on any other factor discussed here. For all described effects and interrelations, we refer to the regional scale of the Baltic Sea region.

### *Natural factors*

The first part of this section addresses "natural" factors that would still be part of the environment even if no humans existed, but which are today also heavily affected by human activities.

### 5.1 Climate change

In the context of this analysis, we describe here very briefly the immediate climate impacts on the environment: warming, precipitation and runoff changes, ice conditions and sea level change. For a detailed analysis on climate change and modeling for the Baltic Sea region, see Christensen et al. (2021), Gröger et al. (2021) and Meier et al. (2021a, 2021b).

Global climate change depends largely on the radiative forcings by changing greenhouse gases, aerosol particles and land use, and what their respective impacts on the Earth´s energy budget are (Cubasch et al., 2013). On regional scales, climate change depends both on regional and local climate forcing factors as well as on changes in the large-scale circulation of the atmosphere and oceans. The latter can be a result of long-term global change, but natural variability on decadal/centennial time scales is also of major importance in this context (Hawkins and Sutton, 2009). All of the above-mentioned factors are associated with uncertainties that make predictions of future climate conditions difficult. Uncertainties of future trends are smaller for thermal variables and larger for precipitation and atmospheric circulation (Kjellström et al., 2013).

The most widely used approach to deal with these difficulties is the use of large ensembles of climate model projections (Collins et al., 2013). A large number of simulations in such ensembles yield a spread that may be used to address the uncertainties: use of different climate models reflect uncertainties in the sensitivity of the climate system; use of different scenarios reflect uncertainties in the future forcing conditions; use of a large number of simulations reflects uncertainties in natural variability.



As the long-term socio-economic development is highly uncertain, scenarios for the future are designed to represent a wide range of forcing conditions. In the scenarios assessed by the AR5 (Stocker et al., 2013), four different Representative Concentration Pathways (RCPs) are considered: RCP2.6, RCP4.5, RCP6.0 and RCP8.5. The numbers indicate the radiative

forcing (in W m$^{-2}$) at the end of the 21$^{st}$ century, relative to that of preindustrial conditions. The large spread between different simulations reflects the combined uncertainty between global climate sensitivity, regional response and natural variability.

Due to its proximity to the northern polar region, the Baltic Sea region is warming faster than the globe. Wintertime changes in **air temperature** are among the strongest climate change signals in Europe, and the land surface has been warming faster than the Baltic Sea. During the past century, an approximate increase of 1°C was observed over the Baltic Sea region (BACC

Author Team, 2008; BACC II Author Team, 2015, Rutgersson et al. 2014), and projected changes until 2100 range between 1.5 - 4.3 °C over land and 1.4 - 3.9 °C over sea, according to coupled atmosphere-ocean projections (Meier et al. 2021a), and depending on the RCP, with stronger warming in the northern part of the basin, especially in winter (Gröger et al. 2021).

For **precipitation**, the uncertainties are larger. While there is a large variability between seasons and regions, there is a trend projected for the future for precipitation, with an increase for the entire region in winter, but only for the northern part in

summer. For the southern part of the basin, the projections vary and a clear trend cannot be given (Christensen and Kjellström, 2018). Also for **wind**, projections for the future vary considerably, so that no clear trend over the whole Baltic Sea region can be identified (Räisänen, 2017; Christensen and Kjellström, 2018; Christensen et al., 2021; Meier et al. 2021a).

**Sea water temperatures** have already begun to increase, both at the surface and in deep waters of the Baltic Sea (Lehmann et al., 2011; Elken et al., 2015; Mohrholz et al., 2006), and are projected to increase further (on average 1.6 -3.2° by the end

of this century, Gröger et al. 2019, Meier and Saraiva 2020, Meier et al. 2021a). The increase in sea surface temperature is projected to be largest in the northern Baltic Sea during early summer, very likely due to the ice-albedo feedback causing earlier warming during the melting season.

All available scenario simulations for **sea ice** suggest a drastic decrease in sea-ice cover in the future (Luomaranta et al., 2014; Meier, 2015). However, even the extreme scenarios do not suggest a complete disappearance of sea ice in the northernmost

part of the Baltic Sea. The large interannual variability is expected to remain, with a decreasing probability of severe ice winters (Höglund et al., 2017).

**Salinity** in the Baltic Sea depends on freshwater inputs (river runoff, precipitation), evaporation and outflows through the Kattegat, as well as saltwater inflows. There have been large decadal variations in mean salinity in the past, but no long-term trend (Lehmann et al., 2021). Salinity projections for the future show large variations as the factors wind, runoff and sea level

rise act antagonistically and their effects are difficult to project (Meier et al., 2021b; Meier et al., 2021c).

**Sea level rise** is treated here as a direct consequence of climate change. It is well documented that sea level variations are caused by the prevailing climate, through warming (continental ice sheet melting, thermal expansion, and atmospheric circulation changes). Post-glacial land uplift is a considerable factor in this region and ranges from a slight sinking in the southwest (-1 mm yr$^{-1}$) to a considerable rise of up to 9mm/yr in the north (Harff et al., 2020).

Warming of the global ocean has the most obvious impact on Baltic Sea sea level. Water masses expand, causing a sea level rise in the Baltic Sea. A second large factor is the melting of glaciers and of polar ice sheets. This contribution is still rather uncertain and models of ice sheet dynamics are still not sophisticated enough to provide an accurate picture under different warming scenarios. Counter-intuitively, future trends in Baltic Sea sea levels will be mostly influenced by the melting of land-ice in Antarctica and other remote regions, and very little by the melting of neighboring land-ice in Greenland (Mitrovica et

al., 2001). However, the response to melting ice sheets on Greenland differs between regions (Hieronymus and Kalén, 2020). Other factors affecting the local sea level in the Baltic Sea are atmospheric circulation (Gräwe et al., 2019), sea ice and salinity (Ekman and Mäkinen, 1996).

Global sea level rise is currently estimated to range between 43 cm and 84 cm, depending on the scenario (Oppenheimer et al., 2019). There are different estimations of how the global sea level rise (corrected for land uplift) will propagate in the Baltic



Sea, ranging between about 80% (Grinsted et al., 2015; Hieroniumus and Kalen, 2020) to 100 % of the global rise. For a full review on seal level change in the Baltic Sea, see Weisse et al. (2021).

**Impacts of climate change on other factors**

• There is good evidence that climate change has a strong impact on **coastal processes (+)**, though sea level rise on coastal erosion (Defeo et al., 2009) and the translocation of sediments through erosion, currents and accretion (Slott et al., 2006; Fitzgerald et al., 2008). In general, coasts are expected to be subject to stronger changes not only by inundation of low-lying coastal regions but also through increased wave-driven sediment translocations Deng et al., 2019). As Baltic Sea waves often approach sedimentary shores under relatively large angles (Soomere and Viška, 375  2014), alongshore sediment transport may be particularly intense with rising sea level in the Baltic Sea. A considerable reduction of sea ice increases erosion rates at soft shores (Orviku et al., 2003; Overeem et al., 2011; Farquharson et al., 2018). Due to the large variability in observations and projections, there is so far no clear indication for changes in storm frequencies, severity and tracks in the Baltic Sea region, so their impact on coastal processes remains speculative.

• There is good evidence that **hypoxia (+)** is strongly affected by climate change indirectly through temperature, salinity and stratification, possibly also by altered precipitation and runoff patterns in the southern Baltic Sea region (Zillén et al., 2008). It is unclear whether the frequency of Baltic inflow events which provide new oxygen-rich deep water for the deep basins of the central Baltic Sea will change. A higher sea level and associated larger cross section in the Danish straits may have an impact on the volumes of future Baltic inflows of high saline waters. This in turn may 385  have an impact on hypoxia in the large basins through stronger salt-water inflows and associated stronger stratification in the deep basins (Meier et al., 2017). With more hypoxia, more phosphorus release from the deep anoxic basins may enhance cyanobacteria blooms and further deteriorate the oxygen situation in the deep basins. Still, the frequencies and consequences of possible stronger oxygen-rich saltwater inflows remain largely unknown and speculative.

• **Marine acidification (+)** is very much a product of anthropogenic $CO_2$ emissions, in the course of fossil fuel combustion and changes in land use (e.g. Doney et al., 2009). The increase in atmospheric $CO_2$ concentrations leads to enhanced dissolution of $CO_2$ in seawater. As $CO_2$ dissolved in seawater forms weak diprotic carbonic acid, its dissociation causes seawater pH to decrease (Kuliński et al., 2017). Acidification in the Baltic Sea is different from other ocean provinces and marginal seas as it is very much affected by alkalinity, in particular in the northern Baltic 395  Sea (Müller et al., 2016). Anther source of anthropogenic acidification is atmospheric deposition of sulfur and nitrogen oxides, being also combustion products. They cause the so-called acid rain, which is especially relevant for soils and inland waters (e.g. Tranvik, 2021). Shipping may also contribute to acidification through scrubber water.

• **Submarine Groundwater Discharge (?)** has only recently been considered as a potential factor to affect coastal waters. There is no evidence that there is a direct impact by climate change on the quantity and quality of these 400  submarine discharges, but looking at the driving forces of submarine groundwater discharges (SGDs), this is plausible. Driving forces of SGDs involve topography-driven flow, wave set-up, precipitation, sea level rise and convection caused by salinity and temperature between the seawater and groundwater (Burnett et al. 2006; Taniguchi et al., 2019). Changed groundwater levels (lowering in dry seasons, or rising at times of strong precipitation) may be the effect of changing precipitation patterns and higher temperatures leading to stronger evaporation. As climate 405  change is expected to affect most of the above-mentioned factors, consequently it can also be expected to influence SGDs. The magnitude and relevance of these changes remain unclear, therefore impacts on SGDs by sea level rise and changes in precipitation and/or evapotranspiration need to be evaluated (Taniguchi et al., 2019). There may be



an effect of rising sea level (and geostatic land rise). Even though it is hard to project the direction and significance of the change due to missing data and investigations (Taniguchi et al., 2019; Kłostowska et al., 2020), modeling efforts signal that the total fluxes of submarine groundwater discharge may eventually decrease significantly with future sea level rise. This process is likely associated with a marked decline in the flux of nutrients and carbon to estuaries and the coastal ocean (Evans et al., 2020). Numerical studies imply that sea level changes may be responsible for an increase in recirculated SGD (RSGD), but this is not clear (Lee et al., 2013).

- Climate change has a multitude of impacts on the **marine ecosystems (+)** of the Baltic Sea, mostly through water temperature and oxygen concentrations, and indirectly through nutrient availability, facilitation of non-indigenous species migration, and complex food web interactions (Viitasalo et al., 2015). These effects are mostly non-linear, and may interact with one another, and they may be beneficial or detrimental for different ecosystem constituents or functions. A comprehensive description of climate change effects on the ecosystems of the Baltic Sea is provided by Viitasalo (2021) and references therein.

- Climate change has generally shifted the species boundaries northwards, so it is a plausible driver for the migration and occurrence of **non-indigenous species (+)**, although there is little direct evidence. Shipping through ballast water or attachment to hulls or the disappearance of physical barriers (e.g. though the construction of canals between separated water bodies) has been identified as a major vector for the introduction of new marine species into the Baltic Sea ecosystem (Ojaveer et al., 2017). Therefore, the physical transfer is not climate change related, but a changing climate can provide favourable growth conditions in the target region, e.g. though changes in temperature and salinity or prey composition (Möllmann et al., 2005). Another way to introduce new species is the direct migration to regions where climate change has established favourable conditions. This is, however, negligible compared to marine traffic and represents a rather gradual introduction. Northward migration of land (Smith et al., 2008) and marine species (for which temperature is critical) has been observed and are expected for the future (McKenzie and Schiedeck, 2007; Holopainen et al., 2016).

- Climate change affects air and water temperature as well as precipitation patterns, so there is a clear impact on **land use** and **land cover (+)**. Sea level rise may also affect land cover and use by increased flooding or eroding coastal areas, which are thus lost to the sea or converted to wetlands (e.g. Gadan et al., 2020). Growth conditions on land and the vegetation zones are affected by changing temperatures and precipitation patterns (Smith et al., 2008), but also by political or management decisions, which in turn may or may not be influenced by climate change (Yli-Pelkonen, 2008). Higher temperatures and $CO_2$ concentrations in the atmosphere lead to thriving vegetation, but declining water availability, presumably in the southern parts of the region, sets limits to this (Smith et al., 2008). The decisions, which part of land is dedicated to which use (e.g. agriculture) is very much a management and political decision, which in turn may be affected by climatic conditions.

- The most important type of land use, at least in the southern part of the Baltic Sea basin, is **agriculture (+)**. Climate change has a strong impact on the different kinds of crops as different crops have different requirements concerning water availability and soil type. So, any changes in temperature and precipitation may lead to the need for better adapted crops as a response (Fronzek and Carter, 2007). Still, socio-economic considerations may be still more important in defining the type of agricultural land cover than climatic ones (e.g. Rounsevell et al., 2005; Bartosova et al., 2019; Pihlainen et al., 2020). Climate change changes precipitation and runoff patterns from agricultural fields and thus largely determine the amounts of **nutrients (+)** entering the sea. Still, the effects of climate change on nutrient loads are rather uncertain. Climate projections indicate that the northern part of the Baltic Sea basin could be wetter but for the south, changes are unknown (Christensen et al., 2021). That would imply that riverine and runoff fluxes of nutrients could decline as most nutrients enter the Baltic Sea in the southern part of the basin, but it is difficult to assess which sources dominate in different sea sub-basins, and catchment-wide models of nutrient source



apportionment are scarce (Bartosova et al., 2019, HELCOM, 2018b, c). Furthermore, is not known how fertilization practices, crops grown, and land use will change in response to climate change. Also unknown is the relative contribution of nitrogen from accumulated, legacy sources to current riverine loads to the sea, and how the accumulation and release of "legacy" nutrients will be impacted by climate change. Warmer winters without snow cover and non-freezing soil have resulted in proportionally more soil erosion, larger runoff and consequently more nutrients transported to the sea (Huttunen et al., 2015). Climate-related changes in the Baltic Sea like warmer temperatures, changed stratification patterns, altered ecosystems and biogeochemical pathways may change the fate of nutrients in the sea (Arheimer et al. 2012; Meier et al., 2012). Sea level rise may have an impact on the internal loading of phosphate, through potentially increased saltwater inflows in the future with rising sea levels, affecting ecosystems in micro-tidal lagoons (Huang et al., 2020) and increasing the hypoxic areas in the deep water and the associated phosphate release (Meier et al., 2017).

- **Aquaculture (+)** is directly affected by rising water temperatures. Surface water (0-10 m) temperatures often exceed the optimal range for rainbow trout, a major aquaculture fish in the Baltic Sea region (Kankainen et al., 2020). Farmers report longer periods of excessively warm waters, causing reduced physical fitness, impaired growth and even fish kills which may be related to warming and other associated impacts (Brander et al., 2017; Reid et al., 2019). Therefore, a trend towards more warm water species as pikeperch, perch etc. is possible. Salinity changes are unlikely to affect currently farmed fish species in the Baltic Sea. Warmer growth seasons might increase the risk of successful establishment of farmed rainbow trout in the wild, demanding the use of sterile fish as mitigation. Blue mussels and macroalgae aquaculture may be negatively affected by both warmer temperatures and, potentially, declining salinity. Any increase in waves and extreme events as well as predation by fish and birds would lead to higher mussel losses. All these climatic effects are to a large extent eliminated in enclosed, controlled, land-based farming facilities.

- **Fisheries (+)** are strongly affected by climate change impacts on the commercially interesting fish populations in the Baltic Sea, that is mostly cod, sprat and herring (Möllmann, 2019). Climate affects salinity and temperature, which in turn affect the reproduction and growth of several fish species (MacKenzie and Köster, 2004; MacKenzie and Schiedeck, 2007; Köster et al., 2017), and thereby the availability of the resources that fisheries can exploit. Growth of planktivorous species or life stages is affected by climatic conditions that are regulating zooplankton dynamics (Casini et al., 2011; Köster et al., 2017). Climate effects are also connected to oxygen conditions in the Baltic Sea, affecting the organisms and fish production in several ways, e.g. for cod (Köster et al., 2005). Climate impacts on one species can also propagate through the food web via food web interactions. For example, a high abundance of sprat due to favourable temperatures increases competition between sprat and herring and reduces their growth and condition (Casini et al., 2011). Changes in ice conditions in the future may affect the duration of fishing season in northern areas in the Baltic Sea, with potential consequences for some fish stocks (Bauer et al., 2019).

- Inland shipping and water management have resulted in **river regulations (?)** for centuries, and the hydrology of many catchment basins, including the Baltic Sea, is heavily modified (e.g. Wanders and Wada, 2015). Climate change may affect river regulations. Increasing droughts with lower river water volume at certain times of the season may affect water management and shipping in the southern catchment basin. On the other hand, extreme rain events may lead to inundation, where the river was regulated and natural inundation areas have been separated from the river by levees and transformed to agricultural surfaces or housing areas (Kundzewicz et al., 2005).

- There is good evidence that **offshore wind farms (+)** and their energy production are impacted by climate change because, firstly, wind is a climate related atmospheric feature and secondly because the wind farms are, at least partly, a political (management) response to mitigate climate change (Tobin et al., 2015; Tobin et al., 2016). With increasing mitigation activities worldwide, we can expect a considerable increase in offshore wind energy production (ECDGE, 2019). Although it is not clear whether the harvested wind energy per unit will be higher in the future due to the



uncertainty of wind projections (Rusu, 2020; Christensen et al., 2021), the number of wind farms will increase in the future due to a politically driven shift to renewable energies and the limited space and low acceptance for land-based wind energy devices. Offshore wind farms may in turn have a certain impact on the regional climate by absorbing atmospheric energy on the regional scale. There is, however, little information on the magnitude of this effect (e.g. Siedersleben et al., 2018; Lundquist et al., 2019). Rising sea levels may have an impact on offshore wind farms, but they are probably not affected severely as there is presumably a sufficient safety margin calculated for storm surges within the life span of a structure. The general perception is that interaction between the foundation and the surrounding soil is a significant source of uncertainty in estimating the safety margins of support structures (Smilden et al., 2020).

- There is strong evidence that **shipping (+)** is heavily affected by climate change. Perils at sea for ships are all climate sensitive, ranging from storms, waves, currents, ice conditions, visibility to sea level affecting navigational fairways. Winter navigation is less impeded as a drastically decreasing winter sea ice cover is projected, but as winters with ice cover can also occur in the future (albeit less frequent), precautions for a safe winter shipping (e.g. the provision of ice breaking vessels in the eastern and northern Baltic Sea) cannot be abandoned. Also, search and rescue missions in winter may increase because engine power may in the future be adapted to the lower expected ice cover and stringent energy efficiency requirements set by the International Maritime Organization (IMO). Inland and archipelago shipping is impacted by floods and depth changes of rivers or straits, which may prevent normal vessel operations during exceptional periods. Further aspects affecting shipping are a potential increase in leisure boating with increasingly warm and longer summers in the Baltic Sea, and different noise propagation through warmer water. Regulations to reduce the $SO_x$ concentrations in air emissions by large ships involve scrubbing, i.e. the stripping of the contaminated combustion air with seawater. The stripping efficiency depends on the alkalinity of the sea water, which eventually ends up contaminated in the Baltic Sea (Endres et al., 2018; Teuchies et al., 2020), and may increase acidification (Turner et al., 2018). Possible impacts by sea level change on shipping could be the modification of fairways/shipping routes. On the one hand, shallow passages may get deeper and wider in the future (Meyers and Luther, 2020) and passages may be safer through shallow and dangerous fairways like the Kadet channel. Increasing water depths at bottlenecks would allow deeper drafts or higher loadings of large vessel, but ship size is constantly increasing, which may offset this hypothetical effect (Lu and Yeh, 2019). On the other hand, harbors and docking terminals will need to be adapted to higher water levels and possible changes in the sediment transport patterns. The associated threats of direct wave attack and overtopping may require substantial modifications of the existing breakwaters (Contestabile et al., 2020), also resulting in an increase in operational shutdowns and subsequent economic losses (Izaguirre et al., 2021). There has already been major damage and disruption to ports across the world from climate-related hazards and such impacts are projected to increase in the years and decades to come (Becker et al., 2018).

- Climate change impacts on the degradation and distribution of chemical **contaminants (+)** include an array of processes. Changing environmental temperatures affect diffusive partitioning between environmental phase-pairs such as air-water, air-aerosols, air-soil, air-vegetation, leading to a different distribution between environmental compartments, like increased volatilization from seawater and soil to air (Macdonald et al., 2003; Noyes et al., 2009). Increasing temperatures can enhance photolysis, hydrolytic degradation and biodegradation of organic contaminants (Noyes et al., 2009). Atmospheric transport and air-water exchange can be influenced by changes in wind fields and, to a lesser extent, wind speeds (Lamon et al., 2009; Kong et al., 2014). Changing precipitation patterns influence chemical transport via atmospheric deposition (rain dissolution and scavenging of particles, Langner et al., 2005; Armitage et al., 2011) and runoff, transporting terrestrial organic carbon and contaminants associated with this carbon (Ripszam et al., 2015). As ice cover in lakes and the sea decreases, more organic contaminants may volatilize to the



atmosphere (Macdonald et al., 2003; Undeman et al., 2015). Extended vegetation periods together with a reduced ice cover in the coastal zone allow both planktonic and benthic organisms to accumulate nutrients and toxic substances for a longer period. It has been estimated that, in an ice-free year, the average mercury concentration can be

substantially higher in phytoplankton and macrophyto- and zoobenthos compared to ice winters (Bełdowska et al., 2016a; Bełdowska and Kobos, 2016). As a result, a greater load of mercury can be remobilized from sediment to benthic organisms (Bełdowska, 2015; Bełdowska et al., 2015). An increase in air temperature, especially in the late autumn-winter-early spring season, contributes to the reduction of coal combustion and consequently to a decrease of toxic metal emissions and other combustion products like dioxins and polyaromatic hydrocarbons (PAHs), as

compared to colder winters (Bełdowska et al., 2016a; Bełdowska, 2015). Still, reductions in emissions have much stronger effects on concentrations of all contaminants than climate change (Simpson et al. 2015).

- **Dumped military material (?)** may be a great danger for the Baltic Sea in the future as poisonous material is expected to leak due to corrosion of hulls. This process may be affected by climate change. Due to longer vegetation periods in a warmer climate, the extended transfer of carcinogenic degradation products of explosives may take place for a

larger part of the year. Corrosion rates are temperature- and oxygen dependent, so that good ventilation of dumping sites can be expected to enhance corrosion rates (Silva and Chock, 2016). Warming can significantly affect munitions in shallow waters, which were mostly used as dumpsites for conventional warfare material. There, ammunition shells as hard metal objects as substrates for colonization in soft sediment areas can increase the local biodiversity of sessile species, but the chunks of organic compounds used as explosives can also attract primary and secondary producers

as a source of nutrients, followed by various biofilm grazers. Longer vegetation seasons may contribute to oxygen deficiencies, which may reduce arsenic constituencies into more mobile and toxic arsenic species (Czub et al., 2021).

- There is no evident direct impact of climate change on **marine litter** or **microplastics (?)**. Still, there may be a connection via increased temperature- and photolysis dependent degradation and dissolution rates of microplastics, and on the distribution due to changing currents. Furthermore, changes in precipitation and frequency of storm events

may affect microplastic concentrations in the Baltic Sea due to changes in microplastic emissions, e.g. via stormwater runoffs, deposition and sediment resuspension.

- There is an impact of climate on **tourism (+)** at the Baltic Sea coasts (Braun et al., 1999; Seetanah and Fauzel, 2019). A warmer climate with projected longer and warmer summers is clearly beneficial for most touristic activities in the Baltic Sea region (swimming, diving, sun bathing, surfing, boating, fishing), with its moderate to subarctic weather

conditions (Nicholls and Amelung, 2015; Perch-Nielsen et al., 2010). Exceptions are the ice-related activities in the northernmost parts in winter, like ice fishing and skating, which is expected to be less possible in the future. There are negative implications of climate change which are relevant for tourism, e.g. potentially deteriorating water quality in coastal waters due to hypoxia and algal blooms (Nilsson and Gössling, 2013; Olofsson et al., 2020), novel toxic algae (Engstrom-Ost et al., 2015), and growth of infectious bacteria (e.g. *Vibrio*) (Baker-Austin et al., 2012). Beaches

may suffer as well (Haller et al., 2011).

- **Coastal management (+)** as the process to mitigate problems in the face of multiple uses of coastal spaces and services is strongly challenged by climate change (Sanchez-Argilla et al., 2016). There is strong evidence that climate change heavily affects coastal structures through sea level rise (Nicholls, 2011) and intensified coastal erosion (Toimil et al., 2017). Storm surges, which run up higher with rising sea level (Needham et al., 2015; Hague et al., 2020;

Stephens et al., 2020) as well as changing current patterns (Nagy et al., 2019) and sediment relocations (Soomere and Viška, 2014) endanger levees, groynes and other coastal structures, and call for coastal management decisions to cope with these changes (Le Cozannet et al., 2017). Harbours and cities are strongly affected, so that there is considerable economic (Di Segni et al., 2017) and ecological (Naylor et al., 2012) value at stake. Beaches as spaces for recreation with multi-billion value in the Baltic Sea only (Czajkowski et al., 2015) and coastal biotopes are under pressure as





well (Harff et al., 2017a; Vitousek et al., 2017; Vousdoukas et al., 2020). Sea level change in particular has a very
        strong impact on coastal management and the defense structures like levees and groynes, and generally on the
        management of low-lying coastal regions (Hoggart et al., 2014). Also coastal cities (Balica et al., 2012) and harbours
        (Sierra, 2019) are highly vulnerable to sea level rise and require management actions. Different vulnerabilities and
        urgencies towards sea level rise require different management approaches in different countries, and between the
southern and the northern regions (e.g. Harff et al., 2017a, Harff et al., 2017, Støttrup et al., 2017): resilient, high,
        rocky coasts with land uplift dominate in the north (Ristaniemi et al., 1997), while vulnerable, low sandy coasts, soft
        cliffs and a slight land subsidence are strongly endangered in the southern regions (Zeidler, 1997).

**5.2 Coastal processes**

        Coastal processes generally describe the impact of the sea on the direct coastline it is in contact with. This includes waves,
currents, sediment translocations and coastal erosion, accretion and silting processes. Different coastal types (sandy beaches
        and soft cliffs vs. rocky coasts) are differently affected by these processes.
        About half of the shores of the Baltic Sea are sedimentary (Harff et al., 2017b) and thus susceptible with respect to different
        hydrometeorological loads. The submerged and soft coastal relief of the southern Baltic Sea area is under most threat (Labuz,
        2015). The other half are either extremely resistant bedrock (granite) or very slowly changing limestone shores. Most of the
sedimentary shores suffer from erosion (Pranzini and Williams, 2013) and only relatively short sections (most notably
        Denmark) exhibit accretion (EMODnet Geology, 2021). Many shore segments exhibit rapid retreat rates (e.g. up to 2.5 m yr$^{-1}$
        in the Neva Bight, Ryabchuk et al., 2012) that may have large impacts on the coastal infrastructure and cause extensive loss
        of land. For example, the projected shoreline retreat in some sections of the Pomeranian Bay may reach 100 m by the end of
        the 21$^{st}$ century (Deng et al., 2015) and 30–40 m on the northern shore of the Neva Bight (Leont'yev et al., 2015).
Technically, the major supplier of energy to the nearshore and the driver of sediment transport are surface waves. Even quite
        small waves contribute to sediment transport processes. The most intense alongshore and cross-shore sediment transport
        usually occurs in the nearshore (surf and swash zones) during very strong wave storms. The most rapid shoreline changes
        (both erosion and accumulation) occur during short term elevated average water level events when high waves approach the
        shore from an unusual and unfortunate angle and attack the unprotected sediment. This happens during high storm surges when
waves affect unprotected sediment. The properties of this energy supply and the resulting wave-driven transport in large parts
        of the Baltic Sea shores are described in (Soomere and Viška, 2014; Kovaleva et al., 2017; Björkqvist et al., 2018) and
        generalized to the entire Baltic Sea context in (Hünicke et al., 2015; Harff et al., 2017a; Harff et al., 2017b).
        Such events are infrequent in the Baltic Sea that hosts a highly intermittent wave climate. On the one hand, as little as 1% of
        the total annual energy arrives on the shores of the eastern Baltic Sea within the calmest 170–200 days. On the other hand,
60% of the annual energy arrives within 20 days with relatively high waves and as much as 30% of the energy within the 3–4
        stormiest days (Soomere and Eelsalu, 2014).
        A simple consequence of this kind of intermittency of wave fields, anisotropy of wind and wave patterns (Soomere, 2003),
        and the complicated geometry of the shoreline is that many beaches are in an almost equilibrium state (Soomere and Healy,
        2011) for long periods, and their evolution is a step-like process. This means that a few events resulting in rapid changes (when
strong waves arrive from a specific direction during high water level events) occur on a background of very slow changes (that
        cover most of the year and require high-resolution measurements to detect; Eelsalu et al., 2015). Thus, the properties of single
        storms and the timing of storms in sequences may become decisive in the evolution of the coast (Coco et al., 2014; Dissanayake
        et al., 2015).
        Hydrodynamic forces effectively reshape the shore, particularly when no ice is present and the sediment is mobile (Orviku et
al., 2003; Ryabchuk et al., 2011; Nielsen et al., 2020). No ice means no protection against severe waves and high water levels
        (Barnhart et al., 2014). Storm surges are much higher during ice-fee time than on the shore of even partially ice-covered water



bodies. Particularly dangerous are thus situations when strong waves reach unprotected and unfrozen mobile sediment during extreme storm surges on higher sections of the shore that are out of reach of waves during periods of usual water levels (Orviku et al., 2003).

Potential changes in the intensity and/or direction of wave-driven processes at the shore impact many aspects of society. The potential additional sedimentation of fairways and river mouths and erosion of shores in the vicinity of built environments have direct economic consequences. The loss of stability of beaches (Haller et al., 2011) may severely damage communities that offer recreational services. Unexpected changes in the transport system may lead to, e.g., intense cross-shore transport of very fine sediment to vulnerable areas such as spawning areas, with substantial consequences to fish stocks.

Many Baltic Sea shores suffer from a deficit of fine sediments (Pranzini and Williams, 2013), which means that they are very vulnerable because of the large hydrodynamic loads affecting them, being geomorphically young and poorly supplied in sand. However, a number of beaches of the Baltic Sea are explicitly or implicitly stabilized by the (mis-)match of the directions of predominant strong storms and geometry of the shoreline: they are relatively infrequently hit by storms and may have stable sections in their bayheads. Also, systematic synchronization of high water levels and strong waves may implicitly protect some

beaches (Soomere et al., 2017). As waves in the Baltic Sea are relatively short and thus their energy spectrum relatively narrow, the surf zone is narrow and wave run-up phenomena are usually less powerful than on the open ocean shores (Didenkulova et al. 2008). Postglacial rebound in some parts of the sea additionally stabilizes the affected beaches. As a consequence, many Baltic Sea beaches with very small amounts of sand are in a fragile almost equilibrium state. As sediment transport direction and its convergence (accumulation) and divergence (erosion) areas are highly sensitive with respect to the wave approach

direction, even a minor climate-change-driven rotation of the predominant wind directions over the Baltic Sea may substantially alter the structural patterns and pathways of wave-driven transport and functioning of large sections of the coastline (Viška and Soomere, 2012).

Direct subtractions from the coastal sediment budget can have significant consequences. Time is "running out for sand" (Bendixen et al, 2019), and this is the case in the Baltic Sea as well. Sediment extraction for the purposes of beach

renourishment (Karaliūnas et al, 2020) or aggregate (Schwarzer, 2010) can negatively affect shorelines if the source zone is at depths less than the beach profile closure (López et al, 2020). A major knowledge gap is the scarcity and low accessibility of data about changes to the coastline (Muis et al., 2020).

**Impacts of coastal processes on other factors**


- Coastal current systems and sediment relocations may possibly influence local **hypoxia (?)** in coastal embayments. Biogeochemical processes in the water column and sediments are the cause for hyp- and anoxia but coastal processes may be important for providing certain conditions (e.g. Conley et al., 2009; Kemp et al., 2009).
- Coastal processes, next to riverine inputs, may affect alkalinity and **acidification (?)** in some regions (Krumins et al.,

2013; Gustafsson et al., 2014; Gustafsson and Gustafsson, 2020).

- **Submarine groundwater discharge (+)** can be defined as a coastal process. It is very much affected by other coastal processes like topography-driven flow, wave set-up, sea level rise and currents (Burnett et al., 2006; Kłostowska et al., 2020).
- Coastal processes may affect **marine ecosystems (?)**, in particular coastal habitats. On the one hand, the loss or

weakening of coastal barriers leads to saltwater intrusion into adjacent low-lying areas. Increased salinity levels moving inland have disrupted many wetland ecological functions (Davidson et al., 2018). On the other hand, improved water quality via coastal retreat may mitigate losses of seagrass from sea level rise (Luijendijk et al., 2018). Major changes in the sediment transport system may lead to increased (but not yet quantified) pressure to vulnerable areas, e.g., via transport of very fine sediment to spawning areas.



- Coastal erosion can affect **land use (+)** close to the coast, beaches and close to endangered cliffs where erosion often increases risk for landslides (Collins and Sitar, 2008). These aspects are well understood for the open ocean shores (e.g., massive changes in the area of fish ponds owing to erosion and accretion, Kalther and Itaya, 2020) but have not been addressed in the Baltic Sea context. Accretion can generate new sand spits (Tõnisson et al., 2008; Ryabchuk et al., 2011) and associated coastal lagoons, marshes or polders and may lead to massive silting of harbours. On a long time scale the morphology and its associated land use are subject to change.

- Similarly, it can be significant for **agricultural (+)** fields and forests very close to the sea or a cliff endangered by coastal erosion. These processes are generally proportional to the shoreline retreat rate but may threaten important areas if, for example, a coastal barrier is lost. Coastal currents affect the distribution of land-borne **nutrients** (e.g. near-shore currents in the Pomeranian and Gdansk Bay, with little mixing with open waters (e.g. Pastuszak et al., 2003; Voss et al., 2005a).

- There is no direct evidence for an impact on **aquaculture (?)**, but erosion and sediment translocations may affect open aquaculture cages in coastal locations.

- There may be an impact of coastal processes on coastal fish habitats and **fishing (?)** grounds. Intense cross-shore transport of very fine sediment to spawning areas may adversely affect fish stock, but this potential effect has not been quantified.

- Coastal processes can possibly affect the regulation of **river (+)** mouths and estuaries, and the associated sediment distributions in these river mouths. They may result in building a sill at the river mouth that partially or totally blocks the river flow, similar to ice jam (Lindenschmidt et al., 2019). This blocking may lead to flooding of the areas around the river mouth and to a degradation of water quality in the closed estuary. This phenomenon is frequent on open ocean shores (e.g. Thom et al., 2020). It usually occurs in wave-dominated environments, which are scarce in the Baltic Sea and are represented, e.g., by the Narva River (Laanearu et al., 2007).

- **Offshore wind farms (+)** can be affected by coastal processes if they are close to the coast, e.g. through currents and sediment transport. Coastal currents may lead to wake effects and scouring, and problems with the stability of pillars and the sediment distribution in the lee of the wind turbines. The interaction between the foundation and the surrounding soil is a major source of uncertainty in estimating the safety margins of support structures (Smilden et al., 2020).

- There is a large influence of coastal processes on **shipping (+)** routes close to the coast and in intracoastal waterways, currents, the generation of shallows and sand spits into fairways either naturally or forced by different coastal engineering structures (Davis and Barnard, 2003), resulting in re-location and the need for dredging of fairways.

- Coastal processes may affect the distribution of chemical **contaminants (?)** and toxic heavy metals through sediment translocations and the modification of habitats. Coastal erosion contributes significantly to the inflow of substances to the sea (including toxic chemicals) and wave-driven alongshore transport may carry such substances to great distances. The concentrations of toxic metals in the eroding cliff are mostly low (Kwasigroch et al., 2018), but a total load of metals introduced in this way is significant, due to the large amounts of eroded sediments (Bełdowska et al., 2016b). Additionally, episodes of erosion occur several times a year, leading to large loads of toxic substances in a relatively short time. Metals introduced in this way to the environment are often bioavailable (Kwasigroch et al., 2018).

- **Chemical munition (?)** dumpsites are located far from the coasts, but current systems may affect the distribution of leaked substances. Conventional munition dumpsites can however be located in shallow water at a close distance to the shore, i.e. in Kiel Bight. As degradation products and trace metals originating from munitions have been detected in the sediments there, coastal processes may enhance the spreading of those contaminations to adjacent areas or increase their bioavailability (Gębka et al., 2016; Bełdowski et al., 2019; Maser and Strehse, 2020).



- Coastal processes have no impact on the generation but a significant impact on the distribution and accumulations of beach wrack (Suursaar et al., 2014) and **litter (+)** on the shoreline (Esiukova, 2017; Haseler et al., 2020; Urban-Malinga et al., 2020).


- There is a very strong impact of coastal processes on **coastal management (+)**. Coastal processes like erosion, sediment translocation, accretion etc. are primary drivers for coastal management (Hapke et al., 2016), which includes the provision of coastal defenses against erosion and inundation by engineering and planning. Sand and gravel extraction from nearshore areas creates an intrinsic danger to the shoreline by removing sedimentary material in the

affected area, and increasing the wave energy reaching the shoreline. Mining of sand or gravel is usually considered acceptable in terms of its effect on the shoreline dynamics if the extraction site is well offshore from the closure depth. For example, mining in Polish waters (Uscinowicz et al., 2014) was performed at a depth twice of 15 to 30 m and several operations in Tromper Wiek, NE of the Island of Rügen, at depths from 14 to 21 m (Kubicki et al., 2007). As these sites had a depth more than twice as large as the closure depth in the open Baltic Sea (Soomere et al., 2017), it

is natural that no impact was reported on the coastal processes in the vicinity. Several operations providing sand for beach nourishment at Palanga, Lithuania, used sand extracted from the Baltic Sea floor at a depth of >50 m (Pupienis et al., 2014). Similarly, no direct impact on coastal processes was reported after extraction in 1997 of approximately 320 000 m³ of sand from a site located about 25 km off Wustrow, Germany (Krause et al., 2010); however, negative impacts on the coastal sediment budget cannot be excluded (Diesing et al., 2006).

**5.3 Hypoxia**

Oxygen ($O_2$) is one of the central biogeochemical elements on Earth. It is produced during photosynthesis and is part of metabolic energy production in almost all organisms. Consequently, its distribution in the oceans is essential for life in the sea. Oxygen in seawater is primarily produced by phytoplankton, but its concentration is also strongly affected by the gas exchange through the sea surface. In most parts of the world ocean, seawater is a source of oxygen; it is estimated that 50-85% of the

atmospheric oxygen is produced in the oceans (NOOA, 2021). Oxygen can be carried to non-productive deeper layers by processes like convection, downwelling and diffusion, so that many parts of the deep ocean are rich in oxygen (Jahnke and Jackson, 1992). Respiration by heterotrophic organisms (bacteria, protists, micro-and macrozooplankton, larger animals) and nitrification (the microbial process in deeper water layers to produce nitrite and nitrate from ammonium) are the sinks for oxygen in the water column. Poor ventilation and enhanced respiration in deeper water layers may lead to *hypoxia*, which is

defined as the approximate oxygen concentration too low to allow the existence of complex multicellular life (<2mL L⁻¹), or even *anoxia* denoting the complete absence of oxygen with the concomitant microbial production of hydrogen sulfide ($H_2S$). Hypoxia in the Baltic Sea (as in other highly productive and stratified regions of the world ocean) is a common phenomenon. It has been shown that in the deep basins of the Baltic Sea (mainly in the Bornolm, Gdansk and Gotland basins) hypoxia, and anoxia in particular, has increased considerably over the course of the past century (by a factor of 10 since the beginning of

the 20[th], Carstensen et al., 2014), and has reached a hypoxic area of roughly 32% of the Baltic Proper in 2019 (Hansson et al., 2020). This in principle goes parallel with the industrialization and eutrophication of the Baltic Sea (Zillén et al., 2008; Conley et al., 2009; Carstensen et al., 2014). Thus, eutrophication has been identified as the foremost cause of this development. However, the decreasing nutrient inputs in recent decades and years have not yet resulted in a re-oxygenation because of legacy nutrient pools, i.e nutrient pools which are temporarily accumulated in the sediments and released to the water column slowly

(McCrackin et al., 2018a).

Major Baltic Inflows (MBIs), i.e. irregular inflows of saline, oxygen-rich deep water through the Kattegat and Belts and Sounds into the deep basins of the western and central Baltic Sea, do not have any long-term positive impact on the oxygen conditions in the deep layers, as the enhanced salinity (and density) of these waters increase stratification and drastically reduce the vertical exchange beween surface and deep waters. Recent studies show that oxygen consumption rates after MBIs have





significantly increased during the last decades, causing hypoxic conditions to prevail or even deteriorate (Meier et al., 2018). Thus, saltwater inflows generally contribute to a reduction of oxygen concentrations in the deep basins (Meier et al., 2017). The peculiar geography and hydrography of the Baltic Sea (sequential deep basins, strong stratification, low and irregular exchange with the North Sea) and strong anthropogenic pressure (nutrient loads from land) generate biogeochemical conditions which have a strong impact on the pelagic ecosystem of the Baltic Sea, manifesting in the occurrence of regular blooms of

filamentous cyanobacteria. As the oxidized nitrogen nutrient compounds (nitrite and nitrate) which can be easily taken up by phytoplankton get depleted in the surface layer during the spring bloom, phytoplankton growth ceases and the pelagic system switches to regenerated production which is fueled by reduced nutrients of the microbial loop (Kuosa, 1991). Under these conditions, the filamentous cyanobacteria have an advantage over other phytoplankton species, as they are able to use the molecular nitrogen ($N_2$) dissolved in surface waters by nitrogen fixation (Larsson et al., 2001). In oxygen-free surface

sediments, bound phosphorus is converted to its labile (and bioavailable) form and released. Simultaneously, at hypoxic conditions denitrification occurs, which lowers nitrate concentrations. This generates shifts in the N:P ratio towards a surplus of bioavailable phosphorus (Vahtera et al., 2007).

This shortage of bioavailable nitrogen species together with a surplus of phosporus gives $N_2$-fixing cyanobacteria a significant advantage over all other algae in these conditions. Furthermore, cyanobacteria blooms enhance eutrophication and organic

matter supply to the deep water layers, which increases oxygen consumption. In the end this mechanism forms the so-called "vicious circle" - the positive feedback self-supporting the Baltic Sea eutrophication (Vahtera et al., 2007; Savchuk, 2018). Thus, the oxygen deficiency conditions in the deep basins have severe consequences for the nutrient budgets and the whole ecosystem of the Baltic Sea. Extensive cyanobacterial blooms are also common in sea areas with similar biogeochemical conditions, extensive hypoxic or anoxic waters, e.g. in the Black Sea, or parts of the tropical oceans (equatorial East Pacific,

Arabian Sea) (Terenko and Nesterova, 2015; Westberry and Siegel, 2006).

In addition to the deep basins, also many coastal waters in the Baltic Sea increasingly suffer from temporal or even permanent an- or hypoxia (Conley et al., 2011, Jokinen et al., 2018). The main causes for coastal hypoxia are nutrient releases from the surrounding land, seasonal thermal stratification and reduced water circulation in coastal embayments.

Climate change affects the extent of hypoxia in the water column. Oxygen deficiencies have increased in the deep basins and

in coastal waters over the past decades (Carstensen et al., 2014; Caballero-Alfonso et al., 2015). Climate warming has contributed to this trend through lower $O_2$ dissolution in water, increased respiration rates and intensified stratification. Eutrophication with subsequent extensive biomass production and remineralization has been identified as the primary driver (Meier et al., 2019b). Sea level rise is also expected to contribute to hypoxic areas, through the potential intensification of saltwater inflows and subsequent stronger stratification. It may be a considerable factor in the future (Meier et al., 2017).

It remains to be seen how quickly the effective nutrient abatement measures of the Baltic Sea Action Plan (HELCOM, 2007) take effect, and hyp-and anoxia, extensive phosphorus release from the deep basins and subsequent extensive cyanobacterial blooms decrease. For more on hypoxia and the role of oxygen in the marine ecosystem of the Baltic Sea, see Kuliński et al. (2021).

**Impacts of hypoxia on other factors**

- Hypoxia is a direct consequence of eutrophication and water column stratification. Such conditions lead to the partial separation of enhanced organic matter production in the surface from its remineralization in the deep water layers and surface sediments. This enhanced respiration in the deep basins lowers $O_2$ concentration (hypoxia), but at the same

time increases $CO_2$ concentration, which contributes directly to **acidification (+)** (Melzner et al., 2013; Kuliński et al., 2021). On the other hand, total alkalinity (a measure of seawater buffer capacity) is generated during an anaerobic organic matter remineralization. Although most of those anerobic reactions are reversible when conditions change





back to oxic, processes like denitrification, and pyrite and vivianite formation are permanent sources of alkalinity, which counteract acidification (Kuliński et al., 2017; Kuliński et al., 2021; Gustafsson and Gustafsson, 2020). In the end, the relevance of these processes and their net effect on ocean acidification in the Baltic Sea (deep basins and coastal waters) remain unclear.

- Hypoxia and anoxia have a strong impact on **marine ecosystems (+)**. The biogeochemistry and redox state of the system is strongly affected though feedbacks on the nitrogen and phosphorus cycles with strong implications for the pelagic and benthic ecosystems, e.g. the occurrence and massive blooming of nitrogen-fixing, filamentous cyanobacteria, and implications for higher trophic levels, e.g. cod (Vahtera et al., 2007; Dippner et al., 2008).

- Oxygen plays a significant role in the nitrogen and phosphorus biogeochemistry. Nitrogen is removed by processes such as anaerobic ammonium oxidation (anammox) and denitrification and, alternatively, recycled through dissimilatory nitrate reduction to ammonia and nitrification. Hypoxia may alter nutrient dynamics by affecting the coupling of these pathways and further exacerbate the effects of eutrophication (Gustafsson et al., 2017; Savchuk, 2018). There is a feedback on internal **nutrient cycling (+)**, i.e. the internal release of phosphorus from anoxic sediments. So, hypoxia in this case would act as a driver for eutrophication, leading to favorable growth conditions for $N_2$-fixing cyanobacteria in summer when free bioavailable nitrogen is depleted in the surface waters. This is an important feedback mechanism, strongly affecting the biogeochemistry of the Baltic Sea, with repercussions on the food web structure. This feedback mechanism is sometimes called the "vicious circle", as the additional cyanobacteria biomass is respired mainly in deeper waters, thereby increasing the oxygen free zones and subsequently the phosphate release in the deep water (Vahtera et al., 2007).

- Coastal **aquaculture (?)** may be affected by local hypoxia, and in turn may also create hypoxic areas, by sedimented unused fish food which is then respired (Díaz, 2010).

- Hypoxic zones affect **fisheries (+)** though the impairment of fish production. The most substantial impacts are demonstrated for cod, where low oxygen has negative effects on egg production and survival (Köster et al., 2017). Furthermore, oxygen is considered to affect the growth and condition of the Eastern Baltic cod both directly and via regulating the availability of benthic food (Casini et al., 2016a; Neuenfeldt et al., 2020).

- Changing oxygen concentrations in bottom waters alter the redox conditions, which in turn may lead to remobilization of contaminants accumulated in sediments and influence their bioavailability. A re-colonization of hypoxic bottoms by benthic animals can lead to increased release of "archived" **contaminants (+)** in the sediments by bioturbation (Thibodeaux and Bierman, 2003; Granberg et al., 2008), but it is unclear how this transport compares to other processes (Kwasigroch et al., 2021).

- Corrosion rates of **dumped munitions (?)** are dependent on oxygen concentration, the presence of specific ions in near-bottom water, and the activity of microbial communities (Silva and Chock, 2016). Although lower oxygen concentrations inhibit corrosion rates, subsequent changes from oxic to anoxic and back to oxic conditions may accelerate corrosion, due to oxidation of hydrogen sulfide to sulfates. Therefore, periodic hypoxic events may speed up corrosion. Additionally, hypoxia can alter the degradation process of chemical warfare agents, leading to greater persistence of degradation products in sediments (Vanninen et al., 2020). At the same time, arsenic released from agents based on this metal in pentavalent form, maybe remobilized from sediments in trivalent form, which is more toxic to biota (Czub et al., 2020; Czub et al., 2021).

## 5.4 Acidification

Acidification is the product of anthropogenic $CO_2$ emissions, mainly due to fossil fuel combustion and changes in land use, e.g. by transferring forests to agricultural lands. The atmospheric $CO_2$ concentration has increased from about 277 parts per million (ppm) at the beginning of the industrial era (1750) to 405 ppm in 2017 (Quéré et al., 2018), and it is continuously





increasing. This accumulation illustrates that human activities like fossil fuel combustion and environmental changes (e.g. land use change) are responsible for more carbon inputs into the atmosphere than other Earth system compartments (land, ocean) can recycle. In the global carbon budget, averaged over the 1959-2017 period, 82% of the total emissions were due to fossil carbon dioxide emissions and 18% by land use change. Of the total emissions, 45% remained in the atmosphere, while 24% and 30% were taken up by the ocean and the land, respectively (Quéré et al., 2018). The global fossil emissions have increased

by a factor of three from the 1960s to 2008-2017. The related generation of atmospheric of oxidized sulphur and nitrogen compounds has resulted in "acid rain" and considerably contributed to the acidification of soils and freshwater bodies (Tranvik, 2021).

The oceanic uptake of $CO_2$ goes with the price of ocean acidification (Gattuso and Hansson, 2011). There has been increasing concern during the past two decades, and new observations have shown declining pH values in the ocean (IPCC, 2013; IPCC,

2019). Ocean acidification changes the ocean´s carbonate chemistry, resulting in a wide range of effects (Gattuso and Hansson, 2011), and the estimation of the potential damage and calculation of costs is a contentious issue (World Ocean Review, 2015). Marine acidification is also expected to affect coastal seas. However, the specific processes there are more complex, due to land-sea interactions such as river and drainage basin biogeochemistry, and effects from anoxic water and sediments. Whether coastal seas act as a source or a sink for atmospheric $CO_2$ depends on an intricate balance between air-sea $CO_2$ exchange,

terrestrial carbon loads (rivers), water exchange with adjacent basins and sediment fluxes.

The strength of acidification depends on the accumulation of acid over basic elements and the buffering capacity of seawater. This is illustrated in Figure 2, showing how changes in $CO_2$ concentrations and total alkalinity ($A_T$) drive the acid-base balance and seawater pH. Alkalinity can be defined as the excess of bases over acids in seawater, thus being a measure of its buffer capacity against acidification. The variation of total alkalinity concentrations is considerable in the Baltic Sea, with low total

alkalinities in the Gulf of Bothnia and the Gulf of Finland, and higher ones in the Gulf of Riga and the southern Baltic Sea. The variations in total alkalinity concentrations are presumably caused by the different geological structures in the Baltic Sea basins: the southern drainage basin is richer in limestone and hence alkalinity than the northern part, where granite dominates the bedrock (Hjalmarsson et al., 2008; Bełdowski et al., 2010).



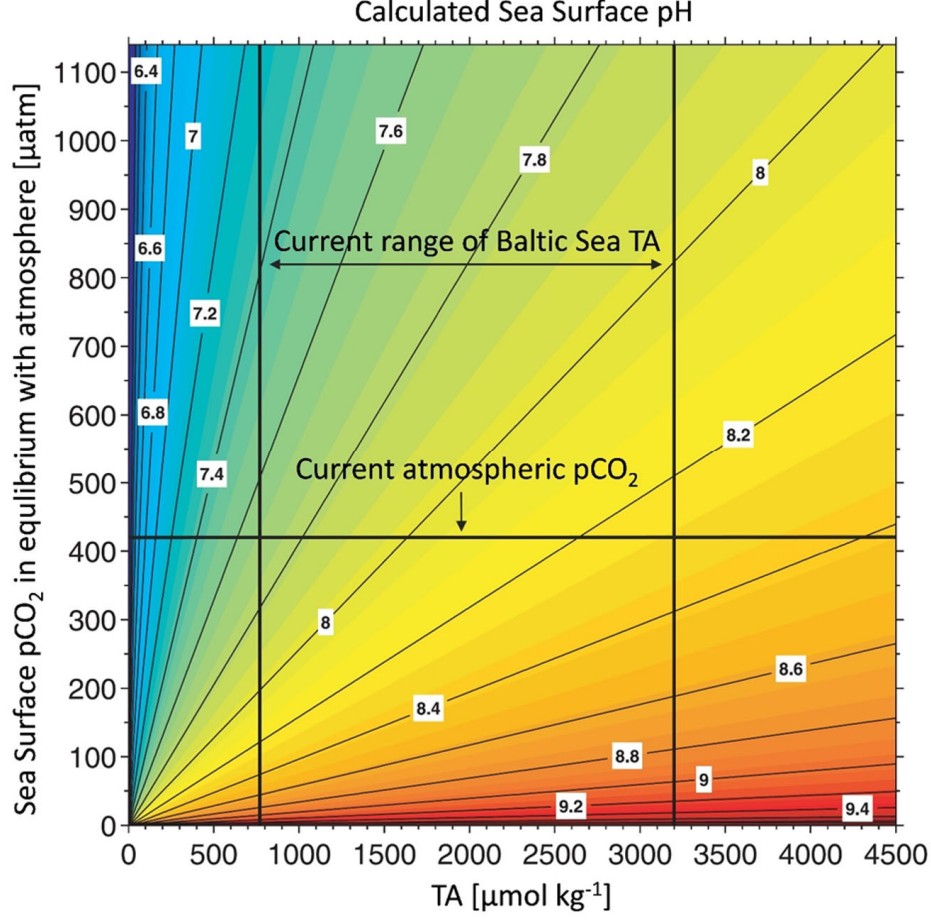


**Figure 2. The figure illustrates how total alkalinity and partial pressure of $CO_2$ (pCO₂ – a measure of $CO_2$ concentration) influence seawater pH. The pH values resulting from variability in pCO₂ and total alkalinity are marked with isolines and color code (from blue – low pH to red – high pH). Redrawn after Omstedt et al. (2010) and Kuliński et al. (2017).**


This shows the complex acid-base system in the Baltic Sea and the importance of riverine discharge in shaping seawater pH. Generally, total alkalinity is lower in the Baltic Sea (especially in the Gulf of Bothnia and the Gulf of Finland) than in the open ocean, except in the direct vicinities of the mouths of the continental rivers (south and south-eastern coast). Since lower alkalinity corresponds to lower buffer capacity, at first approximation the Baltic Sea can be considered more vulnerable to
acidification than oceanic regions. However, recent studies by Müller et al. (2016) reveal that total alkalinity in the Baltic Sea increases. The highest alkalinity increase, observed in the Gulf of Bothnia, entirely makes up for ocean acidification caused by $CO_2$ increase, whereas in the central part (Baltic Proper) ocean acidification is lowered by about 50%. Although the source of that increase is unknown so far, increase in weathering on land and anoxic alkalinity generation in sediments have been suggested.
The central role of the marine $CO_2$ system for biogeohemical processes in the Baltic Sea is discussed by Schneider and Müller (2018). The relative importance of the factors affecting acidification is difficult to disentangle as variability is often large (Wootton et al., 2008), and several time scales need to be considered. For example, modeling exercises indicate that increased





nutrient loads may increase the seasonal pH oscillations but may not inhibit future Baltic Sea acidification on longer time scales (Omstedt et al., 2012). This is further supported by a recent sensitivity study including aspects of eutrophication by Gustafsson and Gustafsson (2020).

Budget calculations imply that the neighboring North Sea and the surrounding drainage basin largely determine the fate of carbon entering the Baltic Sea (Kuliński and Pempkowiak, 2011). Gustafsson et al. (2017) estimated for the period 1980-2014 that net carbon loads from land and atmosphere to the Baltic Sea were mainly lost to the North Sea (90%), 6% was lost to the sediments and only 4% was taken up in seawater. In contrast, in their budget estimates, Kuliński and Pempkowiak (2011) estimated net outgassing of $CO_2$. Hence, it is not entirely clear if the Baltic Sea is or will be a net source or sink for $CO_2$.

Several studies are available on the quantification of the net $CO_2$ flux between the atmosphere and the Baltic Sea. For example, model calculations by Omstedt et al. (2009) indicate that, before industrialization, the partial pressure of carbon dioxide in water was above atmospheric values, with a net release of $CO_2$ to the atmosphere. Seasonal variability increased after industrialization and the onset of eutrophication, making the modern Baltic Sea both a sink (in summer) and source (in winter) of $CO_2$ to the atmosphere, depending on the season. In a model sensitivity study, Gustafsson et al. (2015) showed how outgassing or uptake due to air-water $CO_2$ fluxes depend on river loads of carbon, total alkalinity and nutrients, as well the freshwater import. Lansø et al. (2017) examined the importance of short-term variability of the air-sea exchange in an atmosphere model study, illustrating the need for better estimates of $CO_2$ fluxes when calculating net exchange in complex coastal seas.

Carbon dioxide is not the only fossil fuel combustion product contributing to acidification. Omstedt et al. (2015) examined the effects of historical atmospheric depositions of sulfur and nitrogen oxides as well as ammonium from land and shipping on the acid–base balance in the Baltic Sea, in the period 1750–2014. The results indicate that acidification due to atmospheric deposition of acids peaked around 1980, with a pH decrease of approximately 0.01 pH units in surface waters, which is an order of magnitude less than the acidification estimated due to increased atmospheric $CO_2$. Then again, the contribution of shipping to acidification was found to be one order of magnitude less than that of land emissions. Interestingly, the pH trend due to atmospheric acids has started to reverse due to reduced land emissions, although the effect of shipping is ongoing (Omstedt et al., 2015). While shipping is expected to become a major source of strong acid deposition to the Baltic Sea by 2050, the long-term effect on the pH and alkalinity is projected to be significantly smaller than estimated previously. A significant contribution to this difference is the efficient export of acidified surface waters to the North Sea (Turner et al., 2018). Despite decreasing emissions on land, soil and freshwater acidification due to fuel combustion and related atmospheric sulphur and nitrogen deposition ("acid rain") should be taken in to account in coastal regions (Tranvik, 2021 and related Ambio Special issue) to estimate how this may affect coastal acidification.

A hypothetical further increase in nutrient loads (notwithstanding the Baltic Sea Action Plan) would increase biological primary production and the surface pH (meaning reduced acidification) (Borges and Gypens, 2010) as well as its seasonal amplitude (Omstedt et al., 2009). On the other hand, increased mineralization in intermediate and deeper layers could increase acidification (Lui et al., 2015; Cai et al., 2011). Eutrophication may also increase the extent of anoxic deep waters (Díaz and Rosenberg, 2008), which could increase the total alkalinity and thus the buffer effect (Edman and Omstedt, 2013; Ulfsbo et al., 2014; Kuliński et al., 2017; Gustafsson and Gustafsson, 2020). Anaerobic alkalinity generation in coastal sediments may be a factor (Thomas et al., 2009). Hence, the temporal and regional range of acidification in the Baltic Sea is large and depends on a complex interplay of different factors, and trends in the Baltic Sea are not straightforward (Kulinski et al., 2021). There is a considerable ncertainty related to the sources and regional distributions of alkalinity, which in the Baltic Sea largely determines the strength of acidification (Gustafsson et al., 2019). More details on acidification and alkalinity and their roles in the biogeochemistry of the Baltic Sea are given in Kulinski et al., 2021.

**Impacts of acidification on other factors**


- Acidification may affect the **marine ecosystems (?)**, by disturbing the metabolism of various organisms. Still, there has been no clear evidence of the extent of *in situ* impacts in the Baltic Sea (Brander and Havenhand, 2012; Paul et al., 2016). Effects of lower pH on various organisms have been mainly derived from mesocosm experiments. The impact on *in situ* ecosystems of the different regions of the Baltic Sea is mainly low or unknown (Havenhand, 2012; Doo et al., 2020).

- It is unknown if there is an impact of acidification on **aquaculture (?)**. There is no clear evidence for a considerable impact of moderate acidification (realistic in the near future) on the growth of blue mussels (Thomsen et al., 2010)

- The same holds true for a potential impact of acidification on **fisheries (?)** in the Baltic Sea, e.g. by affecting calcifying food organisms for fish larvae. Otoliths are made from calcium carbonate, and there have been shown no apparent effects by lower pH (Coll-Lladó et al., 2018; Di Franco et al., 2019; Hamilton et al., 2019). There are indications that acidification may affect the auditory behavior of fish (Simpson et al., 2011), but it is entirely unclear if these findings can be transferred to Baltic Sea species.

- Changing pH can directly influence the speciation of dissociating **chemical contaminants (?)**, but the extent and significance for the environment are largely unknown. Marine acidification in surface, deep and sediment layers may change how contaminants like heavy metals and aluminum dissolve in the water body and circulate in the environment, but the effects are largely unknown.

### 5.5 Marine Groundwater Discharge

Submarine groundwater discharge (SGD) has received increased attention over the past decades due to growing concerns over its role as a potential source of chemical substances to the coastal ecosystem (Burnett et al., 2003; Taniguchi et al., 2019). SGD is driven by overlapping land and marine drivers (e.g. precipitation, currents, tides, waves, density gradients) which complicates SGD identification, quantification, and evaluation of SGD impact on marine ecosystems (Taniguchi et al., 2019). Due to its complex nature, by definition, SGD includes all flow of water across the seabed to the water column, regardless of fluid composition or driving force (Burnett et al., 2003). It includes both fresh groundwater discharge derived from terrestrial recharge (e.g. meteoric, precipitation-derived, fresh submarine groundwater discharge, FSGD) and recirculated seawater (salty groundwater, recirculated submarine groundwater discharge, RSGD) (Burnett et al., 2006). Usually, recirculated seawater dominates SGD.

Meteoric groundwater and salty groundwater discharge usually show different geochemical characteristics. The composition of meteoric groundwater discharge mostly depends on local hydrogeologic conditions. The discharge of recirculated groundwater, which can consist of recirculated seawater or a mixture of fresh and recirculated groundwater, goes through various biogeochemical processes. Some chemical substances like nutrients, dissolved organic and inorganic carbon have concentrations several orders of magnitude higher in groundwater than in surface water. Therefore, even if the SGD rate is low, the chemical substances flux via SGD can be relatively important, as groundwater in coastal aquifers tends to be enriched in various chemical substances.

SGD has been investigated in the Gulf of Finland (Viventsowa and Voronow, 2003; Virtasalo et al., 2019); the Gulf of Bothnia (Krall et al., 2017); the Eckernförde Bay (e.g. Schlüter et al., 2004; Scholten et al., 2015); the Bay of Gdańsk and the Bay of Puck (e.g. Szymczycha et al., 2012; Kotwicki et al., 2014; Donis et al., 2017; Kłostowska et al., 2020; Szymczycha et al., 2020a; Szymczycha et al., 2020b). The first assessment of SGD significance in the Baltic Sea was made by Peltonen (2002) who estimated the fresh SGD (FSGD) to the entire Baltic Sea, using a combination of hydrological and hydrogeological methods. The amount of FSGD to the Baltic Sea, compared to total runoff was small — around or even less than 1% (around 4.4 km$^3$ yr-$^1$). However, this calculation is based on hydrogeological methods and modeled without validation by offshore sampling, so it relates only to FSGD and does not include the recirculated water component. Krall et al. (2017) estimated SGD





at Forsmark, Gulf of Bothnia to range from 5.5 (± 3.0) x $10^3$ m$^3$ d$^{-1}$ to 950 (± 520) x $10^3$ m$^3$ d$^{-1}$, using Ra isotopes. These rates
are up to two orders of magnitude higher than those determined from local hydrological models which consider only the fresh
component of SGD. Kłostowska et al. (2020) obtained similar results in the Bay of Puck, southern Baltic Sea. Recently
estimated SGD rates range from 1.4 x $10^6$ m$^3$ d$^{-1}$ to 11.3 x $10^6$ m$^3$ d$^{-1}$ which are 17 to 130 times higher than the results obtained
by Piekarek-Jankowska (1994), including only the freshwater component of SGD. The obtained fluxes were several times
higher than the surface runoff. Additionally, local studies in the Bay of Puck suggest that SGD can be an important source of
methane, dissolved organic and inorganic carbon (Szymczycha et al., 2012; Kotwicki et al., 2014; Donis et al., 2017), nutrients
(Szymczycha et al., 2012; Szymczycha et al., 2020a) and trace metals (Szymczycha et al., 2016).

The ecological impact of submarine groundwater discharge on coastal surface sediments was also assessed. In meiofauna
assemblages in the Bay of Puck, a significant decline of certain meiofaunal taxa (mainly nematodes and harpacticoids), as well
as altered temporal patterns and a changed small-scale vertical zonation was demonstrated (Kotwicki et al., 2014).

Recently, a study was conducted on the annual variability of nutrient loads, concentrations and cycling by SGD in the Bay of
Puck (Szymczycha et al. 2020a). The estimated seasonal and annual loads of both dissolved inorganic nitrogen and phosphates
via SGD to the Bay of Puck were the most significant source of nutrients. It can be assumed that SGD could also be a significant
source of nutrients in other Baltic Sea regions, and therefore affect biogeochemical processes in the coastal zone. It is well
established that nutrient loads from land are filtered by biogeochemical processes and enter the open Baltic Sea in a modified
form (Asmala et al., 2017; Edman et al., 2018). As the effectiveness of the coastal zone is important for a proper understanding
of open sea eutrophication, SGD and accompanied nutrients fluxes should be considered in models characterizing the
biogeochemical process in Baltic Sea coastal areas. Still, the current state of knowledge on SGDs in the Baltic Sea is limited
to local studies which have used different approaches. Therefore, it is hard to draw overall conclusions and projections for the
entire Baltic Sea.

Driving forces of SGD involve topography-driven flow, wave set-up, precipitation, sea level rise and convection caused by
salinity and temperature between the seawater and groundwater. As climate change is expected to affect most of the above-
mentioned factors, consequently it can also be expected to affect SGD. These effects would be observed in changes in the
magnitude and composition of SGD, as well as in the biogeochemistry of the subterranean estuary (mixing zone/transition
zone). Climate change is expected to affect ecosystems differently in different regions of the Baltic Sea. Furthermore, it can
be speculated that SGD fluxes may increase in the northern Baltic Sea, where increased precipitation and groundwater tables
are projected (Christensen et al. 2021), while in the southern part, the opposite trend can be envisaged. Sea level rise and
geostatic land movement will certainly also affect SGD, but due to a lack of SGD data, it is hard to project the direction and
significance of the change.

**Impacts of submarine groundwater discharge on other factors**

- There may be an indirect effect on coastal **hypoxia (?)** through the release of additional nutrients via SGDs, but its
  magnitude and relevance are uncertain. Studies have indicated that nutrients transported through SGD can support
  benthic and water column primary production in various coastal ecosystems and inhibit hypoxia (McMahon and
  Santos, 2017; Adolf et al., 2019)
- SGDs may be enriched in dissolved inorganic carbon, pCO$_2$, low pH and alkalinity in comparison to receiving coastal
  waters (Liu et al., 2014; Szymczycha et al., 2014), and thus may alter coastal water properties. Still, there is no
  indication that SGDs in the Baltic Sea region have any effect on **acidification (?)**, except maybe on a very local scale.
  However, studies on the global level reveal that SGD can reduce the CO$_2$ buffering capacity of the receiving ocean,
  and act as a local driver of ocean acidification in local regions (Liu et al., 2014; Liu et al., 2021).





- Impacts on marine **ecosystems (?)** may be locally important. To date, in many coastal regions marine organisms, respectively their biomass, abundance, productivity, physiology, and community structures have been directly evaluated along SGD gradients. SGD can contribute significantly to reef productivity and/or calcification and altering the phytoplankton community structure (McMahon and Santos, 2017; Adolf et al., 2019). In meiofauna assemblages in the Bay of Puck, southern Baltic Sea, a significant decline of certain meiofaunal taxa (mainly nematodes and harpacticoids), as well as altered temporal patterns and a changed small-scale vertical zonation was demonstrated (Kotwicki et al., 2014).

- The impact of SGD on nutrient loads depends on the hydrogeological and biogeochemical conditions of the coastal region. In many coastal sites, SGD-driven **nutrients loads (+)** are significant.Nutrients as well as dissolved organic and inorganic carbon components usually have concentrations several orders of magnitude higher in groundwater than in surface water. Therefore, even if the SGD rate is low, the chemical substances flux via SGD can be relatively important, as groundwater in coastal aquifers tends to be enriched in various chemical substances. In the Bay of Puck, Poland, the estimated seasonal and annual loads of both dissolved inorganic nitrogen and phosphates via SGD were the most significant source of nutrients (Szymczycha et al., 2020a). It can be assumed that SGD will also be a significant source of nutrients in other Baltic Sea regions, and therefore affect benthic and water column primary productivity, and also the phytoplankton community structure.

- Groundwater discharge may contain considerable concentrations of various **chemical contaminant** (substances, as they accumulate in the freshwater bodies and soils and intrude the aquifers where the may reach high concentrations (Hapworth et al., 2012). SGDs may thus be an essential input route for **chemical contaminants (+)** and metals to the coastal Baltic Sea. Local studies in the Bay of Puck (Poland) suggest that SGD may be an important source of methane, dissolved organic and inorganic carbon (Szymczycha et al., 2012; Kotwicki et al., 2014; Donis et al., 2017), nutrients (Szymczycha et al., 2012; Szymczycha et al., 2020a), pharmaceuticals and caffeine residues (Szymczycha et al., 2020b) and trace metals (Szymczycha et al., 2016).

**5.6 Marine Ecosystem**

The marine ecosystem of the Baltic Sea is largely characterized by the physical and biogeochemical conditions of the water body, which in turn are defined by the physical settings. The marine ecosystem features the taxonomic groups which are present in the oceans, ranging from viruses, bacteria, phytoplankton, mixo- and heterotrophic protists (flagellates, ciliates, amobae, nano and microzooplankton) to multicellular heterotrophic organisms like crustacean plankton (cladocerans, copepods), to fish, mammals (grey seals, ringed seals, harbor porpoises), sea birds and invertebrates like ctenophores and jellyfish. Organisms dwelling on or in the sea floor (benthos) are primarily unicellular foraminifera and invertebrate multicellular bivalves, snails, worms and macrophytes (HELCOM, 2012). The Baltic Sea features a salinity continuum between near freshwater in the North East to near ocean values in the Kattegat, resulting in a comparably low species diversity in the Baltic Sea, as compared to other coastal seas (HELCOM, 2018a).

Due to its geographical location, the Baltic Sea ecosystem is subject to a strong seasonality. The seasonal cycle in the pelagic zone shows a strong growth pulse of phytoplankton in spring, as new nutrients have been distributed throughout the upper water column by thermal convection in winter, and enough light is available to trigger photosynthesis. Spring blooms are characterized by larger phytoplankton, usually diatoms. Summer is typically the phase when regenerated production dominates, i.e. when a strong stratification cuts off the nutrient supply from the deeper layers, and when the microbial loop prevails. Very small (<2μm) unicellular cyanobacterial phytoplankton (*Synechococcus* sp., *Prochlorococcus* sp.) dominate primary production during this phase, and the system is fueled by nutrients regenerated within the euphotic zone by active grazing by very small flagellates, ciliates and other microzooplankton. As nitrogen is a scarce commodity during this phase, the ecological niche opens for the nitrogen-fixing filamentous cyanobacteria, which usually bloom in late summer. As the





strong summer stratification is eroded by cooling and strong winds, new nutrients can again be distributed to the euphotic zone, giving rise to autumn blooms of larger phytoplankton, mostly dinoflagellates. In the past decades, a shift of seasonality (earlier occurrences of spring and cyanobacterial summer blooms (Kahru and Elmgren, 2014) and ocillations, Kahru et al., 2018) and a shift towards a predominance of dinoflagellates over diatoms (Klais et al., 2011; Spilling et al., 2018) has been shown and related to milder winters and related ecosystem changes (Wasmund et al., 2017). There are no coccolithophorids in the Baltic Sea, which is an important group of oceanic phytoplankton, forming extensive booms in the adjacent North Sea. Their absence in the Baltic Sea is difficult to explain, but it may be related to the complex carbonate budget of the Baltic Sea (Tyrrell et al., 2008). There are also no single-celled radiolaria in the Baltic Sea, which may be related to the low salinity (Kruglikova, 1989).

The extensive blooms of filamentous nitrogen-fixing cyanobacteria are an outstanding feature of the Baltic Sea pelagic ecosystem, which are absent from most oceanic provinces, but often found in other marginal seas and in freshwater bodies. The dominance of the cyanobacteria can be attributed to the particular biogeochemistry of the Baltic Sea, which follows its unique oxygen conditions (Bianchi et al., 2000). In times of low nitrogen availability, nitrogen fixers have an advantage over other phytoplankton as they can transfer molecular nitrogen to the bioavailable form of ammonium. As they decay, this nitrogen becomes available for the rest of the autotrophic community. Bioavailable nitrogen is removed from anoxic water through denitrification, and phosphorus is released from the deep water and sediments (internal phosphorus loading). The important N:P ratio is thus heavily modified (e.g. Granéli et al., 1990). Hence, oxygen conditions and cyanobacterial blooms substantially affect the Baltic Sea ecosystem (Motwani et al., 2018), by altering the redox state of the system and shifting nutrient ratios.

Although blooms of cyanobacteria are usually favored by warm temperature (Huisman et al., 2018), nutrient reductions following the BSAP are projected to have a stronger suppressing effect on these blooms than warming (Meier et al., 2019a), although the legacy pools of phosphorus are expected to prevail for the next decades (Vahtera et al., 2007). So far, there is no clear evidence for an overall significant increase or decrease in cyanobacterial blooms (Olofsson et al., 2020).

The comparably low diversity and distribution of fish communities in the Baltic Sea are largely determined by the salinity gradient. Marine species (cod, flounder, herring) are more abundant in the open waters and coasts of the southern and central Baltic sea, while freshwater species such as perch and roach are more found in the northern waters (HELCOM, 2006, HELCOM, 2018a). Migratory fish species like salmon, sea trout and eel, which spawn in other waters, are also present in the Baltic Sea. Fish larvae largely live from zooplankton, of which some marine and brackish copepods species as well as the fresh- and brackish water cladocera are dominant in the Baltic Sea. Fish, in turn, are food for predatory fish species, marine mammals, birds and humans. For commercial fish and fisheries, see Section 5.11).

The marine ecosystem of the Baltic Sea is heavily under pressure by various human activities. Non-indigenous species may invade the Baltic Sea naturally, or through human activity (mostly through shipping). The round goby (*Neogobius melanostomus*) is a prominent example of an invasive species which has interfered with the indigenous ecosystem (Nurkse et al., 2016). Other examples are the pelagic comb jelly (*Mnemopsis sp.*) and the benthic annelid worm *Marenzelleria sp.* (see Section 5.7). Chemical substances and marine litter released to the sea by humans pose a largely unknown threat to the Baltic Sea (see Sections 5.15. and 5.17., respectively). Many effects are unknown, or propagate through the food chain to fish, where they accumulate and may threaten human consumption. As this is a human health issue, the impact on humans is much better studied than the effects on other components of the marine ecosystem. The impact of marine noise caused by human activities (installation and operation of coastal infrastructure, shipping) is also a largely unknown factor of disturbance for fish, mammals or smaller components of the ecosystem (Ströber and Thomsen 2019; Mustonen et al., 2019). The effects of extensive wind farms on pelagic ecosystems are largely unknown, ranging from creating microhabitats at the hard substrates of the pillar bases (reef effects , Andersson and Öhman, 2010), marine noise during the construction phase, to effects on stratification and downwelling (van Berkel et al., 2020)



Food production for human consumption has a substantial impact on the marine ecosystem. Nutrient release through agriculture and (to a lesser degree, aquaculture) causes excessive growth of phytoplankton, generating not only excessive oxygen deficiency zones in the deep central basins and many coastal zones, but generally also an elevated level of food availability for upper trophic levels. It has been shown that increasing fish stocks and catches in the 1980s were partly related to eutrophication (Elmgren, 2021).

Marine ecosystems are affected strongly by climate change through warming and changing seasonality (Viitasalo, 2021; Viitasalo et al., 2015). For phyto- and zooplankton, longer and shifted growth periods have been shown, with selective impacts on specific species and functional groups (e.g. dinoflagellates, certain copepods) (Wasmund et al., 2019). Changes in benthic communities have been related to warming and eutrophication (Ehrnsten et al., 2020). A prominent potential victim of the anticipated reductions in sea ice in the northern Baltic Sea is the ringed seal which breeds on sea ice (Meier et al., 2004), while

other marine mammals e.g. harbor porpoises, may profit from warming (Dippner et al., 2008). Still, there are many complex interactions between species and functional groups within the ecosystem and climate change related impacts on certain species or groups may have indirect propagations to other compartments of the system.

**Impacts of marine ecosystems on other factors**


- There is a strong impact by ecosystems on hypoxia (+). This connection is a strong example for the feedback between different factors: hypoxia strongly affects ecosystems (by altering the redox conditions in the deep water and sediments and hence the microbial communities and metabolic processes, i.e. hydrogen sulfide production, by denitrification and phosphorus release and thus altering the pelagic nitrogen to phosphorus ratio), e.g. by inciting

cyanobacterial blooms. Hypoxia in turn is strongly affected by the pelagic ecosystem as produced and sedimented biomass from the surface layers increases oxygen demands and the hyp- and anoxic area.

- There is a feedback of ecosystems on acidification (+) through primary production and respiration processes affecting the $CO_2$ budget in the water column (Gypens et al., 2011; Cai et al., 2011). Also, feedback mechanisms related to alkalinity are known.

- There are various foodweb interactions possible between the autochthonous ecosystem and the intruding non-indigenous species (+). Prominent examples are round goby (Almqvist et al., 2010), *Marenzelleria viridis* (Quintana et al., 2018), and *Mnemiopsis leidyi* (Riisgaard, 2017).

- Exploitable fish stocks are part of the Baltic Sea ecosystem, so all food web constituents have a strong impact on the recruitments through cascading effects from phytoplankton to fisheries (+).

- There may be a certain impact of the ecosystem on tourism (+), through extensive blooms of filamentous cyanobacteria which may be washed onto beaches, although their occurrence in the future may be reduced with successful abatement measures (Meier et al., 2019a). Some of these blooms may be potentially harmful, and in recent summers, beaches have been closed due to the occurrence of these blooms.

**5.7 Non-indigenous species**

Non-indigenous species are introduced by human activity into environments where they previously had been absent. For the Baltic Sea, it is estimated that 140 to 170 new species have established themselves (HELCOM, 2018a; Ojaveer et al., 2017), while their ecological and economic impact varies widely. The main vectors are ship hulls (biofouling), ballast water, and canals connecting previously separated bodies of water (Gollasch et al., 2000; Keller et al., 2011). Ports are known as hot spots for the distribution of non-indigenous species (HELCOM, 2018a). Temperature and salinity may be limiting or favouring

factors for the distribution of non-indigenous species in the Baltic Sea (Holopainen et al., 2016). Prominent non-indigenous species, which have been shown to affect pelagic and benthic communities in the Baltic Sea, and also have economic



implications, are the benthic polychaete worm *Marenzelleria* ssp. (Stigzelius et al., 1997), the pelagic comb jelly *Mnemiopsis leidyi* (Haslob et al., 2007), and the demersal fish round goby, which is shortly described here.

The bottom-dweller round goby *Neogobius melanostomus (Pallas 1814)* was recently recognized as an important intruder with
the capacity to influence local species (Karlson et al., 2007; Järv et al., 2011; Ustups et al., 2016), and a potential to cause a
regime shift in coastal ecosystems (Nurkse et al., 2016). The small fish (up to 25 cm length) is of Ponto-Caspian origin (Miller, 1986) and was first reported in 1990 from the Gulf of Gdansk (Skóra and Stolarski, 1993). Since then, it has spread to other coastal areas of the Baltic Sea. It is believed that ship ballast water is the main vector for a long-distance transport to the Baltic Sea (Kornis e al., 2012; Kotta et al. 2016); however, it has been demonstrated that round goby is capable of migrating along
the coast at a speed of up to 30 km yr$^{-1}$ (Azour et al., 2015).

The round goby is found in different types of bottom habitats, usually at depths up to 30 m (Cross and Rawding, 2009). During summer, the round gobies breed and feed in shallow coastal waters (Kornis et al., 2012). The migration range during this period is mostly restricted to distances of a few hundred meters (Ray and Corkum, 2001). Longer migrations of up to several kilometers to and from deeper waters take place in spring and autumn (Sapota, 2012). The fish is an opportunistic feeder (Skóra
and Rzeznik, 2001; Kornis et al., 2012) and no statistically significant size-specific preference for the pacific mussel *Mytilus trossulus* was found (Nurkse et al., 2016). Other studies have documented that round goby prefers the shrimp *Crangon crangon* over the blue mussel *Mytilus edulis,* and prefers *Mytilus edulis* over herring eggs (Wiegleb et al., 2018), leaving the issue of round goby food preferences somewhat unresolved.

It was shown that round goby represents the most important prey for the medium sized cod *Gadus morhua*, and perch *Perca*
*fluviatilis* almost exclusively feeds on round goby in the Gulf of Gdansk (Almqvist e al., 2010). Similarly, round goby was found to be an important fraction in the diet of perch in the Pomeranian Bay (Oesterwind et al., 2017). At the same time, Järv et al. (2011) documented that perch in its diet prefers other fish species over round goby, suggesting that round goby is rather a prey of opportunity than of choice. Therefore, it can be assumed that in some Baltic Sea areas, round goby has significantly suppressed several species that used to be preferred food items for other predatory fish species in coastal ecosystems.
Furthermore, it was shown that round goby has become an important food source also for turbot *Psetta maxima* (Sapota and Skora, 2005), and birds like the great cormorant *Phalacrocorax carbo* (Bzoma, 1998; Rakauskas et al., 2013) and grey heron *Ardea cinereal* (Jakubas, 2004; Rakauskas et al., 2013). A dietary overlap between round goby and flounder (Karlson et al., 2007; Järv et al., 2011) and juvenile turbot presumably resulted in lower abundances in these species (Ustups et al., 2016).

It is presently not possible to assess detailed impacts on ecosystems as they seem to be very specific to the ecosystem invaded
(Hirsch et al., 2016). Intraspecific interactions between invasive species could potentially mediate their ecological effects (Kornis et al., 2014). There have been efforts to introduce the round goby to the consumer market ("use and reduce"). While it is well suited for the market in terms of meat quality, the main obstacle seems to be its small size (Brauer et al., 2020).

**Impacts of non-indigenous species on other factors**


- Non-indigenous species may have strong impacts on **marine ecosystems (+)**, e.g., round goby alters coastal habitats, decreasing their value as marine protection areas (Skabeikis et al., 2019). Examples are Round Goby *Neogobius melanostomus* (Almqvist et al., 2010), *Marenzelleria* ssp. (Quintana et al., 2018), and *Mnemiopsis leidyi* (Riisgaard, 2017).

- Non-indigenous fish can either positively or negatively affect food availability and growth of commercial fish (Ojaveer and Kotta, 2015), with impacts on **fisheries (+)** opportunities. Also, non-indigenous species can become a target for fisheries, e.g. round goby (Ojaveer et al., 2015). Also, it has been demonstrated that round goby competes for food with juvenile flatfish (Ustups et al., 2016).



- Regulations to mitigate biofouling and ballast water vectors have economic repercussions on the **shipping (+)**
industries. A convention for the Control and Management of Ships' Ballast Water and Sediments by IMO is in force
as of 2017 (IMO, 2021a) and requires the use of ballast water management systems.

- It has been shown that non-indigenous polychaete *Marenzelleria neglecta* burrows deeper in the sediment than native
species and can enhance bioturbation-mediated transport of **chemical contaminants (+)** to the overlying water
(Granberg et al., 2008; Hanson et al., 2020).

- There may be an impact on **coastal management (?)** if a non-ingenious species becomes a problematic species
causing problems for other species or having a detrimental effect on the ecosystem or driving a regime shift through
a novel predator (Kotta et al., 2018). Eventually, management actions need to be taken to minimize the impacts. There
are considerable impacts by non-indigenous species on biodiversity and ecosystem structure (Lehtiniemi et al., 2015).
However, a complete removal of the new species is impossible once it has established itself (HELCOM, 2018a).

**5.8 Land Use and Land Cover**

Anthropogenic land use in the Baltic Sea region started at least about 6000 years before present (Gaillard et al., 2015; Smith
et al., 2008). Deforestation for firewood, iron mining and agriculture has been the main factor driving land cover changes since
at least 2000 years b.p. (Roberts et al., 2018; Lavento, 2019). Simulations by Strandberg et al. (2014) indicate that, during its
maximal extent around 200 years ago, the deforestation of the southern and eastern part of the Baltic Sea catchment may have
had an impact on the regional climate, comparable to present day greenhouse gas emissions driven climate change.

Land cover as a driver of environmental changes is well recognized when addressing the global and regional challenges related
to the mitigation of anthropogenic $CO_2$ fluxes. It is widely discussed as an important part of the Earth´s carbon cycle, both as
the second largest source of anthropogenic $CO_2$ emissions due to the ongoing large-scale deforestation of tropical areas (IPCC,
2013), and for its potential to mitigate the effects of anthropogenic $CO_2$ emissions through increased carbon uptake by
reforestation (Sonntag et al., 2016; Law et al., 2018). However, many land cover driven environmental changes and the possible
feedbacks from those are not clear and under debate (Gaillard et al., 2015).

During the last decades, the ongoing deforestation of the tropical and subtropical regions is accompanied by accelerated
reforestation of northern mid to high latitudes (IPCC, 2013). The projections of future land use and cover in general anticipate
a global increase in cropland and a reduction in the pasture and forest extent, but show considerable differences in the
predictions of land use and cover development at continental/sub-continental scales, and incorporate large uncertainties
(Prestele et al., 2016). The political efforts to mitigate climate change are aimed at decreasing the deforestation rates in tropical
regions, and further increasing the reforestation pace at mid and high latitudes (UNFCCC, 2014; EU renewable energy
Directive 2018/2001/EU), in order to decrease the land cover related $CO_2$ emissions and to increase the amount of carbon
sequestered by terrestrial land cover.

Since the introduction of agriculture millennia ago, anthropogenic deforestation was, at the continental scale, a major human
impact on land cover (e.g. Ellis, 2011; Roberts et al., 2018). Next to the biogeochemical feedbacks, i.e. changes in the capacity
of the $CO_2$ sequestration through photosynthetic fixation into biomass, there are also considerable biogeophysical feedbacks
on climate through the change in the reflectivity of the land surface (albedo). Dark surfaces (e.g. forests, waters) absorb the
incoming heat better than bright surfaces (deserts, agricultural lands). The type of land cover has thus an impact on the regional
climate (Bala et al., 2007; Deng et al., 2013). Most importantly, both effects may counteract each other, e.g. reforestation may
contribute to a drawdown of $CO_2$ from the atmosphere, thereby theoretically contributing to a cooling effect; but that additional
forest area may increase the dark surface and lead to a weaker albedo, contributing so to warming. These trade-off effects are
difficult to quantify (Mykleby et al., 2017), but it has been assumed that reforestation as a measure for carbon drawdown and
cooling, at least in the already mostly forested northern Baltic Sea region, will probably be of little effect (Arora and
Montenegro, 2011). However, recent modeling studies imply that a massive reforestation may lead to a significant lowering





of summer maximum temperatures and a reduction of summer heat waves south of the Baltic Sea (Strandberg and Kjellström, 2019).

There are major uncertainties related to projections of the speed and direction of terrestrial land cover change and its ability to provide the service as a reducer of atmospheric $CO_2$ concentrations. Projected land use and land cover change will have

implications for the functioning and structure of terrestrial ecosystems, and on the amount and nature of the ecosystem services supported by the land cover, regardless if we consider more deforestation or reforestation. As terrestrial land cover is a slowly changing system with long-term implications, it is crucial to investigate both short and long-term effects.

The role of land use and land cover change as a driver of terrestrial organic matter transport into aquatic systems is not well understood (Kayler et al., 2019). While the importance of terrestrial vegetation around drinking water resources is well

recognized at the local scale, the impacts on the aquatic environments at a regional to global scale are much less known and studied. Cross-system studies with a focus on matter transfer between terrestrial and aquatic environments are rare, but when conducted, they show considerable land use driven changes in the composition and amount of terrestrial origin dissolved organic matter transported into the aquatic systems (Ning et al., 2018; Bragée et al., 2013). Dissolved organic carbon (DOC) from the land (Humborg et al., 2015) is one of the major sources of nutrients in terrestrial surface water systems, and can have

considerable impacts on coastal marine environments (Ning et al., 2018).

**Impacts of land use and cover on other factors**

- There is evidence that land use and cover may have an impact on the regional **climate (+)**, through biogeochemical
and biogeophysical effects (Strandberg et al., 2014; Mahmood et al., 2010). Albedo, i.e. the reflectance of the land surface affects the amount of energy reflected back into space and the fraction, which is converted to warming. Bright surfaces like agricultural lands reflect more energy than dark surfaces as forests and waters (biogeophysical effect). Thus, the fraction of land cover may affect regional warming (Strandberg et al., 2014, Strandberg and Kjellström, 2019). There is also evidence that an increase in $CO_2$ concentrations leads to an increase in vegetation
(biogeochemical effect), at least in regions where water is not a limiting factor, i.e. the northern part of the Baltic Sea basin (Smith et al., 2008). It is, however, not clear what the respective impacts of these effects are and whether reforestation as a measure to mitigate climate change can be successful (Gaillard et al., 2015).

- There is an indirect but clear relation between land use and **hypoxia (+)** through nutrient release from agricultural fields and associated eutrophication (Altieri and Diaz, 2019).

- There is a well-documented connection between land use and soil erosion/**acidification (+)**, both can be accelerated or slowed down by the choice of agricultural practices, fertilizers and crops (Bolan et al., 2005; Xiong et al., 2019). Furthermore, these processes affect matter exchange between aquatic and terrestrial systems and can lead to accelerated deterioration (eutrophication and acidification) of water quality (Hornung et al., 1990).

- There can be a considerable impact of land use on **Submarine Groundwater Discharge (+)**. It can be expected that
the type of land use and associated soils affect the quantity and quality of water seeping to groundwater and eventually reaching coastal discharge spots. However, the extent of this relationship is unknown (Rufí-Salís Martí et al., 2019).

- Land use (other than agriculture) may have an impact on effluents (nutrients, contaminants) from soils and land surface, potentially affecting coastal or **marine ecosystems (?)**, but this has not been assessed (Langlois et al., 2011).

- While forestry predominates in the northern part of the Baltic Sea basin, **agriculture (+)** is the dominant type of land
use in the southern part of the basin, and it will be severely affected by a projected decrease in precipitation in the south. The decisions on which part of the land is dedicated to agriculture (or any other type of land use) are very much management and political decisions, which are influenced by climatic conditions (Wiréhn, 2018; Mendelsohn and



Dinar, 2009). There is a strong interrelation between the type of land use and **nutrient loads (+)**, as it strongly affects the amount of nutrients leaking to the sea, predominantly from agricultural land (Dambeniece-Migliniece et al., 2018).

- Land use indirectly affects certain branches of **fisheries (+)**, e.g. by affecting rivers where salmons and other migrating fish spawn (Drouineau et al., 2018), see also Section 5.11 and 5.12.

- Land use change is a major force driving **river regulations (+).** Regulation of river basins and drainage works in agriculture and forestry have been major factors for changes of hydrological and water quality responses in watersheds (Wörman et al., 2010). Moreover, damming of rivers has increased the area of freshwater bodies in the Baltic Sea

region (Smedberg et al., 2009; Humborg et al., 2015).

- There may be an indirect connection between land use and **offshore wind farms (?)** as there may be a competition for space between land-based wind farms and other types of land use. If the spaces on land become rare because of regulations and protests against extensive land use for wind farms, political decisions may be taken to build more at sea (Ladenburg, 2008).

- Fertilizers and insecticides on cultivated land may leak to the soil and sea, thus, it can be expected that change of land use has a considerable impact on emissions of **contaminants (+)** to the coastal sea. In addition, remobilization of toxic mercury from the soil (where it has been accumulated for decades) and transport to rivers and into the sea has been shown (Saniewska et al., 2014; Saniewska et al., 2019; Gębka et al., 2020).

- Land use is expected to have a certain impact on **tourism (+)** because coastal resort areas, beach developments and

golf courses, among others, have high demands to areas and infrastructure which are in competition with other types of land use (Kropinova, 2012; Cottrell and Raadik Cottrell, 2015).

- There is a clear connection between coastal land use and **coastal management (+)**. Housing areas close to the coast, on sand spits or on cliffs that are affected by coastal erosion and sea level rise are in peril of being lost to the sea. While it is customary to protect endangered segments of the coastline against erosion also in the parts of the Baltic

Sea where the coasts are sinking (Pruszak and Zawadzka, 2005), the concept of managed retreat (Hino et al., 2017) is gradually being included into planning measures and several decisions have been made to abandon sections of shoreline (Schernewski et al., 2018). The same holds for agricultural lands and forests (Rekolainen, 1997; Nordström et al., 2015; Gopalakrishnan et al., 2019) close to affected shorelines even though the value of the unit of such areas is less than that of urbanized regions. Furthermore, land use affects terrestrial biodiversity, by degrading natural

biotopes, reducing habitat and population sizes (Hansen et al., 2012), and hindering migration through fragmented landscapes (Oliver and Morecroft, 2014; Smith et al., 2008). Changed land cover e.g. the replacement of permeable soils with sealed urban spaces may lead to an increased vulnerability for flooding and inundation, with consequences for local economies, of which there is usually a high concentration in urban areas (Saniewska et al., 2014; Saniewska et al., 2018).


*Human-induced factors*

The following sections describe the factors and impacts of direct anthropogenic origin, i.e. these factors would not exist in the absence of humans. Often, coastal management is an integrating activity managing the factors and impacts described below. These focus on human benefits and may be in conflict with benefits for ecosystems.

**5.9 Agriculture and nutrient loads**

Agriculture is a strong driver of earth system changes. Agriculture accounts for 40% of global land area, 30% of greenhouse gas emissions, 70% of water withdrawals, and has doubled the amounts of nitrogen and phosphorus in circulation (Foley et al., 2011). In the Baltic Sea region, about 20% of the total catchment area is agriculture, varying from about 7% of the area for Sweden and Finland to 60% for Denmark (Svanbäck et al., 2019). In the past several decades, fertilizer use has decreased,



while yield has increased due to improvements in crop varieties and agronomic practices (Lassaletta et al., 2014). There is a strong, positive linear correlation between agricultural nutrient surpluses and nutrient loads to the sea (surpluses are calculated as the sum of nutrients in fertilizer, manure, N-fixation by crops (N only), and atmospheric deposition (N only) minus removal due to crop harvest) (Hong et al., 2017).

Diffuse losses of nutrients from agriculture are a core driver of nutrient loads to the Baltic Sea (Andersen et al., 2015). For the
drainage basin as a whole, about 14% of net anthropogenic nitrogen inputs and 4% of net anthropogenic phosphorus inputs are transferred to the sea annually (Hong et al. 2017). There is often an inverse correlation between nutrient use efficiency and agricultural nutrient surpluses (e.g., low use efficiency often results in high surpluses). Use efficiency is calculated as nutrients removed in crop harvest divided by the sum of manure and fertilizer added. Nitrogen and phosphorus use efficiency in crop production is about 55% for both but varies greatly by country. For example, phosphorus use efficiency is <40% in Russia and
Belarus but > 90% in Germany, Denmark, Estonia, Latvia, Lithuania, and Sweden (McCrackin et al., 2018).

Livestock is a driver of nutrient cycling in the drainage basin. About two thirds of nutrients in crops grown in the region are fed to livestock animals (not humans). In addition, substantial amounts of nutrients for livestock are imported in the form of soy. There is a positive relationship between the density of livestock and nutrient surpluses. Nutrients in manure are not always used efficiently in crop production because of increased specialization and separation of crop and livestock production in the
landscape. It is often more economical for farmers to purchase commercial fertilizers than to use nutrients in manure for crops (Wang et al., 2018; Svanbäck et al., 2019). Model studies suggest that redistributing manure nutrients, together with improving agronomic practices, could meet 54–82% of the remaining nitrogen reductions targets (28–43 kt N reduction) and 38–64% of phosphorus reduction targets (4–6.6 kt P) under the Baltic Sea Action Plan (McCrackin et al., 2018b).

It is not known how fertilization practices, crops grown, and land use will change in response to climate change. However, it
appears plausible that changes in temperature and precipitation patterns could change the types of crops grown in the region, with potential changes in fertilizer practices and diffuse nutrient losses, as well as riverine runoff and the magnitude of nutrient loads attributable to agriculture.

Globally, nutrient loads to coastal areas are increasing despite increased retention along the land-to-sea pathway (Beusen et al., 2016). Trends in the Baltic Sea region are much the opposite of global trends. Since external nutrient loads peaked around
1980, total waterborne and airborne N and P loads decreased by 42% and 56%, respectively (Savchuk et al., 2012). Current nutrient reduction plans (e.g. under the Baltic Sea Action Plan) do not consider climate change. Therefore, additional reductions in land-based nutrient loads could be needed to reach environmental goals for the sea.

Sources of nutrients transported to the sea by rivers vary by country and nutrient type (e.g. nitrogen or phosphorus, HELCOM, 2018b, c). In Finland, for example, agriculture is the largest source of phosphorus and natural background sources are the
largest source of nitrogen. For Poland, agriculture is the largest source of nitrogen while point sources are the largest source of phosphorus.

Regardless of near-term nutrient reductions, symptoms of eutrophication will eventually be present in the Baltic Sea for the next several decades, due to slow response times (Savchuk, 2018). However, recent evidence suggests that conditions in the sea are improving (Andersen et al., 2017) even though the sea as whole was recently assessed as not achieving environmental
targets (HELCOM, 2017b). Thus, there will continue to be a loss of social welfare as long as the sea remains eutrophied (Ahtiainen et al., 2014). The extent to which the countries around the Baltic Sea can meet the nutrient load reductions specific under the Baltic Sea Action plan is not known.

Next to the external loads through atmospheric, river and diffuse sources, internal loads of phosphorus play a particular role in the Baltic Sea. Large amounts of phosphorus are stored in the sediment and deep waters, which are released in anoxic
conditions (Puttonen et al., 2016). This phosphorus surplus in the large basins in combination with depleted available nitrogen at the surface gives nitrogen-fixing cyanobacteria a competitive advantage over other phytoplankton, resulting in extensive cyanobacterial booms in late summer, still exacerbating the oxygen situation in the deep basins (Reed et al., 2011). Nitrogen





from rivers does not effectively reach the central Baltic proper due to coastal denitrification and turnover. The cyanobacetrial blooms in this area, however, are fueled by additional phosphorus from land sources, and internal loading (Voss et al., 2005b).

The effects of climate change on nutrient loads are highly uncertain. Nutrient loads depend on runoff and riverine discharge. Climate models suggest that the north of the Baltic Sea region would be wetter and the south would be unchanged or drier (Christensen et al., 2021; Meier et al., 2021a). The agricultural south may experience a reduced runoff, hence reduced nutrient loads to the sea. Currently, there is no consistent catchment-wide model of nutrient source apportionment (HELCOM, 2018c), so it is difficult to assess where which sources predominate in different sea sub-basins. It is unknown how fertilization practices,

crops grown, and land use will change in response to climate change. Also unknown is the relative contribution of nitrogen from accumulated, legacy sources to current riverine loads to the sea, and how the accumulation and release of "legacy" nutrients will be impacted by climate change. Nutrient loads are strongly influenced by runoff and discharge, so riverine nutrient loads strongly influence the eutrophication status of the sea.

Global population growth will likely increase demand for food products. Increased wealth in countries like China will likely

increase demand for livestock products (Yu et al., 2016). It will be challenging to reduce nutrient loads from agriculture and increase food production if farm structures and practices changes remain unchanged.

**Impacts of agriculture on other factors**

- A possible feedback by agriculture, or land use in general, on the regional **climate (+)** may be through albedo. Agricultural areas have a higher albedo than forests and waters, so increased agricultural areas may be a cooling factor but the extent is unknown (Gaillard et al., 2015).

- Agricultural practices, i.e. fertilization of fields, are the primary source for nutrient loads to the Baltic Sea, so there is a strong impact of nutrient loads on **hypoxia (+)**. Increased nutrient (mostly phosphorus) loads to the large basins

since the 1950s are responsible for expanding oxygen-poor or -free zones in the past decades. There is clear and unequivocal evidence for this connection (e.g. Gustafsson et al., 2012).

- Agriculture affects the carbon chemistry of the coastal sea due to carbon and nutrient loads and thus may have an impact on **acidification (+)** through nitrogen fertilizers and agricultural procedures (Peters et al. 2011), the stimulation of primary production, stimulating the drawdown of $CO_2$ in the water column and thereby affecting acidification.

- The amounts and types of dissolved substances in the groundwater are strongly determined by agriculture, which thus strongly affects the quantity and quality of **submarine groundwater discharges (+)** (Szymczycha et al., 2020a; Szymczycha et al., 2020b)

- The impacts of agriculture on **marine ecosystems (+)** are clearly the release of excess nutrients from agricultural fields. These excess nutrients are washed into the aquifers and rivers and a large fraction ends up in the sea, where it

leads to eutrophication, phytoplankton blooms, and in the Baltic Sea to increased oxygen deficiency zones with extensive further repercussions for the ecosystems, e.g. extensive cyanobacterial blooms.

- Agriculture is the dominant type of **land use (+)** in the southern Baltic Sea region.

- There is a strong effect on **fisheries (+)** and **aquaculture (+)** as nutrients are the basis of the aquatic food web affecting fish production through multiple trophic processes. Generally, more nutrients mean more fish production. However,

a too high nutrient availability contributes to eutrophication and oxygen deficiency, with negative impacts on fish growth and reproduction, e.g. for cod (Köster et al., 2017; Casini et al., 2016a). Furthermore, the composition of prey species, which may be affected by nutrients, is important for the production of specific fish species (e.g. Möllmann et al., 2005; Neuenfeldt et al., 2020).


- Agriculture is a major driver for **river regulations (+).** A multitude of drainage works in agricultural land has
gradually led to a more rapid hydrologic response and may affect the transport of nutrient from land to sea, including
Si retention by damming (Humborg et al., 2000).

- Like for land use, agriculture does affect the amounts and distribution of **chemical contaminants (+)** through
agricultural practices, using fertilizers, veterinary pharmaceuticals and pesticides. Nutrient loads can affect chemical
contaminants and heavy metals indirectly via eutrophication and organic carbon content/dynamics in the sea.
Increased organic carbon in the surface water affects the air-sea exchange of airborne contaminants and transport via
settling particles (Dachs et al., 2002). Increased organic carbon content and lower oxygen levels may promote the
methylation of mercury present in the sediments, enhancing its toxicity and bioavailability (Avramescu et al., 2011;
Bełdowski et al., 2015).

- There may be a connection between agriculture and **coastal management (?)** in regions where coastal infrastructure
and agricultural fields compete for space. Coastal management may need to respond where agricultural fields are at
stake where they are close to cliffs and other coastal features that are subject to erosion; however, these threats are
currently considered minor compared to the potential loss of infrastructure. Nutrient loads stimulate actions to
mitigate the effects of eutrophication (e.g. HELCOM, 2007), affecting coastal management actions (or management
actions in general). On the short-term scale, such actions include warnings about cyanobacteria blooms and increased
concentrations of other adverse substances in the bathing water (Kowalewska et al., 2014). On the long-term scale,
actions may be necessary to manage the growth of reed or other plants that may decrease the recreational value of the
beaches but at the same time reduce damages caused by coastal erosion and flooding (Osorio-Cano et al., 2019), and
also act as a coastal filter for nutrient fluxes from the adjacent land (Kochi et al., 2020)

### 5.10 Aquaculture

Aquaculture, in its broadest sense, includes all cultured breeding and commercial production of plants and animals in water,
ranging from fish, bivalves, macroalgae and microbes to wetland grass. Aquaculture can be roughly divided into *fed* and
*extractive* aquaculture: the former depending on external resources, the latter extracting resources from its surroundings.
Aquaculture in open waters (in contrast to closed recirculating systems) has an impact on its direct environment, and is in turn
affected by environmental conditions.

In the Baltic Sea region, Denmark and Finland are the major producers in marine aquaculture. There is also well-developed
freshwater aquaculture, primarily in Poland and Denmark. Rainbow trout (*Oncorhynchus mykiss*) is by far the most produced
fish in the region, with a share of roughly 70% of the region's total production, followed by common carp with 20%. Other
cultured fish species are European eel, sturgeon, pikeperch, pike, tench, and others. Shellfish (i.e. bivalves like mussels) as
another important aquaculture product is primarily grown in Germany, Sweden and Denmark (Paisely et al., 2010; Eurofish,
2015).

Rainbow trout cage farming in brackish waters is concentrated at Åland, Åbo archipelago, the southwestern Finnish coast and
in the Danish straits, while freshwater cage farming is mainly located in the sheltered waters of the large and exploited rivers
of northern Sweden. Land-based farms, which include traditional flow-through systems as well as the more modern closed
recirculation systems are found throughout the catchment area. Open fish farming is mainly associated with nutrient losses to
the environment and has been shown to cause local eutrophication (Talbot and Hole, 1994; Diaz, 2010).

Increasing water temperatures as well as sea level rise, changed precipitation patterns and extreme weather events have been
identified as the main climate impacts on open fish aquaculture (De Silva and Soto, 2009; Galappaththi et al., 2020). In the
Bothnian Bay, farmers already now see longer periods of suboptimal to lethal temperatures in the upper water layers
(Kankainen et al., 2020). This might promote a growing interest in warm water tolerant species such as perch and pikeperch,



but also in closed land-based systems as procedures and technology constantly evolves and becomes economically competitive. In general, adaptation strategies are necessary for a sustainable aquaculture in the future (e.g. Reid et al., 2019)

A poorly understood climate-related risk is the changed microbial biota in fish guts as water temperatures rise (Huyben et al., 2018), and the risk of infections and diseases (Reid et al., 2019) and parasites (Unger and Palm, 2017). Changes in salinity might also affect the biota, with possible physiological consequences for the fish. Still, none of the present or future candidates

for aquaculture is expected to be affected by a decrease in salinity.

Another problem related to open cage farming and increased water temperatures is the risk of escaping fish (e.g. Atalah and Sanchez-Jerez, 2020) For instance, escaping rainbow trout (accidentally or intentionally released for sport fishing) is a non-indigenious species that may establish permanent wild populations (Skilbrei and Wennevik, 2006; Stanković at al., 2015). This is now addressed by growing sterile fish.

For some cultivation types, a location within offshore wind farms has been envisaged. The closure of certain wind farms to traffic and fishing, together with a location in open, well-ventilated sea areas seems attractive for certain types of aquaculture production (e.g. Mikkola et al., 2018).

Extractive aquaculture includes filtering organism such as bivalves, but also macroalgae and other plants like seaweed (Critchley et al., 2019; Weinberger et al., 2020) and reed (Karstens et al., 2019). Similar to seaweed farming, mussel farming

has the potential to reduce the environmental impact of marine aquaculture by acting as a nutrient sink, transferring nutrients into harvestable biomass ("mussel mitigation farming", Petersen et al., 2014; Holbach et al., 2020). Germany and Denmark are the leading producers of blue mussel (Eurofish, 2015). Macroalgae farming has been tested, also in conjunction with fish (Wang et al., 2014). However, very few suitable species tolerate the low salinity of the Baltic Sea, a problem that may be exacerbated by potentially lower salinities in the future. Blue mussels have been shown to be particularly sensitive to changes

in either temperature and/or salinity (e.g. Westerbom et al., 2002; Braby and Somero, 2006).

In the Baltic Sea region, there is also a strong interest in wetland farming and reed harvest (Karstens et al., 2019). While the blue mussel is a candidate for a "blue" solution for pelagic nutrient abatement on the local scale, reed (*Phragmites australis*), represents a "blue-green" catch crop for nutrient runoff and habitat enhancement in the inner littoral zone. Management of this zone, i.e. the targeted cultivation and harvest of reed, may be an option for nutrient removal. Nutrients leaking from land seem

to pause here before moving on to the pelagic zone and the sediments in deeper water.

Reed has a long history as backup fodder for livestock, but also as roofing material around the Baltic Sea (Köbbing et al., 2013). Since the mid 20th century, eutrophication has resulted in a massive increase in wild reed growth and harvest both at the banks of the Baltic Sea and at lakes in its catchment area, a phenomenon yet to be decribed in the literature but which has been observed in other coastal regions of the world (Xu et al., 2021). This development is likely to continue due to a longer

growing season and reduced ice cover in combination with increased or prevailing high nutrient runoff from agricultural lands. Reduced ice coverage is likely to contribute to a thickening of the reed meadows, both above and below the water surface, as ice removes old reed. Harvest of reeds is an ongoing process all around the Baltic Sea, with an interest to "clean" the water surface in order to improve outdoor activities like boating, swimming and wild life experience like bird watching, fishing and duck hunting.


**Impacts of aquaculture on other factors**

- Aquaculture may be important for local **hypoxia (+)** in enclosed or semi-enclosed coastal regions with little water exchange, where fish cages release excess nutrients and alleviate local eutrophication and possibly hypoxia (e.g. Talbot
and Hole, 1994).
- Aquaculture may have an impact on **marine ecosystems (+)** through local eutrophication near open cage farms, or by escaping cultured specimens from cages into the wild. Indirectly, it could act to reduce fishing pressure on certain





species if there is an aquaculture alternative and the natural populations are less exploited. Mussel farms could enhance wild declining populations by releasing a spate of precisely the same genetic heritage as the wild populations. It has recently been observed that lost mussels from the farm can establish wild colonies under the farm. Also, mussel farms could be used to clean up the seawater at a large scale (Kotta et al., 2020).

- **Non-indigenous species (+)** which are grown in open water cages may escape and establish stable wind populations in the new environment. In 2016, a cargo ship crashed into a Danish fish farm, releasing some 80.000 rainbow trout specimen from the cages (Reuters, 2016). It is unsure whether escaped rainbow trouts are able to establish stable wild populations, but this will be more probable with increasing warming. Continuous escapes during routine operation may also be a problem. Efforts to mitigate this problem are to culture indigenous species preferentially or to grow sterile non-indigenous fish. Another prominent example is the escape of the pacific oyster from oyster farms, following the successful establishment of wild populations in the German Wadden Sea, with subsequent strong competition and suppession of blue mussel beds (Reise et al., 2017).

- An increase in land-based aquaculture is anticipated as technology improves and becomes economic. If so, this will require a complex energy intensive industrial infrastructure (e.g. around 7.5 GWh per year for 1.000 ton production. If renewable energy (e.g. solar power) is used for a carbon neutral production process, ca. 3.5 ha of sun cells per 1.000 tons yearly production must be dedicated. This means additional **land use (+)** beyond the ceiling of the fish factory itself.

- **Nutrient release (+)** to the open water is important on the local scale near fish farms through excess fodder and fish excrements (e.g. Talbot and Hole, 1994). On the other hand, blue mussel, seaweed and macroalgae farming may act to remove nutrients from the vicinity of the farms (Kochi et al., 2020; Kotta et al., 2020) and may help to counteract eutrophication by catching plankton and other small particles and improve water transparency, at least on the local scale (Petersen et al., 2014). By this, mussel farming may deal with increased nutrient release from sediments due to hypoxia and changed nutrient run of from land (Kotta et al., 2020). Fish farming in the Baltic Sea presently represents only 0.5% of **nutrient** losses to the Baltic Sea, but could be the use of recapture based feed sources, i.e. interacting with fisheries, be a net remover of nutrients from the Baltic Sea. Aquaculture is the extension of land-based **agriculture** and farming into the waters. As such, it complements and extends the benefits and detriments of food production to the waters. However, new technologies for particle collection from open cages, land-based farming, and circular feeds, obtaining the nutrient from the recipient load using blue catchcrops and ecosystem management fishing products, can make aquaculture a net contributor to reduced nutrient pressure. Reet management and harvest for fodder production and other uses on sheltered coastal stretches may mitigate local nutrient runoff from nearby agricultural fields.

- Aquaculture may complement **fisheries (?)** and ease the fishing pressure on certain species by growing them in controlled conditions, or provide cultured alternatives to wild catches, and thus reduce the imact of the fishing pressure on the ecosystem.

- There are no direct impacts of aquaculture on **offshore windfarms (+)**. However, wind farms represent suitable locations for certain aquaculture types, as they are installed in open water areas which ensure constant ventilation and exchange of water. These areas are banned from shipping and fishing and have a cerain infrastructure, so aquaculture in wind farms may be synergistic (Buck et al., 2017; Mikkola et al., 2018).

- Some cultivated fish species are loaded with anthropogenic **chemical contaminants (?)** (e.g. antibiotics, pesticides, persistent organic pollutants) to a higher degree than wild populations (Cole et al., 2009).

- Aquaculture may be a source for **marine litter (?)** and microplastics (as the bulk of handling material and equipment is from plastics), but the scale and relevance are largely unknown (Lusher et al., 2017; Sandra et al., 2019).

- There may be an impact of aquaculture on **touristic (+)** activities, for instance by culinary tourism (Kim et al., 2017).

Cerain types of aquaculture farming (e.g. waters reed, macroalgae, blue mussel) may improve the water quality and





light penetration in inner coastal regions and may thus be beneficial for certain touristic activities in these protected inner coastal areas (boating, swimming, fishing, duck hunting.

- **Coastal management (+)** strongly affects coastal aquaculture by allocating areas for farms; conversely, the aquaculture industry is a strong coastal stakeholder and thus has a certain influence on the management of the coastal waters (e.g. Primavera, 2006)

### 5.11 Fisheries

Fisheries affect the Baltic Sea ecosystems primarily through selective extraction of species and physical disturbance to the seabed. The latter is most relevant in the southern Baltic Sea, where gears that come into contact with the seabed (e.g. bottom trawls) are commonly used (ICES, 2018a). Furthermore, some gears, especially gill nets have incidental bycatches of marine mammals and seabirds, affecting these populations (HELCOM, 2017a).

The main target species in commercial fisheries in the Baltic Sea are cod, herring and sprat. Other target fish species include salmon, plaice, flounder, dab, brill, turbot, pike-perch, pike, perch, vendace, whitefish, eel and sea-trout. Fisheries removals are generally recorded in the form of fisheries statistics. Additionally, biological information on the catch (size/age structure, weight of the fish) is collected for the commercially important fish stocks. For several key fish species in the Baltic Sea, long time series of monitoring data and fisheries statistics are available, which have allowed describing the development of fishing pressure over multiple decades, in some cases since the beginning of the intensified fishing in the 1950s-1960s (Eero et al., 2008; Eero, 2012).

According to EU Common Fisheries Policy (CFP), fishing should be conducted in an environmentally, economically and socially sustainable way, and catch limits should be set at levels that ensure maximum sustainable long-term yields. For the major fish stocks in the Baltic Sea, a multiannual EU management plan (EU, 2016) aims to contribute to the achievement of the objectives of the CFP. Fisheries for the major Baltic Sea fish stocks are expected to further align with the targets of these policies in the future.

Fish is a valuable food, and fisheries, together with related industries, are locally an important source of income and employment in the Baltic Sea region, especially in small coastal communities. Therefore, sustainable use of marine resources should be ensured, as also stated in the global policy goals (United Nations, 1982). The goal for an ecosystem-based approach for fisheries management is to consider conservation, economic profit and social equity simultaneously. There are tradeoffs between reaching these goals, and there are competing interests between different users of major fish stocks, which are related to food web interactions between species. For example, management strategies prioritizing overall profit would favour economically more valuable species, such as cod, which may cause conservation and equity issues concerning fisheries targeting other species (Voss et al., 2014).

Both natural and human-induced processes, including species interactions influence the status of the fish stocks. Applying the most appropriate management actions under given ecosystem conditions, i.e. a holistic ecosystem-based approach to fisheries management, requires knowledge of pressure-state links in the ecosystem. One of the major scientific challenges concerning fisheries impacts, is the quantification of fishing impacts relative to other human or ecosystem factors. An example here is cod, where fishing for its prey species potentially influences cod growth and condition (ICES, 2018b). However, as several other factors influence cod growth and condition at the same time (e.g., oxygen conditions, parasites from grey seals) (Casini et al., 2016b; Horbowy et al., 2016), the possible effects of fisheries management actions are difficult to determine. Another example is fishing impacts on the seabed, for which little is known about the sensitivity of different organisms and communities to fishing gear disturbances, at the Baltic Sea scale. In this area, further research and data are needed to parameterize models and establish quantitative links to other pressures (e.g. anoxia) (ICES, 2018a).

Fish stocks and fisheries are also affected by climate (salinity, temperature) and eutrophication, whose effects are closely connected through the oxygen conditions in the Baltic Sea. For example, recruitment of the Eastern Baltic cod is largely





influenced by salinity and oxygen conditions (Köster et al., 2017), and temperature significantly affects sprat recruitment (MacKenzie and Köster, 2004 MacKenzie et al. ,2007). A combination of oxygen content and temperature was found to have

significant effects on egg/larva development and survival of the Western Baltic cod (Hüssy, 2011). The growth of planktivorous species or life stages is also affected by climatic conditions regulating zooplankton dynamics (Casini et al., 2011; Köster et al., 2017). Furthermore, oxygen is considered to affect the growth and condition of the Eastern Baltic cod both directly and via regulating the availability of benthic food (Casini et al., 2016a). Climate impacts on one species can also propagate through the food web and affect other species via food web interactions. For example, a high abundance of sprat

due to favourable temperatures increases competition between sprat and herring and reduces their growth and condition (Casini et al., 2011). Möllmann (2019) provides an overview of the effects of climate change and fisheries on the fish stocks of the Baltic Sea.

A combination of several factors is often responsible for extensive changes in fish abundances. For example, a combination of high fishing pressure and unfavourable salinity and oxygen conditions for cod reduced the cod stock in the late 1980s, which

released sprat from predation pressure and allowed for an increase in sprat stock in the 1990s, which was additionally favoured by suitable temperatures (Köster et al., 2003; Möllmann et al., 2008). In another example, the increase in the cod stock in the late 1970s to the highest level on record was found to be due to a combination of favourable climate and a temporary reduction in fishing pressure (Eero et al., 2011). These examples demonstrate how combinations of different forcings can interact and affect population dynamics.

Eutrophication has presently negative effects on fish resources via deteriorated oxygen conditions, especially in deeper basins. In contrast, some coastal species may benefit from the associated high nutrient levels. Historically, the increase in nutrient concentrations from the level before the 1950s to the 1980s possibly improved the growth of sprat and herring (Eero et al., 2016) and may have slightly enhanced the productivity of cod (Eero et al., 2011). By removing fish, fishing is considered to remove nutrients from the Baltic Sea (Nielsen et al., 2019), which is another interaction between eutrophication and fisheries.

Hazardous substances in the Baltic Sea interact with fisheries as well. Contaminants in fish above accepted thresholds have implications for marketing possibilities of the fish. In addition, contaminants can affect fish stocks via food web interactions. For example, a reduced level of hazardous substances has allowed the top predator grey seal population to increase in abundance. Seals are preying on fish resources and their increased abundance has led to an increased infection of cod with the seal-associated liver worm (Sokolova et al., 2018), which may affect cod condition and cause mortality (Horbowy et al., 2016).

Invasive species is another driver interacting with fisheries through food web interactions. An example here is round goby, *Neogobius melanostomus*, which has become a new exploitable resource for some fisheries, but also has negative impacts on some other native commercial species (Ojaveer et al, 2015). Importantly, it has been found to increase bioaccumulation of sediment-related toxins in the food chain, and thereby increase risks for fish consumers (Ojaveer et al., 2015 and references therein).


**Impacts of fisheries on other factors**

- For **hypoxia (?)**, there may be a cascading effect, e.g. fishing out large predators may affect consumption at lower trophic levels. In the end, this trophic cascade has repercussions on nutrient concentrations, eutrophication and

hypoxia (Nielsen et al., 2019).
- Fisheries strongly affects the **marine ecosystem (+)**. The withdrawal of large quantities of intermediate to top predators may cascade down to lower trophic levels in the pelagic and benthic zone, and affect higher trophic levels like marine mammals and sea birds (Jennings and Kaiser, 1998; Bergström et al. 2019; Elmgren 2021). There can be considerable detrimental effects to benthic communities through bottom trawls.





- Fisheries may directly or indirectly affect **non-indigenous species (?)** by altering the food web structure and opening up an ecological niche for new species, or may be a new commercially interesting species, e.g. the round goby (Ojaveer et al., 2015).

- There is no direct effect of fisheries on **nutrient loads (?)** but there can be feedback to coastal nutrient concentrations due to cascading effects if certain species are removed by fishing. By removing fish, fishing is considered to extract nutrients from the Baltic Sea (Nielsen et al., 2019).

- Fisheries may affect **aquaculture (?)** by giving an indication which fish species may be interesting to culture, or for which species there is a demand, which fisheries cannot satisfy. Here, management instruments like fishing quota may play a role.

- There is a connection between fisheries and **offshore wind farms (+)** regarding the competition for space and resources in the coastal areas (Methratta 2020). On the other hand, the bases of pillars have been shown to form artificial reefs, which can act as nursery grounds for specific fish species (Wilson and Elliot, 2009; Degraer et al., 2020)

- Likewise, a possible impact of fisheries on **shipping (?)** regards the competition for space.

- Fishing vessels have been a source for **marine litter (+)** and contamination, e.g. nylon nets, buoyancy gear, solid and liquid waste, similar to general shipping. Abandoned fishing nets, however are a special threat not only for fish, marine mammals (Stelfox et al., 2016) but also for birds (e.g. Merlino et al., 2018).

- Fisheries may have an impact on **tourism (?)**. In some holiday resorts, fishermen have switched from traditional fishing to offering recreational fishing tours to tourists, to selling fish (not necessarily own catches) from fishing vessels, or to touristic sightseeing boat tours altogether.

- Fisheries is an integral part of **coastal management (+)** in areas where fishing grounds or essential fish habitats potentially overlap with other uses of the coastal zone. While conflicts of this type are scarce in the Baltic Sea region, they are common in regions that use massively fish ponds at the coast (Kalther and Itaya, 2020).

**5.12 River regulation and stream restoration**

Many rivers in the Baltic Sea catchment basin are regulated, i.e. their natural course has been altered for power generation, municipal water supply, and irrigation for agricultural purposes, flood protection, shipping and navigation (e.g. Kelly et al., 2017). Damming for hydropower generation is more common in the northern, boreal part of Europe, where a considerable fraction of electric power generation (up to 82% in Norway and 77% in Finland) is by hydropower (Lehner et al., 2005, Humborg et al., 2015). Regulation of river basins and drainage works in agriculture and forestry have been major factors for changes in the hydrological and water quality responses in watersheds (Wörman et al., 2010). Nilsson et al., (2005) describe how the majority of the world's river basins are regulated. Such river regulations affect the river hydrograph, sediment transport and local erosion (Kumar et al., 2011). The dam structure and associated damming can cause permanent flooding with increased green house gas emissions from the flooded terrestrial-carbon pool, river fragmentation, higher sedimentation rate in the reservoir, retention of solute pollutions and risk for eutrophication in the reservoir. It can change the amounts and the stoichiometric ratios of the nutrients reaching the coastal waters, hence potentially modifying the coastal biogeochemistry (Humborg et al., 2006).

A multitude of drainage works in all agricultural land has gradually led to a more rapid hydrologic response over the last centuries and probably in general a corresponding decreased retention capacity for nutrients downstream, from land to sea (Weigelhofer et al., 2018). Traditionally, there has been an understanding that the stream network topology has the major importance for the hydrological response of a river basin – this effect is due to the so-called geomorphological dispersion affecting both water flow and solute responses (Rinaldo et al., 1991; Rodriguez-Iturbe and Rinaldo, 1997). However, more recently there has been a growing understanding that spatial heterogeneity of in-stream channel properties, such as those





affected by drainage works, play an additional important role for hydrological responses – the associated phenomena are termed hydraulic and kinematic dispersions (Saco and Kumar, 2002). Consequently, numerous drainage projects in agricultural land during several centuries have gradually implied a more rapid flow response (Åkesson et al., 2016) with consequences

also for the water quality response in streams and river basins (Riml and Wörman, 2015). Also, the interaction between the stream flow and shallow groundwater immediately below the stream water – the hyporheic zone – is highly important for filtering the stream water (Boano et al., 2014) and, thus, for the water quality of estuaries.

It has been shown that damming leads to a reduced silica transport to the sea (Humborg et al., 2000), due to diatom blooming in reservoirs and reduced weathering in the regulated rivers (Humborg et al., 2015). One option to reduce nutrient loading (in

particular nitrogen and phosphorus) is to implement local remediation measures within the agricultural drainage system that utilize the "self-purification" of the stream network. Such local measures – structures built in stream channels – limit the eutrophication both in downstream inland waters and can potentially have a major role in reducing the nitrogen loading to the sea (Seitzinger, 1988; Boano et al., 2014). A general understanding is that nitrogen removal in streams is controlled by biochemistry, but recently it has been found that stream hydro-mechanics impose a limit on the rate at which nitrate is removed

in the reactive zones of streams (Gomez-Velez et al., 2014; Grant et al., 2018; Morén et al., 2018). The past few decades of research on stream hydrology and biogeochemistry provide a picture of the so-called hyporheic zone as a hotspot for stream biogeochemistry and self-purification of the stream water (Boano et al., 2014). Restoration actions in streams can offer an important contribution to a system of management plans to reduce nitrogen and phosphorus loadings to inland waters and the Baltic Sea.

Stream restoration projects tend to reverse effects of previous drainage works by introducing engineered structures like cross-vanes, riffle-and-pools, new bed substrate or checker dams (Wortley et al., 2013). Such stream structures create localized hydraulic head drops in streams, which can increase the water flux into the hyporheic zone and, thereby, reduce both nitrate and phosphate transport (Morén et al., 2018). An engineering design approach is based on many, in practice, uncertain and variable factors that can also be difficult to control by engineered structures.

Nevertheless, a recent study of the potential for reducing nitrogen export through denitrification in agricultural streams via restoration actions indicates that the effectiveness is highly heterogeneous, depending on local stream conditions, but also that significant reduction in nitrogen export can be achieved through such actions if implemented wide-spread (Refsgaard et al., 2019).

The hydrological response changes over time due to landscape changes, periodicity of climate as well as global warming. The

energy level available for the transport of sediment and solutes in streams is highly variable (Wörman et al., 2017), which has significant environmental implications for decadal or longer time-scales. Climate change may generate higher runoff due to overall increasing precipitation, especially in the winter, with large regional differences (Graham et al., 2008; Räisänen, 2017; Christensen et al., 2021). Runoff peaks are expected to come earlier in the year in some regions and be less pronounced due to a lower snowmelt peak and a more spread-out precipitation volume across the winter. An increased runoff would generally

enhance nitrogen transport and decrease retention and transformation of nitrogen in streams. Impacts by river regulations on flow regimes and temperatures may be stronger or comparable to those by current climate change (Arheimer et al., 2017; Ashraf et al., 2016)

**Impacts of river regulation on other factors**


- There is a close connection to **coastal processes (+)**, as regulated rivers carry different amounts of sediments to the sea (Tena and Batalla, 2013), which can thus alter coastal processes and morphology in the vicinity (downstream) of river mouths of regulated rivers. Marinas at the river mouth may have a similar impact (Soomere et al., 2007). While



extensive damming of major rivers is detrimental for their deltas (Li et al., 2017), many small beaches experience permanent erosion because of the regulation of rivers that fed them with sand in the past (Vitousek et al., 2017).

- River regulations may have an impact on **hypoxia (?)** near river mouths, through altered nutrient loads, eutrophication and increased oxygen demand/depletion.

- River loading of total carbon and alkalinity is associated with weathering processes in the drainage areas where some are rich and some poor in limestone, affecting alkalinity and **acidification (+)** (Hjalmarsson et al., 2008). Differences in river concentrations of organic carbon and organic alkalinity (Kuliński et al., 2014; Ulfsbo et al., 2014) and in some drainage basins are associated with acid sulphur soils (Nordmyr et al., 2006).

- There is some evidence that regulated rivers have an impact on a drainage basin´s groundwater budget (Hancock, 2002), so there is a plausible connection to **groundwater discharges (?)** to the sea, but the uncertainty is high.

- River regulation and damming may lead to reduced nutrient transport to the sea, especially for silica (Humborg et al., 2000), due to diatom blooming in reservoirs and reduced weathering in the regulated rivers (Humborg et al., 2015). The changed amounts and stoichiometric ratios of the nutrients entering the coastal waters may modify the coastal biogeochemistry and hence affect the **marine ecosystem (+)**.

- There is a clear impact of river regulations on **land use (+)** and **agriculture (+)**. River regulation and restorations can be seen as a type of land use, and the fraction and distribution of useable land are partly determined by regulated rivers.

- River regulations substantially affect **nutrient loads (+)** through damming and sedimentation, changing the river´s nutrient concentrations and biogeochemistry, particularly for Si (Humborg et al., 2015). Still, it has been estimated that the global riverine nitrogen and phosporus transport has increased despite all regulation efforts (Beusen et al., 2016).

- If river passages are opened for upstream migrating fish, there is the risk of introducing pathogens from the marine environment into protected areas. The majority of Swedish **aquaculture (?)** of coldwater fish is harboured from marine pathogens in shielded areas. This needs to be managed if upstream migration is facilitated.

- There is a connection of regulated rivers with **fisheries (+)**, at least for some branches, as regulated rivers (barriers, dams, locks, modifications of the riverbed), make it difficult or impossible for some fish to migrate to their spawning grounds through the rivers (anadromous species like salmon, eel). Fish passes have been installed at many locks and barrier but passages are significantly lower than without barriers and differ tremendously between species and types of passes (Noonan et al., 2012; Bunt et al., 2012)

- For some **chemical contaminants (?)** (e.g. pharmaceuticals, Lindim et al., 2016) and trace metals, riverine transport is the major transport route to the sea. Degradation of chemical contaminants during river transport depends on several environmental factors including shading, nutrient conditions, turbidity, exchange between surface water and the hyporheic zone, the bacterial community in the sediment, and attenuation is therefore dependent on watershed characteristics and water residence time (e.g. Rieger et al., 2012; Posselt et al., 2020). It is therefore plausible that the extensive regulation of rivers in the Baltic Sea catchment impacts the attenuation of many organic contaminants and metals (Saniewska et al., 2014; Saniewska et al., 2018; Gębka et al., 2020).

- Rivers represent major pathways for microplastic emissions to the sea (Schmidt et al., 2017). In the Baltic Sea, microplastic input via rivers can be expected as well, although retention rates are unknown (Schernewski et al., 2021). River regulations may influence how much or which fraction of **marine litter (?)** reaches the sea. However, this is largely unknown.

- Regulated rivers change the amount and quality of water entering coastal waters so they have an impact on **coastal management (+)**.





### 5.13 Offshore Wind Farms

Regenerative power generation is on the rise to help reach decarbonization goals. For Europe, an increase of up to 15% in regenerative wind power generation is projected for the late 21$^{st}$ century (Tobin et al., 2015; Tobin et al., 2016). Wind power installations have increased in recent years especially in the southern Baltic Sea, and the prospects are substantial. By late 2019, 2 GW offshore wind power has been installed in the Baltic Sea and it is expected to be 9 GW to 14 GW by 2030 (Pineda and Fraile, 2019). WindEurope's latest scenario projects installations of 85 GW by 2050, making the Baltic Sea the second-largest basin for offshore wind in Europe (after the North Sea). Offshore wind power is expected to be competitive with other power sources in 2030, according to the ECDGE report (2019). Wind power generation is of course depening on future wind conditions, but they are not clear. Climate models do not project any consistent future trend for wind speeds in the Baltic Sea region (Räisänen, 2017; Christensen et al., 2021), except for an increase in near-surface wind speed in areas that today are covered by sea ice, and which are projected to have largely disappeared in a future warmer climate (Kjellström et al., 2018). Hence, only a drastic increase in the number of wind farms can yield a considerable increase in renewable energy production, with all its potential consequences on ecosystems and potential feedbacks to the regional climate. Moreover, the variability and expected technological development in turbine effectivity are expected to be larger than the estimated climate effects (Tobin et al., 2016). Irrespective of future wind conditions, problems regarding wind wake effects are to be expected in very large wind farms, with consequences for the efficiencies of these large farms (Akhtar et al., 2021).

There are several socio-economic and psychological aspects, which may affect the development of offshore wind energy generation in the Baltic Sea. The visual impact of offshore wind energy infrastructure is a considerable hurdle shaping the social acceptance of the surrounding communities to wind energy development (e.g. Upham and Johansen, 2020). On occasions, this has triggered economic compensation demands by citizens living in coastal areas as a retribution scheme to allow the development of offshore wind energy, an issue which can a) significantly further slowdown the development of new wind energy infrastructure, and b) add additional development costs should financial compensation need to take place. In that respect, various governments across the Baltic Sea region have included compensation mechanisms within their renewable energy system support policies.

Furthermore, the "viewshed effect" of offshore wind energy may have a particularly acute economic impact on coastal touristic destinations, an observation corroborated across multiple case studies in Spain (Voltaire and Koutchade, 2020), US (Landry et al., 2012; Lilley et al., 2010; Parsons et al., 2020), France (Westerberg et al., 2013), and Denmark (Ladenburg and Dubgaard, 2007), among other jurisdictions. This may lead to significant revenue losses for tourism-dependent businesses, potentially outweighing the economic profits stemming from offshore wind farm developments and ultimately resulting in a net welfare loss for the affected coastal region (Voltaire and Koutchade, 2020). However, there may also be limited benefits (Hooper et al., 2017).

Wind energy development may enhance social acceptance and positive economic distributive impacts under more collaborative procedural and co-ownership conditions whereby individual citizens are offered the opportunity to more proactively participate in the development of wind projects (Langer et al., 2017; Pons-Seres de Brauwer and Cohen, 2020). Importantly, the aggregated 'social potential' stemming from citizen-financed wind energy infrastructure development is significant under a European context (Pons-Seres de Brauwer and Cohen, 2020), and thus highly relevant (and potentially replicable) for the Baltic Sea Region (Pons-Seres de Brauwer and Cohen, 2019).

As of November 2020, there are in the Baltic Sea region (including Kattegat, Belt Sea and Lake Vänern), 19 wind parks in operation, with a total production of 2.2 GW, 2 out of operation and 4 under construction or in planning (https://de.wikipedia.org/wiki/Liste_der_Offshore-Windparks). Figure 3 shows a map of the offshore wind farms in the Baltic Sea (https://www.4coffshore.com/offshorewind/).



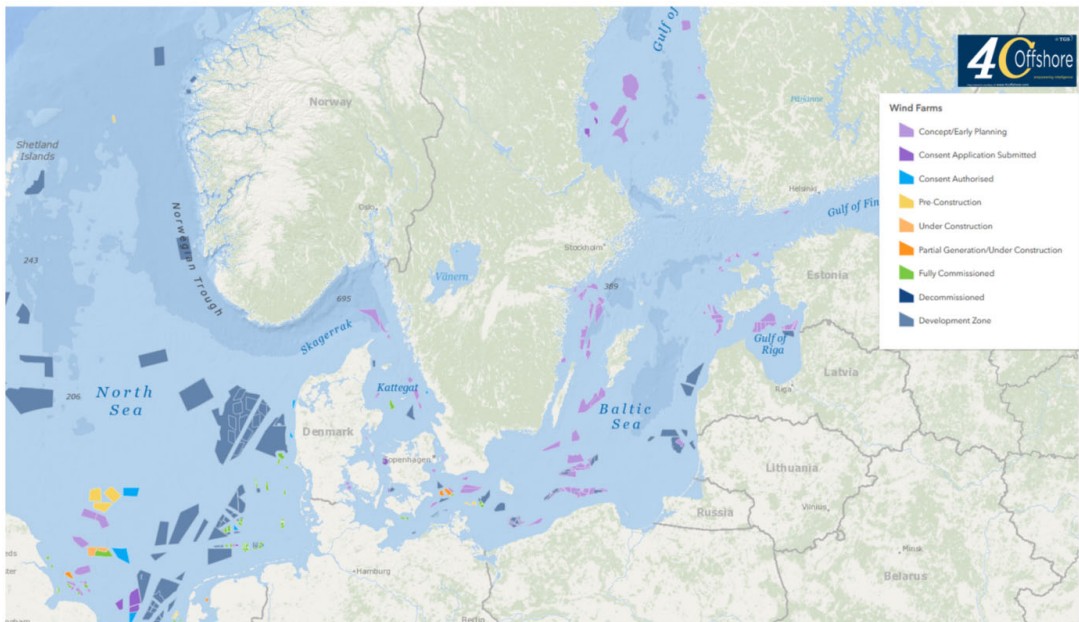

**Figure 3: Offshore wind farms in the Baltic Sea (https://www.4coffshore.com/offshorewind/), as of 24 June 2021, courtesy of 4COffshore.com.**

**Impacts of offshore wind farms on other factors**

- There may be a certain impact on the regional **climate (+)** by offshore wind farms through absorbing atmospheric energy. There is little information on the magnitude of this effect, and modelling exercises have found varying impacts on the regional climate at current densities of wind warms (Fitch et al., 2013; Vautard et al., 2014; Akhtar et al., 2021). Considerable impacts cannot be excluded in the future with an extensive development of renewable energy production to meet climate goals. Studies suggest that with climate change, the wind resource change in the Baltic Sea is not as significant as other European seas, with a majority of studies suggesting a gentle tendency of increasing resource in the Northern part (Devis et al., 2018; Reyers et al., 2016; Hahmann et al., 2020). However, the impact of the farms on the regional climate has not been assessed in this area, partly due to the scale of the offshore wind industry being still rather small. While observational evidence is scarce, numerical studies in other regions have suggested possible impacts of wind farms on local meteorology and climate, depending on the farm size and density, and turbine types. The impact from the farms on local meteorology can be seen in the formation of fog (Emeis, 2010; Hasager et al., 2017, North Sea), the change of spatial distribution of precipitation (Pan et al., 2018, US coast), cloud cover (Boettcher et al., 2015, North Sea), decrease in sensible heat flux (Foreman et al., 2017, North Sea), and local temperature (Roy and Traiteur, 2010, conceptual). Observational evidence has shown that offshore wind farms may affect the local climate by modifying the marine boundary layer (Siedersleben et al., 2018). The impact is mostly assessed through temperature, and the findings are rather consistent: there is only a statistically insignificant change in mean temperature, with seasonal peak values up to 0.5°C (Vautard et al., 2014; Keith et al., 2004; Pryor et al., 2018). In contrast, Wang and Prinn (2011) and Huang and Hall (2015) found a potential cooling effect in the vicinity of the offshore wind farms due to an increase in latent heat flux. Airborne observations confirm that wind farm wakes can extend 50–70 km under stable atmospheric conditions (Platis et al., 2018; Akhtar et al., 2021). These





measurements show that wakes can increase the temperature by 0.5 °C and humidity by 0.5 g kg$^{-1}$ at hub height, even as far as 60 km downwind (Siedersleben et al., 2018).

- The impact of wind farms on **coastal processes (+)** depends on the vicinity to the coast. Currents may be affected by pillars and sediment transport may be affected locally (Zhang and Wang, 2009; Besio and Losada, 2008). Coastal currents may lead to scouring and problems with the stability of pillars (Whitehouse et al., 2011). The disturbance to the downwind wave field heights was estimated to be minor (Alari and Raudsepp, 2012).

- There is a considerable impact of offshore wind farms on **marine ecosystems (\*)**. Noise from pile driving can cause temporal to permanent damage to marine mammals to different degrees, and cause their behavior changes in communication and travel (Southall et al., 2007). Cables during construction and electromagnetic field can also affect the orientation of those who use geomagnetic cues during migration (Lovich and Ennen, 2013). Tougaard and Michaelsen (2018) examined the impact of the wind farm Kriegers Flak in the Baltic Sea on marine mammals (specifically two species of seals) regarding underwater noise suggesting that noise from construction and operation are without significant long-term impact on the marine mammals. Wind parks may also host fish and sessile assemblages of organisms (Andersson and Öhman, 2010), and be selective hunting areas for harbor porpoises due to high fish abundances there (absence of fishing and artificial reef conditions (shown for the North Sea, Scheidat et al., 2011). Many species of water birds have been observed to react to the presence of a wind farm, from a few hundreds of meters to a few kilometers ahead, as observed over both the Baltic Sea and the North Sea (Hueppop et al., 2006). Most of them change flying route and fly around the farm, and very few (less than 1%) fly riskily close to the farm and end with collision, according to the observation around the Nysted wind farm in the Baltic Sea (Desholm and Kahlert, 2005). Large wind farm clusters may form a barrier effect to migrating birds, though some may fly into the space between the farms (Larsen and Guillemette, 2007). Studies for land birds affected by offshore wind farm are lacking. Potential impacts of wind wakes on hydrodynamics features of the downstream waterbody and ecosystems are discussed by van Berkel (2020).

- There is no clear connection between offshore wind farms and **land use (?)** except that an extension of offshore wind farms may result in a reduced number of wind turbine constructions on land. The same holds for the connection with **agriculture (?)**; land-based wind farms may need to be reduced to give space for agricultural fields. It needs to be taken into account that land-based wind energy constructions need to fulfill certain regulations concerning the vicinity to housing settlements, and that local communities often reject the construction of new sites in their neighborhoods, so that land areas for wind generation may be increasingly scarce in the future.

- There is a potential synergistic use of offshore wind farms related to certain types of **aquaculture (+)** in the Baltic Sea (Mikkola et al., 2018). The installation of open cages between the pillars was proposed to grow seaweed, rainbow trout or Atlantic salmon (Stuiver et al., 2016; Lengoburu et al., 2018). The good (shared) infrastructure, the placement in clean and open waters as well as the exemption from fishing and shipping have been reasons for considering this type of synergetic use. There are however, certain risks and obstacles associated which may have so far prevented a successful application (Buck et al., 2017; Chen et al., 2020; van den Burg et al., 2020).

- There is an array of possible impacts of wind farms on **fisheries (+)**. Wind farms cover large areas which are exempt from fishing, so there is competition for space. Studies have suggested that some fish species are affected by noise from foundation construction or operation (Thomsen et al., 2006). Some found evidence of injury from pile driving sounds for several fish species (e.g. Casper et al., 2012; Casper et al., 2013), and noises and consequent vibration produced by the turbines can negatively affect the communication and orientation signals of fish (Wahlberg and Westerberg, 2005). Their behaviors (e.g. swimming route) can be disrupted by the magnetic fields from the electrical currents in the transmission cables (Ohman et al., 2007). On the other hand, these large areas banned from fishing may act as spawning grounds for fish due to banned fishing and the functioning of windmill pillars as artificial reefs



(Wilson and Elliot, 2009; Degraer et al., 2020). There is evidence for increased fish populations in the presence of wind farms (Leonhard et al., 2011, Methratta, 2020).

- Offshore wind energy infrastructure may have important disruptive impacts on the **shipping (+)** routes of cargo vessels (Samoteskul et al., 2014). In case of route obstruction, wind farm owners must financially compensate cargo vessel operators for detouring from their shipping routes. One such example is the Anholt wind farm in the Baltic Sea (Petersen et al., 2015). This represents a significantly high added cost to be internalized during the offshore wind farm development process (Samoteskul et al., 2014). Consequently, offshore wind energy infrastructure may therefore be built in areas away from recognized shipping routes and anchoring locations (so as to avoid collision and subsequent financial compensation to vessel operators) while simultaneously avoid nearshore siting, as this may reduce social acceptance due to the infrastructure's visual impact on the population living in coastal areas, an effect that can have significant economic implications particularly in coastal touristic areas with high recreational value (see social acceptance section below). Alternatively, cargo vessel routes ought to be modified on a permanent basis, an action that could significantly reduce the financial cost of future offshore wind farm developments (Samoteskul et al., 2014).

- There are potential emissions of **chemical contaminants (+)** from all offshore activities due to increased emissions from constructions and traffic leading to disturbance of seabed sediments (release of contaminants in sediments and chemicals used in the infrastructure, leakage through lubricants, other material etc., e.g. metals, biocides, oils, coolants, dielectric fluids. However, there are no investigations on the magnitude of this potential contamination (Tornero and Hanke, 2016; Ytreberg et al., 2020)

- As with any offshore activity, windfarm constructions may affect **dumped munitions (?)**, due to possible breach of munition hulls. Since wind farms are built away from official dumpsites, solitary munitions or unofficial dumpsites are the main risk factors. In 2017, the construction of a wind farm in the North Sea released an abandoned seamine from the sediments that was later found floating between the piles of the GodeWind 2 farm (Schuler, 2017).

- **Marine litter (?)** could be generated through the maintenance and traffic related to the offshore constructions, but there are no investigations on this connection.

- Offshore wind farms may have an impact on coastal **tourism (+).** The so-called "viewshed effect" or "horizon pollution" (in Germany) of offshore wind farms may have adverse economic impacts on coastal touristic destinations (e.g. Ladenburg and Dubgaard, 2007), and may result in a net welfare loss for the affected coastal region (Voltaire and Koutchade, 2020). There are, however, controversial views (Hooper et al., 2017).

- Wind farm planning is a part of **coastal management (+)**, i.e. governments and local authorities attempt to balance and manage their use and development in coastal zones. On the one hand, socio-economic effects, such as revenue losses for tourism-dependent businesses, possibly outweigh the economic benefits from offshore wind farm developments and resulting in a net welfare loss for the affected coastal region (e.g. Voltaire and Koutchade, 2020). On the other hand, recent modeling results indicate that large offshore wind farms may affect the wind resources and impact power production in the future (Lundquist et al., 2019). Wind resources are limited and large wind farms may reduce the harvestable wind due to shadowing effects. Due to the large size of the upwind farms, the power production of downwind turbines may be compromised.

### 5.14 Shipping

Shipping has a significant impact on the environment, both in the water and in the atmosphere. The impacts are manifold and include the release of toxic bio/antifouling agents from ship hulls, the release of ballast water and waste, black and grey water, scrubbing and bilge water, the generation of surface waves and underwater noise, the unintended transport of alien passengers on ship hulls and in ballast water tanks, and the release of combustion products to the atmosphere.

The Baltic Sea has some of the densest maritime traffic in the world with more than 2,000 ships in the area, on an average day (IMO, 2021b). Nowadays, 80 % of the world´s trade is operated by sea traffic (UNCTAD, 2019), and 15 % of the global cargo

trafficis via the Baltic Sea (BalticLINes, 2016). Ships carry oil, gas, containers and large freight. In the Baltic Sea, the main shipping route is from the Belt Sea in the west to Saint Petersburg and other ports in the eastern Baltic Sea. The main hazards on this route are the shallow and narrow Katet channel between Falster, Denmark and the Mecklenburg coast, Germany, and the Danish straits. The northern part of the Baltic Sea and the Gulf of Finland and Gulf of Riga can also be affected by severe ice conditions in winter.


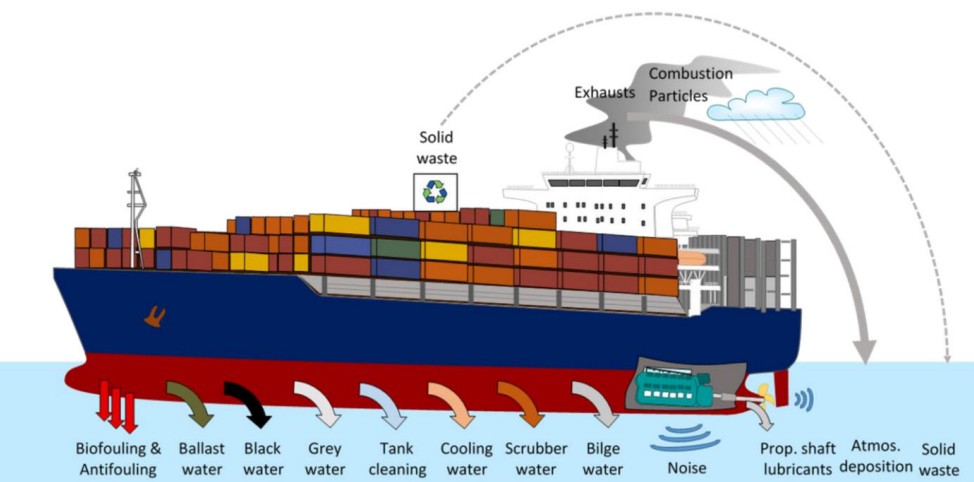

**Figure 4:** Different impacts of shipping on the environment. Hydrodynamic impacts such as wakes are not depicted but described in the text. From Moldanová et al., 2018.

Air emissions from shipping are fairly well known as vessel activity can be tracked using ship specific positioning systems like AIS (Automatic Identification System), and emissions as well as discharges can be estimated using modeling tools (Jalkanen et al., 2009; 2012; 2018; 2020; Johansson et al., 2013; Johansson et al., 2017). Shipping contributes to various environmental pressures, some of which are depicted in Figure 4. Various pollutant streams are regulated by conventions by the International Maritime Organisation (IMO MARPOL, IMO 2021c), which sets the rules for air emissions and various

discharges from ships. Antifouling and ballast water releases are governed by separate conventions outside the MARPOL framework, but underwater noise from ships remains unregulated. The reduction of air emissions has been primarily motivated by impacts on human health (Sofiev et al., 2018; Jonson et al., 2015; Brandt et al., 2013; Mwase et al., 2020; Lin et al., 2018; Karl et al., 2019; Ramacher et al., 2019; Soares et al., 2014). Ship-emitted $NO_x$ depositions contribute to eutrophication with less than 10% of various biogeochemical variables, but this share is three to eight times larger than shipping contribution to

total nitrogen input to the Baltic Sea (Raudsepp et al., 2013; Raudsepp et al., 2019).

Sulphur emissions from shipping fuels (crude oil) are a particular problem for the environment and human health (Barregard et al., 2019). Efforts to reduce sulphur have led to the adoption of SOx scrubbers, which are used to clean the exhaust gases by spraying them with water, which is released back to the sea. The wastewater release from scrubbers represents the second largest discharge from ships to the sea. The full impacts of $SO_x$ scrubbing are currently unknown, but several countries have

prohibited the use of $SO_x$ scrubbers in port areas, anticipating potential water quality problems. As of 2021, $NO_x$ emission regulations are planned to be in force for new ships (IMO, 2021b). This requirement is not applied retroactively for old vessels, which means that the full 80% reduction on ship emitted $NO_x$ will be visible only after the whole Baltic Sea fleet has undergone





a renewal cycle, which may take up to 30 years. It is possible to adapt to both sulphur and nitrogen regulatory changes by using Liquid Natural Gas (LNG) as a shipping fuel. To this date, LNG mostly consists of methane, which is a fossil fuel.

Unburnt methane may also escape the ships' engines thus leading to methane slip, making it more difficult to achieve GHG reduction targets set for international shipping (Ushakov et al., 2019).

Sewage releases to the Baltic Sea will become illegal for passenger traffic in 2023, which will reduce nitrogen inflow from ships to the sea by 90% (Jalkanen et al., 2020). The introduction of non-indigenous species through vessel hulls can be mitigated by using antifouling paints. Organotin compounds have been banned for more than a decade (IMO, 2021d), but these

and various other organometallic compounds remain in use especially in recreational boats (Eklund et al., 2018). The traffic patterns of ships and recreational boats are different: large vessels travel along designated shipping lanes, whereas small boats mainly operate in coastal waters. The maximum release of organometallic compounds from antifouling paints occurs during the summer months, when contributions from both shipping and boating are at maximum. Estimated annual quantities of copper released to the Baltic Sea are about 282 and 57 tonnes for ships and boats (Jalkanen et al., 2020; Johansson et al., 2020).

Oily bilge water release is allowed if their oil content is below 15 ppm and the vessel is not in coastal waters. Discharges of grey water (wash water from sinks, washing machines etc), emissions of energy (noise, light, heat) to the sea are currently not regulated, but the importance of noise as pollution has been recognized (Mustonen et al., 2019; Jalkanen et al., 2018).

**Impacts of shipping on other factors**


- Shipping has a considerable impact on **climate (+)** through the emission of combustion gases and particles/aerosols (black carbon, methane, $CO_2$) to the atmosphere. In the Baltic Sea region, $CO_2$ from shipping is less than two percent of global $CO_2$ emissions from ships (Johansson et al., 2017). Global shipping emitted about over billion tonnes of $CO_2$ to the atmosphere in 2018 (IMO, 2021e), with a Baltic Sea fleet share of about 15 million tonnes (Johansson et

al., 2017).

- Shipping effects on **coastal processes (+)** such as erosion become noticeable along shorelines in relatively sheltered coastal regions where the impact of ship-induced waves add to the impact of natural waves. Hydrodynamic impacts of shipping in rivers, navigational channels and archipelago waterways have been known for decades (Madekivi, 1993). Shipping may dominate the coastal processes regime by generating dangerous waves and swell-like

disturbances in narrow passages and rivers and even on relatively open shores with potential harm for banks, beaches and coastal infrastructure (Kelpšaite et al., 2009; Soomere et al., 2009; Zaggia et al., 2017; Scarpa et al., 2019; Ulm et al., 2020), cause extensive resuspension of bottom sediments (Erm and Soomere, 2006), trigger ecological disturbance and cause harm to the aquatic wildlife (Ali et al., 1999; Lindholm et al., 2001). The impact of ship-generated waves may become substantial in areas where either the wave period or approach direction deviates from

those typical of natural waves. Several parts of the Baltic Sea (most notably Tallinn Bay) with high traffic of strongly powered ships are affected by much longer waves than wind waves in the area (Soomere, 2007). Such waves cause unusually strong impacts at a certain depth that first becomes evident via intense sediment resuspension (Erm and Soomere, 2006) and later may be compensated by sediment from the beach profile (Soomere, 2007). The increased local hydrodynamic activity may damage various structures and archaeological sites, and safety problems for

navigation and users of the beach and nearshore may arise (Parnell and Kofoed-Hansen, 2001).

- Scrubber water increases **acidification (+)**. According to the IMO requirements, the pH of the effluent discharge must not be lower than 6.5 and the difference between inlet and released water must be less than two pH units (IMO, 2009). Even with these requirements, gradual acidification of ocean areas may occur with a high adoption rate of open loop scrubbers as a means to comply with sulphur emission restrictions. Ocean acidification because of the climate change

and $CO_2$ solubility is estimated as 0.002 pH units per year (Rhein et al., 2013). In contrast, scrubber adoption is





estimated to reduce pH with an additional 0.0001 pH units per year (Turner et al., 2018). Confined water areas, like estuaries and ports, may experience larger reductions (up to 0.015 pH units, Teuchies et al., 2020).

- Shipping has various impacts on **marine ecosystems (+)**, e.g. the pollution by chemical substances and antifouling agents, the release of nutrients to the water and to the atmosphere, acidification by scrubber water, contribution to marine litter and marine noise. Shipping contributes to continuous low-frequency underwater noise, which may have adverse effects on marine life (Nedwell et al., 2004; Rolland et al., 2012; Mustonen et al., 2019). Furthermore, the leaching of organometallic compounds, especially those of Cu and Zn from antifouling paints on ship hulls are high (Eklund and Watermann, 2018; Jalkanen et al., 2020; Lagerström et al., 2020) and affect organisms.

- There is a clear connection to the introduction of **non-indigenous species (+)** as ballast water or attachment on hulls are a major pathway for the introduction of new species (Bressy and Lejars, 2014; Davidson et al., 2009).

- It has been shown that shipping is a significant source for the emission of airborne nitrogen into the atmosphere. Its contribution from ships may be less than 3%, but its share from various biogeochemical variables may be as high as 10% (Raudsepp et al., 2019). Direct discharge from ships to the sea includes **nutrients (+)** and pharmaceuticals in the form of black, grey and bilge water, but also as food waste (Jalkanen et al., 2020).

- Shipping may have an impact on open water **aquaculture (?)** farming by excluding shipping routes or endangering safe cages and potential escape of non-inidigenous species to the environment by collision or swell damage. In October 2016, a cargo vessel collided with an aquaculture cage in Danish coastal waters, causing 80.000 rainbow trouts to escape the closed farm, with unknown consequences for the coastal ecosystem (Reuters, 2016); similar incidents have been reported by local fishermen.

- There is a connection between shipping and **fisheries (+)** through competition between fishing grounds and shipping routes, the generation of underwater noise, and the contamination of fish by heavy metals and antifouling agents.

- There may be an impact of shipping on **offshore wind farms (+)** through the danger of collisions in detrimental conditions (storms, loss of maneuverability). Furthermore, the location and approval of wind farms are dependent on shipping routes. Areas for specific purposes are allocated by maritime spatial planning (HELCOM, 2013, an example for the Gulf of Bothnia).

- Shipping is a significant source of water pollution in general, also for **chemical contaminants (+).** This is through the release of organic contaminants and heavy metals to seawater through scrubber water, and other contaminated water (black and grey water) and antifoulings, in particular copper and zinc (Jalkanen et al., 2020; Magnusson et al., 2018; Ytreberg et al., 2020). Polyaromatic hydrocarbons (PAH), e.g. pyrene, are carcinogenic compounds formed during combustion, of which particularly high concentrations are found along shipping lanes due to the release of bilge and scrubber water.

- Furthermore, shipping can be a source of **marine litter (+)** although it is not considered the main source (e.g. Graca et al., 2017)

- Shipping has an impact on **tourism (+)** as coastal touristic activities involve recreational boating, either on a guided basis (touristic boat trips or recreational fishing trips), or on an individual basis (recreational small-vessel leisure boating). Another dimension is the growing cruise ship sector, which as grown to a large commercial sector, providing many jobs in various branches, also in the target harbours, but having a detrimental impact on the environment (air pollution, scubber water, litter, marine noise) and disturbance of local communities.

- Shipping may have an impact on **coastal management (+)** as some impacts on coasts and coastal structures as well as in rivers (damage through waves and swell, unprotected coastlines affected by swell) is evident (e.g. Zaggia et al., 2017; Jägerbrand et al. 2019).



**5.15 Chemical contaminants (with an emphasis on organic contaminants)**

Thousands of organic chemicals, both synthetic and naturally occurring, are released intentionally or unintentionally to the Baltic Sea environment due to human activities. It is well known that organic contaminants can negatively affect aquatic organisms at different trophic levels, illustrated e.g. by previous severe effects on marine predator populations (white tailed eagle, seals etc.) in the 1970s and 1980s (Sonne et al., 2020). It is unclear if the total anthropogenic chemical stress to the Baltic Sea is currently increasing or decreasing. In many cases, e.g. for banned chemicals that are monitored, environmental/biotic concentrations are declining, although emissions from remaining reservoirs in the technosphere and buffering by secondary sources such as soils and sediments delay their elimination (Breivik et al., 2016; Glüge et al., 2017; Abbasi et al., 2019; Sobek et al., 2016). Dioxins and dioxin-like polychlorinated biphenyls (PCBs) are for example still present in Baltic Sea fish in too high levels, making sales restrictions and recommendations of maximum fish intake necessary to protect human health (Pihlajamäki et al., 2018).

Legacy pollutants still dominate the burden of some groups of persistent organic pollutants analyzed in Baltic Sea marine mammals and birds, due to their persistence and exceptional bioaccumulation potential. However, analysis of less well-studied organic contaminants, often replacements for the legacy pollutants, indicate that levels in fish and mussels are now similar or exceeding their predecessors (de Wit et al., 2020). The lacking control of identity and amount of emitted substances hampers characterization and quantification of combined toxic effects in the Baltic Sea (Lethonen et al., 2017; van den Brink et al., 2018). It is notable that Baltic Sea organisms are particularly sensitive to toxic chemicals, as many marine species already live in brackish water at non-optimal salinity, i.e. under osmotic stress. Organic contaminants can reduce the resilience to other stressors by influencing the fitness of the organism, e.g. the key physiological mechanisms to maintain homeostasis (Noyes et al., 2009).

Direct effects of climate change include an array of processes. Changing environmental temperatures affect diffusive partitioning between environmental phase-pairs such as air-water, air-aerosols, air-soil, air-vegetation, leading to a different distribution between environmental compartments, like increased volatilization from seawater to air (Macdonald et al., 2003). Increasing temperatures can enhance photo- and hydrolytic degradation as well as biodegradation of organic contaminants (Noyes et al., 2009). Atmospheric transport and air-water exchange can be influenced by changes in wind fields and, to a lesser extent, wind speeds (Lamon et al., 2009; Kong et al., 2014). Changing precipitation patterns influence chemical transport via atmospheric deposition (rain dissolution and scavenging of particles, Armitage et al., 2011) and runoff, in turn transporting terrestrial organic carbon (Gustavsson et al., 2019; Josefsson et al. 2016; Ripszam et al., 2015). As ice cover in lakes and the sea decreases, more organic contaminants may volatilize to the atmosphere (Macdonald et al., 2003; Undeman et al., 2015).

Hence, climate change and other factors can directly or indirectly influence (reduce or enhance) concentrations of organic contaminants in different environmental matrices by impacting emissions, transport and transformation of chemicals (Macdonald et al., 2003; Noyes et al., 2009; Kallenborn et al., 2012; Balbus et al., 2013). How an organic pollutant is impacted by changing environmental characteristics (such as temperatures, wind speeds, organic carbon content in soil, water, air, food chain structure) depends on the emission patterns (if emissions occur to air, soil or water), as well as physical-chemical properties of the pollutant (e.g. water solubility, vapor pressure, hydrophobicity, degradability in various media and organisms) (Wania and Mackay, 1999; Meyer et al., 2005). The responses are complex and several processes can act antagonistically. For example, warmer temperatures may lead to re-volatilization of organic contaminants in soils, but may also lead to increased degradation in the atmosphere and the environment in general. This latter effect, however, can be expected to be weaker than the former (Armitage et al., 2011).

Modeling studies that assess the total effect of climate change on environmental concentrations for a wide range of organic pollutants are scarce. However, several case studies for selected substances including PCBs, dioxins, toxic fungicides such as Hexachlorobenzene (HCB) and hexachlorocyclohexane isomers (HCH) have been conducted (Gouin et al., 2013; Wagner et al., 2019). Due to counteracting effects, the total impact of climate change on environmental concentrations compared to



baseline are for many persistent organic pollutants simulated to be within a factor of about two (Gouin et al., 2013; Wagner et al., 2019). Wagner et al. (2019) estimated that changes in sea ice retreat, surface temperatures and changing ocean circulation has decreased water concentrations in the Northern hemisphere of lighter molecular weight PCBs, but on the other hand slowed concentration declines of high molecular weight PCBs; the different responses explained by their differing volatility and

sorption to organic matter. A modeling study by Kong et al. (2014) assessed the impact of climate change on concentrations of different organic pollutants specifically in the Baltic Sea. Concentrations of highly volatile compounds were projected to typically increase due to climate change in all compartments and under all emission modes. For more water soluble and hydrophobic compounds, projected concentrations increased due to climate change mainly in air and the marine compartments, and decreased in soil and freshwater (Kong et al., 2014).

Hydrophobic organic contaminants adsorb to organic carbon, hence, changes in organic carbon cycling may influence the distribution of organic contaminants (Nizzetto et al., 2012). Increased primary production in the sea influences the air-water exchange of some organic contaminants (Dachs et al., 2002). The downwards transport of organic contaminants via sedimentation of particulate matter increases with increasing primary production (Nizzetto et al., 2012). The concentration of particulate organic matter in the water column reduces the bioavailability of organic contaminants as they adsorb to the

particles (Borgå et al., 2010). In the Baltic Sea, eutrophication leads to hypoxia and anoxia in bottom sediments, which reduces the activity of benthic organisms, and hence bioturbation (Thibodeaux and Bierman, 2003; Granberg et al., 2008). This may lead to a reduced release of organic contaminants, archived in the sediments. Invasive species such as the deep-burrowing polychaete *Marenzelleria spp.* or higher abundances of the native bioturbating species *Monoporeia affinis* may cause the opposite effect (Hanson et al. 2020). Moreover, eutrophication as well as increasing terrestrial inputs of organic matter can

lead to changes in marine food web structure, which indirectly may influence bioaccumulation (Wikner and Andersson 2012). The same factors also change the light regime in the water column, which in turn affects photolysis of organic contaminants, e.g., polybrominated diphenyl ethers (PBDEs) (Kuivikko et al., 2007; Leal et al. 2013; McGovern et al., 2020).

Climate change may affect bioaccumulation in food webs directly by influencing body size, growth rates and conditions, temperature dependent ventilation rates, or biotransformation rates (Borgå et al., 2010; Alava et al., 2017). Indirect impacts

may also be due to changes in primary production, the number of trophic levels (increases bioaccumulation if biotransformation is slow) and diet preferences (e.g. a shift from pelagic to benthic food chain, or to prey at a higher trophic level). Concentration trends of legacy pollutants in Arctic wildlife have been attributed to climate change-induced changes. With earlier ice break-up in the year, polar bears starve and switch to more contaminated prey which results in higher concentrations of contaminants in the tissue (McKinney et al. 2009; Jenssen et al., 2015). Increasing PCB concentrations in burbot were connected to increased

organic matter concentrations (Armitage et al., 2011). In the Bay of Bothnia, low growth rates may explain the observed lack of decreasing dioxin levels in herring during the last decades (Miller et al., 2013).

Indirect effects can be induced by changed human activities due to climate change. Increasing temperatures can enhance the volatilization of chemical components in materials and stockpiles, affect land use, yields and types of crops, leading to a different use of pesticides. Similarly, a wider distribution of pests in changing ecosystems may be controlled by an increased

or changed use of pesticides. Potentially increasing forest fires may result in elevated emissions of combustion by-products such as polycyclic aromatic hdrocarbons (PAHs) (Gouin et al., 2013).

Emissions, transport and transformation of organic contaminants, as well as susceptibility of aquatic organisms are influenced by climate change via various mechanisms. Observed ecotoxicological effects of organic contaminants affecting health and reproduction in various species indicate that future changes in environmental concentrations can affect food web composition

and biodiversity in the Baltic Sea (Sonne et al., 2020). Although considerably less studied, it has in recent years been shown that organic contaminants can modulate microbial communities' composition and functioning, and potentially also biogeochemical processes of importance to Earth system functioning (Vila-Costa et al., 2020).



**Impacts of chemical contaminants on other factors**

- Toxic effects of organic contaminants and metals affect all types of organisms, e.g. health conditions and reproduction, and **marine ecosystems (+)** and biogeochemical processes in general (Vila-Costa et al., 2020).
- Chemical contaminants are ubiquitous in the environment and many are used on purpose in **agriculture (+)** (e.g. pesticides, insecticides). So, the desired effects on growth efficiency and agricultural yield are accompanied by unwanted and largely unknown negative effects on organisms and the food chain (Kumar et al., 2019).
- **Aquaculture (+)** fish may be strongly affected by organic contaminants, sometimes more than wild fish, as they are exposed to higher concentrations of deliberately dispensed pharmaceuticals, e.g. antibiotics (Cole et al., 2009).
- Organic contaminants have a strong indirect impact on **fisheries (+).** Contaminants in fish above accepted thresholds have implications for marketing possibilities of the fish. High concentrations of contaminants, e.g. dioxins, affect the marketing of the fish (fatty fish cannot be marketed in Europe, although exemptions exist for Sweden, Finland and Latvia). In addition, contaminants can affect fish stocks via food web interactions. For example, a reduced level of hazardous substances has allowed the top predator grey seal population to increase in abundance. Seals are preying on fish resources and their increased abundance has led to an increased infection of cod with the seal-associated liver worm (Sokolova et al., 2018), which may affect cod condition and cause mortality (Horbowy et al., 2016).

### 5.16 Unexploded ordnance and discarded military material

As a result of military conflicts in the 20[th] century, large quantities of warfare material ended up in global rivers, lakes, seas and oceans. Thousands of tons of various poisonous chemicals were purposely and accidentally submerged in coastal and deep-sea areas, and the Baltic Sea is no exception (Figure 5). Navy units and munition transports lost cargo or were destroyed during battles and later sunk, while aerial raids dropped significant amounts of bombs and aerial mines to coastal areas. Mine warfare, intense in both World Wars, introduced about 160.000 mines to the Baltic Sea, of which barely 20% has so far been removed or destroyed in clearance operations. This "Unexploded Ordnance" (UXO) is dispersed in many areas of the Baltic Sea. In addition, the Baltic Sea was used as a dumpsite for at least 40.000 tons of chemical munitions (Discarded Military Material, DMM). Sea dumping operations took place soon after World War II, leaving official and unofficial underwater dumpsites unmonitored for several decades (Knobloch et al., 2013).

Dumped chemical munitions pose a recognized environmental hazard for marine ecosystems. Recent studies performed in the dumpsites revealed that 50% of inspected unexploded ordnance (UXO) and Discarded Military Material (DMM) have already corroded, and their constituents have leaked to the surrounding sediments. Many substances among explosives and chemical warfare agents used as munition fillings have a demonstrated toxicity on terrestrial organisms; therefore, the material poses a potential threat also to aquatic organisms. On the other hand, the conditions in water are dramatically different. Solubility, oxidation and hydrolyzation are among various factors that shape the fate and bioavailability of a chemical compound in aquatic ecosystems. The environmental pathways of degradation, transport and transformation of explosives and chemical warfare agents are complex and generally depend on multiple factors. However, it can be concluded, that those substances are persistent, and their degradation products may be as toxic as their parent compounds (Czub et al., 2020). Hydrodynamic models indicate that they may spread into neighboring areas, increasing chances of biological uptake. Indeed, there are first reports of bioaccumulation of explosives and chemical warfare agents in organisms in the Baltic Sea (Niemikoski et al., 2017). Recently, complex risk assessment methods, using neural networks have been developed, taking into account environmental parameters, state of munitions, and the potential impact on biota, to assess present and future risks of dumped munitions.

The corrosion process of munitions at the Baltic Sea bottom progresses, and the impact on the environment is expected to increase in the near future. According to corrosion models, many containers have already released toxic substances to the environment, while others could do so in the next 30-40 years. At the same time, intensifying anthropogenic activities could



disturb munitions and accelerate this process. This is mostly connected with the offshore industry and increased use of the sea bottom. Hence, several scenarios should be considered:

1.  Slow release of toxicants and local contamination maintained
2.  Gradual increase of release and spread of contaminated areas
3.  Rapid release of contamination and massive pollution
4.  Possible beaching of munitions or munition fragments, and impact on tourists

The first two scenarios depend mostly on natural conditions, and the magnitude of pollution can be assessed by existing models. In this situation, munition is only one of many stressors acting on the Baltic Sea, and can be included in an overall assessment. In the third scenario, severe consequences for the entire Baltic Sea or specific areas adjacent to dumpsites result from
anthropogenic intervention, which is hard to predict. The last scenario is already ongoing – periodic encounters of beach strollers with UXO or fragments of munitions, especially incendiary like white phosphorus, happen every year (Frank et al., 2008; Knobloch et al., 2013). The process may intensify in case of progressing corrosion of containers or anthropogenic disturbance of munitions due to offshore activities.

With increasing marine traffic and expansion of offshore activities, the presence of scattered explosives and dangerous
chemicals pose a threat to workers and overall safety in the seas. The first two scenarios may have an impact on fisheries, by affecting fish health and diminishing recruitment and, a limited impact on fish consumers, as they assume low contamination. There are also no existing quality regulations for food contaminated by chemical warfare agents. Safe consumption rates of fish are not known yet, which puts the whole Baltic Sea area fisheries industry at potential risk. For example, the Bornholm Deep is a prominent dumpsite of warfare agents and it is at the same time the only spawning area for the migrating Eastern
stock of Baltic Sea cod, a heavily harvested fish population. The third scenario may have a detrimental impact on offshore economy, a loss of fisheries and loss of tourism. The fourth scenario may have negative implications on tourism, and high investments of coastal communities on maintaining safety on the beaches.

The first scientific examinations of dumpsites did not start before the late 1990s (Glasby, 1997) and gained momentum in the last decade, so considerable knowledge gaps exist. Most of the research performed in the Baltic Sea area focused on chemical
munitions. Degradation processes of chemical warfare agents and explosives are almost fully recognized, as well as transport mechanisms, although the list of breakdown products is still incomplete (Mazurek et al., 2001; Sonderstrom et al., 2018). However, not much is known about the metabolic pathways of munition related compounds in biota. This may lead to an underestimation of sublethal effects of those compounds. Further studies are needed to identify all the degradation products, their lifetime in the marine environment, and toxicity thresholds of their metabolites. Additional surveys of scattered munitions
are needed to quantify their amount and state of corrosion.

The deep basins of the Baltic Sea with their partly hypoxic and anoxic "benthic deserts" overlap largely with the deep-sea warfare agent dumpsites. Local biodiversity is low but irregular Major Baltic Inflows can replenish the oxygen supply, resulting in a temporal return of benthic organisms to close vicinity to the leaking objects (Czub et al., 2018). Simultaneously, the munition related pollution is greatly dependent on corrosion. It is especially enhanced during anoxic to oxic transitions in
the bottom water, exceeding the rates in stable oxic environments (Vanninen et al., 2020). Therefore, the frequency of anoxic events may amplify the release of the pollutants, while oxygen-rich conditions can increase their bioavailability.

Elongated warm periods caused by climate change can significantly affect munitions in shallow waters, where mostly conventional warfare materials were dumped. Not only the presence of hard metal objects as substrates for colonialization in soft sediment areas can increase the local biodiversity of sessile species, but the chunks of organic compounds used as
explosives can also attract primary and secondary producers as a source of nutrients. This is caused by the release of nitrates during the biodegradation of TNT and similar substances (Jessim, 2018). Higher-level organisms, e.g. nematodes, were found in the contaminated sediments, followed by various biofilm grazers.





Due to longer vegetation periods in a warmer climate, the extended transfer of carcinogenic degradation products of explosives may occur for a larger part of the year. Apart from that, the sympathetic effects of other pollutants, such as heavy metals, that

are often associated with munitions (Gębka et al., 2016) and persistent organic pollutants (POPs), may further enhance the toxic effects of munition related contaminants. The analysis of biomarkers for environmental stress in fish and mussel from the dumpsites shows that chemical warfare agents and explosives act in a similar way on marine organisms, therefore the existence of other stressors can amplify the adverse effect.

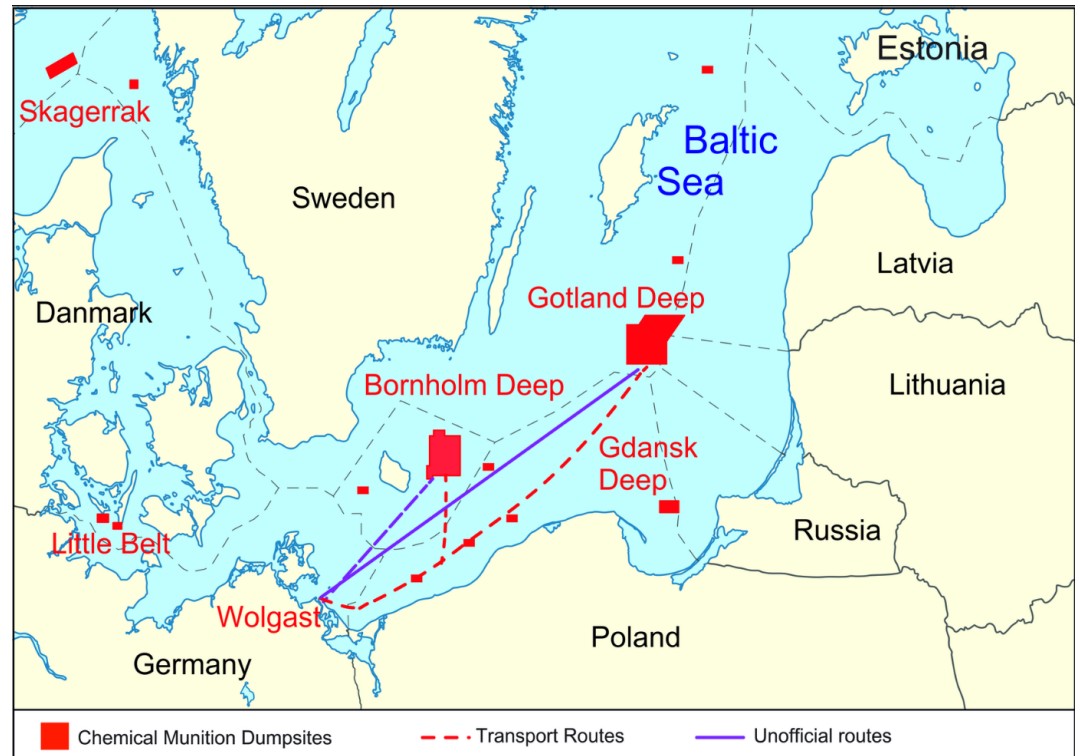


**Figure 5: Chemical munition dumpsites in the Baltic Sea (Jacek Bełdowski, CHEMSEA project 2013)**

**Impacts of dumped military material on other factors**

• Impacts on **Marine ecosystems (?)** and effects beyond local vicinity aound the sources of contamination are largely unknow (Maser and Strehse, 2020). However, effects on the upper food web and consumable fish and shellfish have been shown (Maser and Strehse, 2021).
         • Dumped ammunition sites can be dangerous for **fisheries (+)**, and poisonous substances can cause fish diseases. Fisheries can be affected by fish health, diminished recruitment, and consumer health. The consequences are largely
unknown but may be significant. For example, the Bornholm Deep is a prominent dumpsite of warfare agents and it is at the same time the only functioning spawning area for the migrating Eastern stock of Baltic Sea cod (Sanderson et al., 2008; Köster et al., 2017; Lang et al., 2018). There have been reports of fishing vessels being affected by dumped military material. In total about 200 fishermen were injured by exposure to chemical warfare agents since dumping (Sanderson and Fauser, 2015). Despite the fact that bottom trawling is restricted or not advised in chemical
munition dumpsites, trawlmarks are present on the bottom there, some of them freshly made (Klusek and Grabowski,



2018). As conventional and chemical single munitions are located outside official dumping grounds, the risk of encounter still exists. Disturbance of munitions by trawling gear can both speed up munition casing breach, and endanger crew by explosion or contamination.

- Dumped munitions may affect the development of **offshore wind farms (+)** as these installations need to be installed at a safe distance from dumping sites. This also holds for all offshore activities affecting the sea bottom (Appleyard, 2015).

- **Chemical contaminants (+)** concentrations can be affected by leaking substances from dumped ammunition at least on the local scale, as the poisons in question are largely organic substances. Apart from that, the sympathetic effects of other pollutants, such as heavy metals, that are often associated with munitions (Gębka et al., 2016) and persistent organic pollutants (POPs), may further enhance the toxic effects of munition related contaminants (Czub et al., 2020).

- The management and treatment of dumped military material and unexploded ammunition may be an issue for **coastal management (?)** and maritime spatial planning, as allocating space for the construction of pipelines and other infrastructure needs to consider dumping site. Actions may be necessary to cope with the consequences of leaking ammunition, which is generally offshore in deep basins, but sometimes closer to the coast (Frey et al., 2020; Maser and Strehse, 2020).

### 5.17 Marine litter and microplastics

Plastic litter has been recognized as a problem in the oceans since the 1970s, but public, scientific, and political awareness has increased tremendously over the last decade. Data on concentrations of plastic particles in the Baltic Sea are rare, and we are just beginning to understand the sources, distribution and fate of this pollution.

Plastic litter is generally categorized as macro- (>25mm), meso- (5-25mm), and micro-(<5mm) litter, with the smallest size classes being the most abundant ones in the environment, but at the same time the most difficult to detect. Larger particles (>2mm) can be easily sampled and implemented in cost-effective monitoring, meeting the requirements of the Marine Strategy Framework Directive (Haseler et al., 2019). Sampling, processing and analysis of smaller microplastics require a more elaborate procedure (Enders et al., 2020).

Plastic contributes the largest share of human-generated litter entering the oceans from both land and offshore sources (Derraik, 2002). Land-based litter sources include municipal, commercial, industrial, agricultural, construction, and demolition activities (Barnes et al., 2009). Offshore sources encompass vessels or offshore platforms, lost containers from cargo shipping, fisheries, and marine aquaculture (Andrady, 2011; Derraik, 2002; Hinojosa and Thiel, 2009; Richardson et al., 2019).

In the Baltic Sea, litter dropped at beaches is a major source for larger micro- to macroplastics, including cigarette butts (Haseler et al., 2020). Regarding the smaller size fractions, municipal waste water was identified as substantial source for microplastics into the Baltic Sea (Baresel and Olshammar, 2019; Schernewski et al., 2021), especially stormwater runoffs including sewer overflow events, wastewater treatment plants (despite relatively good removal efficiencies), and untreated wastewater. Other potential sources for plastics into the Baltic Sea are marinas, agriculture, and industrial spills. Tire wear particles may form a considerable fraction in microparticle pollution in waters but there is hardly any information on concentrations and impacts (Wagner et al., 2018).

Data on concentrations of plastic particles in the Baltic Sea are scarce and highly variable due to challenging and not yet harmonized methodologies. Generally, the polymers detected most frequently are the ones produced in highest quantities, such as polyethylene and polypropylene. The beaches of the Baltic Sea are significantly polluted with plastic particles, with reported numbers ranging between less than 10 to over 1000 plastic particles per kg dry weight (Urban-Malinga et al., 2020). An extensive survey of 190 sandy beaches across the whole Baltic Sea area yielded 4921 plastic particles >2mm, mostly industrial pellets (19.8 %), non-identifiable plastic pieces 2–25 mm (17.3 %), and cigarette butts (15.3 %) (Haseler et al., 2020). The Warnow estuary in the southern Baltic Sea, as an example for non-beach sediment, showed microplastic abundances (>0.5mm)



ranging between 46 and 379 particles per kg dry weight, with concentrations decreasing towards the opening to the Baltic Sea (2 particles per kg). The abundance of plastic floating on the water surface appears comparable or lower than that in other world regions (Gewert et al., 2017; Tamminga et al., 2018; Rothäusler et al., 2019). Generally, distinct differences can be detected between areas with high versus low anthropogenic activity, with higher abundances of plastic particles and fibers close to major cities, freshwater discharges, and beaches (Zobkov et al., 2019; Gewert et al., 2017). Simulations based on emission data for the Baltic Sea region indicate a relatively short average residence time of about 14 days for polymers (0.02–0.5mm) in the water body, assuming beaches as a sink for microplastics (Schernewski et al., 2020).

Microplastic in fish varies across the Baltic Sea and with fish species. Particles were detected in 3.4% of demersal to 10.7% of pelagic fish in the North Sea and Southern Baltic Sea (Rummel et al., 2016), in 22% of Western Baltic herring (Ogonowski et al., 2019) and up to 1.8% in different northern Baltic Sea fish (Budimir et al., 2018). Long-term microplastics exposure on early life stages of sea trout showed no effects on hatching rate, larvae survival, or growth. Still, it generated nuclear abnormalities and chromosomal damage, indicating potential genotoxic effects (Jakubowska et al., 2020). Further data on ecotoxicological effects of microplastics on Baltic Sea biota are still rare. Methodological challenges exist, particularly for experimental studies targeting small microplastic fractions. In addition, environmental contaminants can mask microplastic-related effects.

Baltic Sea-wide investigations of microplastic-associated microbial biofilms and the potential of plastic degradation by Baltic Sea microorganisms indicate a low relevance of the interactions between microplastics and microorganisms. Environmental parameters, such as nutrient concentrations or salinity, strongly influence the composition of biofilm communities colonizing plastics. In contrast, the polymer properties of plastic themselves seem to affect these communities to a lesser extent (Kesy et al., 2019, Oberbeckmann et al., 2018). A specific enrichment of microplastics with potentially pathogenic bacteria, e.g. *Vibrio*, as compared to natural particles, does not occur in the Baltic Sea (Oberbeckmann and Labrenz, 2020). While some physiochemical properties of plastic beads changed significantly after exposure to bacterioplankton from the Baltic Sea (McGivney et al., 2020), the microbial degradation and metabolization of full plastic polymers is unlikely to occur in the Baltic environment at time scales relevant for human society (Oberbeckmann and Labrenz, 2020). Plastics often contain residual monomers, which are more likely to be degraded by microorganisms than polymers (Klaeger et al., 2019). This should be considered, in order to avoid the overestimation of plastic polymer degradation. Likewise, plastic additives or pollutants accumulating on plastic particles, such as polycyclic aromatic hydrocarbons (PAH), are more susceptible to bacterial degradation. In any case, published data on PAH accumulation on plastic and subsequent degradation are still missing for the Baltic Sea. Hence, microorganisms obviously cannot help to mitigate plastic pollution in the Baltic Sea, therefore the emissions of plastic particles need to be reduced.

In order to mitigate plastic pollution in the Baltic Sea, several measures are possible. For example, a reduction of cigarette butts at beaches may be prevented via environmental education, fines, or a smoking ban (Kataržytė et al., 2020). With regard to microplastics from municipal wastewater, a reduction of sewer overflows from currently 1.5% to 0.3% of the annual wastewater volume would notably lead to 50% less total emissions from wastewater-based sources into the Baltic Sea (Schernewski et al., 2021). Thus, both socio-economic and technological approaches need to be taken into account.

**Impacts of marine litter and microplastics on other factors**

- Marine litter has an impact on the **marine ecosystem (+)**, but concrete data are rare. Health consequences of microplastics in higher trophic levels like fish and birds and ultimately humans are unknown to date. Large plastic particles and items like abandoned nets and lines can result in lethal entaglements or be taken up as food items as they resemble prey organisms, which then may cause starvation. It is not known what the frequency of such events is in the Baltic Sea.



- The **fishing industry (+)** is affected by increasing public concern about microplastics. While microplastic uptake from other sources (e.g. plastic drinking bottles) is often neglected, the general concern is mainly focused on fish consumption. At the same time, the fishing industry is contributing to plastic pollution with lost fishing gear. There is little information on abandoned fishing gear in the Baltic Sea (e.g. Richardson et al., 2019) but it has been attributed as one of the largest sources of plastic in the Pacific (Lebreton et al., 2018). Ingestion of microplastics has been demonstrated for diverse **marine species** ranging from zooplankton to bivalves and fish (Ivar do Sul and Costa, 2014). Ecotoxicological effects are a matter of constant research, with methodological challenges especially in the small microplastic range. Studies on Baltic Sea biota indicate indifferent or minor to genotoxic effects, but this is still very uncertain (Oberbeckmann and Labrenz, 2020)**.** The same holds for open cage and extractive **aquaculture (?)**.

- As any naturally occurring particles in the water column, plastic litter particles can accumulate **chemical contaminants (?)** that ad- or absorb on the particle surface (Endo et al., 2013, Rochman et al. 2013). PAHs were found to accumulate on plastic particles in contrast to natural control particles (Oberbeckmann and Labrenz, 2020) but this depends both on the type of plastic studied and the chemical assessed. It has been discussed whether such contaminants can enter the food web via uptake of microplastics, but so far there is no sufficient evidence (Koelmans et al., 2016; Galloway et al., 2017).

- The presence of marine litter on beaches has an impact on the **tourism (+)** industry. Visible pollution can devaluate a touristic region and lead to a decrease in visitor numbers in the long term. Simultaneously, tourism was identified as a major pollution source on Baltic beaches (e.g. pieces from firework and cigarette butts) (Haseler et al., 2020; Schernewski et al., 2018)

- There is presumably an impact on **coastal management (?)**, as plastic litter should be included in existing strategies. Removing plastics from coastal areas has been shown to be more efficient than removing them from garbage patches in the ocean (Sherman and van Sebille, 2016). Plastic monitoring and beach cleanings are successful management tools to reduce plastic loads in the marine environment (Kataržytė et al., 2020, Haseler et al., 2020).

**5.18 Tourism**

The Baltic Sea region is an important destination for coastal and maritime tourism (Hall et al., 2009; Agarin et al., 2010). It is estimated that the region's tourism industry employs approximately 640,000 people, based on 88 million visiting tourists creating over 227 million registered overnight stays annually (Jacobsen, 2018). Geographically, the tourism industry in the Baltic Sea region involves the sea area, the coastal zone and the catchment area. Although the overall catchment area is important for tourism, the impacts of tourism and their relations with different human drivers are most concrete and visible in the coastal zone and the Baltic Sea area.

The cruise tourism sector constitutes a relatively small part of the shipping industry with approximately 5 % of total maritime traffic on the Baltic Sea (Polack, 2012) but it is growing fast. The share of cruise shipping is substantial in the fast growing international tourism (Więckowski and Cerić, 2016). As a result, the Port of Helsinki was in 2019 the busiest international passenger port in Europe with a total of 12.2 million passengers (Port of Helsinki, 2020). In general, the environmental impacts of cruises on the Baltic Sea are similar to shipping; they include the release of toxic materials from ship hulls, the release of black/grey water and air pollution (Jalkanen et al., 2020), but with a scale of several thousand passengers. In addition to direct impacts on the sea areas, cruise shipping creates significant environmental impacts on the coastal zone and especially in port environments and nearby urban structures. The key environmental issues in the ports and coastal areas relate to waste management, water and soil quality, noise and air emissions such as nitrous oxide ($NO_x$) and particulate matter (PM2.5) (Simonsen et al., 2019). According to the Organisation for Economic Co-operation and Development OECD report, Pallis, 2015), the key sustainability management targets for the handling of waste and garbage are the development of effective policies and practices (so called Port Reception Facilities). According to the report, it is estimated that a cruise ship with 3000



passengers (plus crew) produces 50 tons of solid waste in a week, which is considerably more than a regular ship of comparable size. Although the average emissions may not be significant, the local population may be highly exposed as cruise terminals

are typically close to city centers (Pallis, 2015).

Coastal and marine environments and their attractivity are essential for the tourism industry; so-called 'sun-sand-sea tourism' and nature-based tourism form two of the most important sub-sectors of global tourism (Nilsson and Gössling, 2013). In the Baltic Sea region, the coastal zone provides opportunities for a variety of tourist activities, which are concentrated on designated resort areas, spas but also urban centers along the coastal line (see Smith, 2015; Jacobsen, 2018). The main tourism

activities take place during the summer season and include sunbathing and beach activities, boating, fishing, yachting and second recreational homes, for example. Winter season activities are based on spas, skiing, ice-skating and ice-fishing (Hall et al., 2009). Some of these activities can be practiced sustainably, to a certain extent (Figure 6).

Climate change affects both summer and winter season activities. Shorter and unstable snow and ice conditions reduce outdoor activities in the winter season without necessarily providing alternatives. In contrast, summer activities and resources are

expected to benefit directly from a warming climate (Hamilton et al., 2005). Furthermore, there are already indications that, compared to the Mediterranean region, more temperate summer conditions may attract increasing numbers of coastal tourists to the Baltic Sea region, especially from Northern Europe (see Rutty and Scott, 2010; Grillakis et al., 2016). In conclusion, coastal and maritime tourism in the Baltic Sea region is expected to grow above the global average in the future.

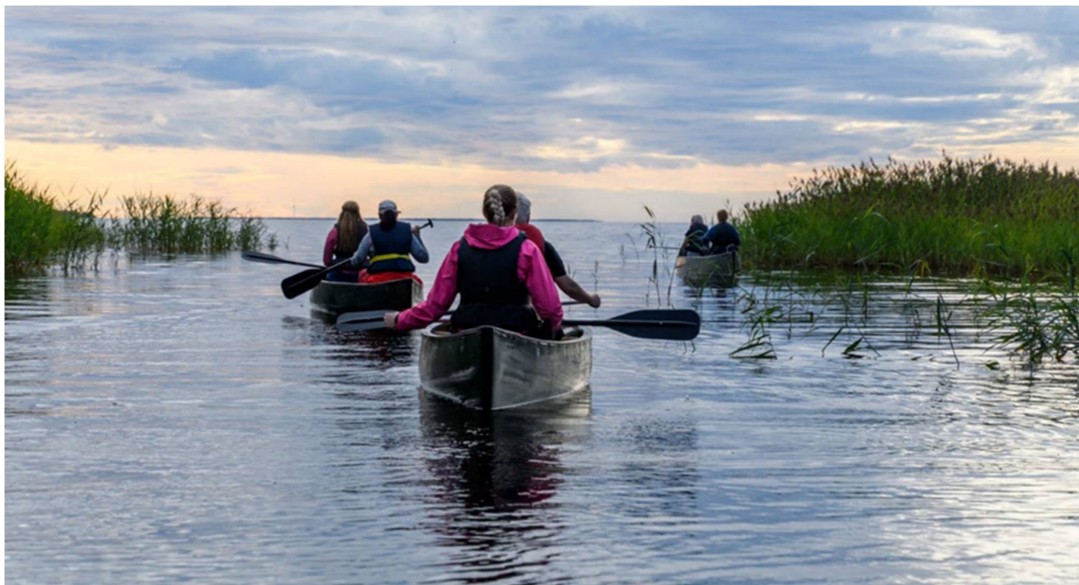


**Figure 6: Kayaking in coastal stretches may be a sustainable form of tourism (Photo: Visit Oulu, https://visitoulu.fi/en/kuvapankkikuvat/)**

However, potentially negative impacts of climate and environmental change may evoke an image problem for marketing. This

may contribute to a lower recreational value or impose real health risks, like a higher probability of extremes in weather conditions (e.g. heat waves or floods, Christensen et al, 2021), or more frequent and extensive mass blooms of blue-green algae (O'Neil et al., 2012; Hogfors et al., 2014). According to Schernewski et al. (2001), for example, the poor water quality "is the main obstacle for reaching the [tourism] development goals" in Pomerania, Germany. Indeed, algal blooms have had localized impacts on summer tourism in the Baltic Sea region, as the increasing periods of blooms have already resulted in

restrictions of coastal zone uses in many countries and resorts. Based on an extensive case study by Nilsson and Gössling



(2013), which covered mainly tourists from Sweden, but also from Denmark, Germany, Norway and Finland, a significant share of tourists who have experienced algal blooms have shortened or even canceled their holidays in certain coastal destinations. Moreover, the study indicated that algal blooms had an impact on the tourist´s willingness to return to the same destination where they had experienced blooms during previous visits. Thus, the impacts of increasing algae blooms in the
coastal zone can be significant.

**Impacts of tourism on other factors**

- Tourism is one of the largest economic sectors in the world. The related individual and organized land, sea and air-based mobility comes with the cost of fossil carbon emissions, contributing to global **climate change (+)** (Scott et al., 2012; Nilsson and Gössling, 2013; Terrenoire et al., 2019).
- Impacts on **marine ecosystems (?)** by tourism are not well studied in the region. However, cruise tourism produces similar underwater noise as shipping but on a much lower scale, as cruise tourism constitutes a very small part of the overall shipping industry on the Baltic Sea (Polack, 2012). It is unclear what kind of impact the current level of cruise
tourism noise has on marine life in the Baltic Sea (see Hawkins and Popper, 2017; Jalkanen et al., 2018).
- Tourism is a land use form having localized impacts on **land cover (+)**. Especially coastal resort areas are modified to fulfil tourist and recreational requirements, such as beach developments and golf courses (Kropinova, 2012; Cottrell and Raadik Cottrell, 2015). In addition, cruise tourism, ports and related transportation channels have an impact on urban structures (Pallis, 2015).
- Coastal zone and maritime tourism cause **nutrient loads (+)** that are mainly based on poorly regulated resort (accommodation and other services) wastewater management and cruise ships (Schernewski et al., 2001). Wilewska-Bien et al. (2016) estimate that the annually generated food waste onboard ships in the Baltic Sea contains about 182 tonnes of nitrogen and 34 tonnes of phosphorus.
- **Offshore wind farms (+)** may form aesthetically disturbing elements for visiting tourists (Veidemane and Nikodemus,
2015). As offshore wind farms, coastal tourism and yachting will grow, the probability of conflicts may increase in the future.
- **Shipping (+)** and cruise tourism are using partially the same port facility areas in some cases, which may have a positive impact on the development of Port Reception Facilities for the sustainable managing of waste (see Pallis, 2015).
- Especially coastal tourism activities create a substantial amount of traceable **marine litter (+)** (Lewin et al., 2020).
Based on previous studies, macro- and microplastics are the dominant types of waste, both in the coastal zone and marine area (Balčiūnas and Blažauskas, 2014; Haseler et al., 2019; Rothäusler et al., 2019). Marine litter has an impact on tourist perceptions and satisfaction (Lewin et al., 2020).
- Tourism is a significant factor for coastal use and change, causing both pressures and possibilities for **coastal management (+)**. Pressures are based on over-development and unregulated growth of tourism activities (Schernewski
and Sterr, 2002) but tourism may also have a symbiotic relationship with coastal management which provides safety and stability for environments of touristic activity (Haller et al., 2011; Weisner and Schernewski, 2013).

**5.19 Coastal management**

Coastal management (including marine spatial planning and marine protected area management) is an integrating factor that is used to regulate human activities in the coastal zone and thus has a strong impact on most other coastal, and many marine,
factors.

With respect to the physical environment, the term 'coastal management' is often used synonymously with the provision of coastal defenses against erosion and inundation (Pilkey and Cooper, 2014), concentrating specifically on hard engineering



(such as groynes, seawalls, revetments and offshore structures), soft engineering (renourishment, beach dewatering, dune stabilization) and planning (managed retreat, limiting development, reclamation). When considering factors of change in the

physical coastal environment, we normally think of waves (of different types), water levels, coastal currents and sediment availability and their relationship to erosion or accretion. We rarely think of the plans and decisions made by environmental managers and decision makers *per se* being a driver of coastal change. We generally expect that such plans and decisions are made based on the best scientific knowledge and advice, but management decisions can also be based on political expediency or a range of socio-economic necessities or factors. These decisions can, in themlves, result in further environmental change.

This argument can be made with respect to both the physical and biological environments.  Sometimes, management decisions can have a direct effect, such as when an approval is given for the construction of port facilities (Pupienis et al., 2013; Žilinskas et al., 2020) or a seawall, or the effect may be more indirect, through mechanisms of land use and maritime spatial planning (Zaucha, 2014), fisheries allocations (Reusch et al., 2018), or river regulation affecting sediment supply.

Ideally, coastal management should be a reasoned, achievable and sustainable long-term response to coastal use and change

that protects the environment and provides for the use and enjoyment of the coast by people. It should be forward looking, identifying how future human activities will interact with natural factors (wind, waves, currents, water levels etc.) and processes (sediment transport, erosion, deposition etc.) by providing a framework to assess, mitigate and minimize adverse impacts while promoting positive changes. However, very frequently the actions resulting from management attempts to use the coast become factors in their own right, resulting in further, often detrimental, changes. The implementation of plans for

coastal erosion 'protection', sea defenses, public infrastructure (e.g. ports), coastal development areas, and public space and amenity creation, can change the physical factors (waves, currents etc.) and sediment transport, resulting in new morphodynamic equilibrium conditions that may be unwanted and unpredictable. Many such situations in the Baltic Sea are described in detail in Pranzini and Williams (2013).

Many attempts to reactively mitigate coastal problems, undertaken with the best intentions but with limited understanding of

processes, have resulted in negative impacts. Specific examples include the use of groynes that create downdrift effects, and seawalls that result in upper beach loss. It has long been known that manipulation of one part of a system can cause effects in other parts of the system, often is unexpected ways. This 'Law of Unintended Consequences', first formalized by Locke in 1691 (Mottershead et al., 2016) has meant that many coastal management actions have resulted in unanticipated outcomes, some of which have been beneficial, but the majority of which have made the problem worse, or have created new problems.

Negative unintended consequences are most frequently caused by ignorance, error, immediacy (e.g. to protect human life), or basic values (Merton, 1936) (e.g. private property rights, freedom of navigation, rights to use resources, sovereign rights). Thus, coastal management relies on the scientific basis of the best available knowledge for the best available response, but also depends on social, economic and political constraints, in balancing claims, which are hardly compatible.

A good example of coastal management conflicting with basic values is the case of hard protection (such as a seawall)

constructed to protect private property from coastal erosion. It is well understood that building a seawall which protects property effectively degrades the beach (passive erosion; Griggs, 2005) when the system is in sediment deficit, causing a loss of public amenity. This results in a conflict between private property rights and public or environmental rights. Until such issues are resolved, managed retreat (Cooper, 2003) as a management response to coastal erosion is difficult or almost impossible.

The tools available now make the assessment of coastal projects much more reliable (e.g. Bagdanavičiūtė et al., 2019). Modelling tools such as the MIKE suite (DHI) and DELFT 3D suite can effectively alert managers to cases where management actions can result in consequences that must be considered further. For every coastal project, a simple sediment budget (Figure 7), applied within an appropriately sized coastal compartment (for example a bayhead confined beach) should be applied. Where a project changes any of the sediment transport pathways over the active beach profile, unintended consequences may

result and need to be addressed. Similarly, if a project is expected to change the fluid motions, the coastal morphology or





natural sediment transport, further investigations are required. In most parts of the Baltic Sea, due to its small size and limited fetch, waves are generally of short period and length, with sediment transport being largely confined to quite shallow waters. Therefore, even small-scale projects, such as small boat harbours, can have significant coastal impacts.

## SEDIMENT BUDGET

**Figure 7.** The coastal sediment budget considered within an appropriately sized coastal compartment. Sediment transfers beyond the profile closure depth are in grey (Kevin Parnell).

Coastal management actions can become factors of coastal change. As the understanding of coastal processes improves, negative consequences of actions should become less common, and the application of simple conceptual models, along with sophisticated tools that are now available should result in fewer mistakes being made. A bigger challenge, however, is the resolution of the conflicts between best practice and long-held societal values and practices. Coastal management must be undertaken with specific consideration of climate change, particularly sea level rise. Vitousek et al. (2017) pose the question: Can beaches survive climate change? They conclude that "the future of the coastline will be what we engineer it to be", thereby putting forward the view that coastal management actions may be the most significant coastal driver in the future.

**Impacts of coastal management on other factors**

- Coastal management decisions have a substantial impact on **coastal processes (+)** through the control of sediment translocations, a reduction or enhancement of erosion, and through coastal constructions like groynes, levees and other coastal and offshore constructions. A current and emerging coastal management factor influencing shorelines is the management of sand extraction, already a problem in the southern Baltic Sea (e.g. Uścinowicz et al., 2014).
- **Submarine groundwater discharge (?)** may be impacted by coastal management decisions or infrastructure which could have an impact on the coastal groundwater level and the conditions and obstruction of groundwater seeps.
- Coastal management can be expected to have an impact on **marine ecosystems (?)** as it affects the land-sea linkages which are important for the terrestrial and riverine loads of nutrients and pollutants.
- Possibly, **non-indigenous species (?)** could profit or suffer from coastal management decisions, through the management or protection of certain habitats, or the deterioration of those habitats, by coastal constructions.
- Coastal management decisions impact directly on **land use (+)**, **agriculture(+)** and **aquaculture (+)** in coastal areas.





- • Coastal management can have a considerable impact on **fisheries (+)** and fish stocks by regulating fishing grounds and deteriorating coastal habitats of fish species (Kraufvelin et al., 2018)

- • There may be a connection between coastal management on **river regulations (?)** at least in the estuaries (e.g. Zedler, 2017)

- • Coastal management, through maritime and coastal planning, regulate the allocation of space for **offshore wind farms (+)** (e.g. Chaouachi et al., 2017; Sobotka et al., 2021).

- • There is a considerable impact of coastal management on **shipping (+)** through the construction of ports, wind farms, allocation of fairways, regulations, and marine protected areas from which shipping is banned. Areas are regulated by local authorities considering the protected areas defined e.g in the Birds and Habitat directives (92/43/EEC and 2009/147/EC). (e.g. Andersen et al., 2020)

- • **Dumped military material (+)** is in some cases located in the coastal zone, although it concerns rather terrestrial dumpsites or solitary munitions resulting from military activities. In such cases, munition may be disturbed during the construction of coastal defences. In Germany, a special programme has been initiated to locate and remediate munitions in the coastal zone (BLANO, https://www.schleswig-holstein.de/DE/UXO/uxo_node.html).

- • Coastal management regulations are used in the management of **marine litter (?)**.

- • Coastal management has a strong impact on **tourism (+)**, as the tourist industry is a major stakeholder in the competition for space in the coastal area.

## 6 Key messages and knowledge gaps

A clear outcome of our analysis is that all sectors are strongly affected by human activities. Here we summarize the key messages and the knowledge gaps in our analysis. For references, see the previous section.

### 6.1 Climate change

**Key messages**

Climate change is the overarching, integrating factor affecting almost all of the other factors described here, but it is not necessarily the strongest or most detrimental factor. The warming and shift of seasons are by far the most evident signs of climate change in the region and will likely be so in a long time. It has direct or indirect consequences for most of the areas 2475 discussed here. The associated changes in the hydrological cycle, i.e. changed precipitation, evaporation and runoff patterns are likely to affect land use (agriculture) and possibly nutrient loads in the southern part of the Baltic Sea region. Sea level rise, a direct consequence of climate change, threatens not only human settlements and infrastructure in the southern part but also coastal ecosystems and may have an influence on the future salinity, stratification and biogechemical status of the Baltic Sea. It affects coastal processes, generally increasing erosion on soft coasts through higher run-ups under storm conditions and 2480 changed angles of wave attack and reduced protective ice cover. It will presumably enhance the problem of hypoxia and anoxia with related consequences for the biogeochemistry and extended cyanobacteria blooms. More $CO_2$ in the air means more dissolved $CO_2$ in seawater, which results in a pH decrease (ocean acidification). However, the consequences in the Baltic Sea's different basins are unclear as there is a strong alkalinity impact, which has a buffering effect. Basins with low alkalinities will become more sensitive to marine acidification. There is also a strong impact of the climate on land cover and land use as 2485 precipitation patterns and temperature are strong decisive factors. The same holds for aquaculture, for which water temperature is the strongest factor. Fisheries are strongly affected, as commercially important species are sensitive to climatic impacts on the food web and there are cascading effects within the food web. The development of wind farms in the Baltic Sea is highly dependent on climate change, as wind is the resource for energy production. There is a socio-economic pressure to replace fossil fuel energy production, with wind power being a primary source of renewable energy. Climate change affects shipping,



as the conditions at sea are all weather-, hence climate sensitive. In the north, shipping may profit from a much reduced ice cover in winter, and shallow navigation routes will slowly get deeper and safer to travel though rising sea levels in the south. Climate change may affect the distribution and pathways of organic contaminants, but there are various unknowns concerning this potentially important linkage. Unexploded and dumped warfare agents may be subject to climate change related enhanced corrosion rates of shells and release rates of toxic substances, but there is a high uncertainty. Climate change might affect

microplastic concentrations in the sea through changed precipitation and runoff patterns. Tourism is a climate-dependent sector, making an expansion in the Baltic Sea probable with rising temperatures, despite potentially harmful implications. Coastal management as the integrating activity responding to challenges posed by human activities in the coastal zone, is much affected by climate change through the climate-related dangers (sea level rise, storm surges, rain floods) for human settlements and infrastructure, but also coastal ecosystems.

There are possible feedback mechanisms of some regional factors affecting climate. Land cover affects the albedo, i.e. the reflectivity of the surface, which has conseqences on the energy absorbed or reflected. Reforestation efforts to enhance the $CO_2$ uptake by vegetation to mitigate climate change may thus be compromised, as forests are dark surfaces, weakening the albedo and enhancing the heating effect. As reforestation also influences evapotranspiration, the total response is even more complex. As a result, reforestation in the Baltic Sea region could lead not just to an overall cooling but also to a reduced

amplitude between maximum summertime and minimum wintertime temperatures (Strandberg and Kjellström, 2019). There is currently no consensus whether the albedo effect of reforestation renders these efforts futile or not. A reduced aerosol concentration (not treated in this paper) relative to decades ago may contribute to the perceived warming (Barkhordarian et al. 2016), but there is so far little evidence to confirm this hypothesis. Large offshore wind turbines may affect the local climate, but the effect beyond the very local scale is unclear. Shipping and tourism, associated with extensive mobility and travel, have

a feedback on the climate by their fossil fuel carbon emissions.

**Knowledge gaps**

The large spread between different atmospheric simulations reflects the combined uncertainty between global climate sensitivity, regional response and natural variability. For assessing future climate change, modelling is the primary tool

available, and fundamental research questions remain. These include a better representation of the large-scale thermohaline circulation influencing the North Atlantic, the large-scale atmospheric circulation including storm-tracks influencing cyclone activity and high-pressure blocking situations, representation of microphysical processes involving clouds and aerosols, exchange processes at the surface including soil moisture and snow conditions, and a better representation of precipitation processes including convection-permitting modeling to better represent precipitation extremes.

Simulations with large ensembles of climate models where the only difference is the initial conditions show that such changes can be very large and this is a considerable source of uncertainty in assessing changes in the regional climate. A source of considerable uncertainty for sea level estimations is that the projections of precipitation change provided by climate models are regionally not robust. Climate models do generally not agree on the line dividing precipitation increases and decreases. Evaporation, on the other hand, will very certainly increase. As a consequence, there is an uncertainty to what extent conditions

for drought will be more or less pronounced in the future and the extent to which they may influence areas in the north. Furthermore, it is so far not clear whether salinity in the Baltic Sea will increase or decrease in the future, and thus how the modified salinities may affect sea-level changes. Further question are:

- *Climate change and river regulations:* What is the connection between climate change and river regulations? Is climate change affecting how rivers are regulated, and how, e.g. through increasing inundations, floodings, etc? How

does climate change influence the decisions to restore regulated rivers?

- *Climate change and dumped military material:* What is the effect of warming temperatures on the corrosion rate and release of toxic substances from dumped materials?



- *Climate change and marine litter:* Hardly anything is known about plastic degradation rates (not to be confused with fragmentation into smaller plastics), even less so for the marine system.

**6.2 Coastal processes**

**Key messages**

Coastal processes strongly affect coastal infrastructure by sediment relocations, erosion and accretion processes. They may contribute to coastal land loss through enhanced erosion. A re-distribution of coastal currents may lead to sediment relocations, which in turn may lead to problems for coastal infrastructure (e.g. harbours) and shipping routes. Contaminants may be distributed along the coast and carried to locations where they may have a detrimental effect. The same holds for toxic substances released from dumped and unexploded warfare agents. Submarine seawater discharge can be defined as a coastal process and will also be affected by groundwater levels and consequently by precipitation patterns. Coastal processes can be affected by the climate, river regulations (at least in estuarine areas), some wind farms which may affect coastal current systems, shipping through swell damaging fragile coastlines and embankments, all of which is a subject for coastal management.

**Knowledge gaps**

There are today major gaps in the understanding of the functioning of sedimentary compartments and the wave-driven mobility of sediment between these cells in the eastern Baltic Sea. A large gap is the scarcity and low accessibility of data about changes to the coastline. They are crucial for the validation of modeled sediment fluxes and for understanding and forecasting coastline changes. Modelling with much finer resolution (about 500 m alongshore) is necessary to properly identify the structural features of sediment transport over long interconnected sedimentary systems. To resolve both rapid and slow phases of coastal evolution it is necessary to combine high-resolution scanning techniques with detailed bathymetric data and to develop and validate methods for approximate estimates of underwater sediment transport. Further questions are:

- *Coastal processes and hypoxia:* What effects do coastal processes, i.e. currents, erosion processes and sediment translocations have on coastal hypoxia?
- *Coastal processes and acidification:* What effects do coastal processes have on acidification? What is the impact of coastal erosion on alkalinity and acidification? It is assumed that weathering in the northern basins and rivers may contribute to an increase in alkalinity. What contribution may coastal processes have on alkalinity, acidification and the carbon system in general?
- *Coastal processes and marine ecosystems:* To which extent and how are coastal marine ecosystems impacted by coastal processes?
- *Coastal processes and fisheries:* How do coastal processes have an impact on coastal fisheries, e.g. gill nets?
- *Coastal processes and chemical contaminants:* How do coastal processes affect the release and distribution of chemical contaminants from rivers and sediments?
- *Coastal processes and Dumped Military Material:* Do coastal processes affect dumped military material, e.g. by current systems or sediment transports? Are there any effects expected at the dumpsites and how strong could they be?

**6.3 Hypoxia**

**Key messages**

Hypoxia (and anoxia) in the deep basins and in coastal areas is a consequence of enhanced organic matter production (eutrophication) and its respiration. Hence, it is affected by land use and agriculture, as this is the primary source of excess nutrients. In turn, these oxygen deficiency areas are subject to strong abatement measures (e.g. HELCOM Baltic Sea Action



Plan), and hence may affect agricultural procedures. Climate affects hypoxia by a strengthened thermohaline stratification, higher respiration and turnover rates in warmer waters, and possibly also by sea level change which may allow stronger saltwater inflows in the future, potentially further strengthening stratification. Furthermore, hypoxia has an impact on marine ecosystems and fisheries. Chemical contaminants bound in the anoxic sediments may be released by bioturbation at times of reoxygenation.

**Knowledge gaps**

A realistic estimation of the anoxic and hypoxic areas in the deep basins and coastal zones is very difficult due to the lack of a highly resolved measurement grid. The current estimations are based on sporadic measurements, extrapolations and modeling, and may differ by up to 20%. It is insufficiently known which processes govern the observed frequencies of major saltwater inflows, and the mixing with stagnant deep waters. Also, there is no sufficient quantification of sources and sinks in the oxgen budgets. Further questions are:

- *Hypoxia – Acidification:* The connection between anoxia and acidification or alkalinity are strong and should be considered in acidification studies. Is anaerobic alkalinity generation in coastal sediments an essential process in the Baltic Sea?
- *Hypoxia – Aquaculture:* Does coastal hypoxia have any impact on coastal open cage or extractive aquaculture sites?
- *Hypoxia – Dumped Miltary Material:* Does hypoxia or anoxia at the dumpsites have any impact on the corrosion rates of the hulls and shells, and what could be potential chemical reactions of leaked substances in the oxygen-free biogeochemical environment?

**6.4 Acidification**

**Key messages**

Acidification is a consequence of enhanced $CO_2$ concentrations in the environment. It has been shown in oceans around the world and its effects on marine biota have been described in many investigations. There are considerable differences in pH variability and trends between the different basins, and alkalinity, which is a measure of buffer capacity of seawater influenced by other chemical traits in the tributary basins, is an essential but poorly understood factor. Also, the magnitude of detrimental effects of acidification on organisms in the Baltic Sea is unclear. There are few and partly contradicting reports e.g. on otolith growth in fish larvae.

Acidification of waters and soils due to the deposition of sulphur and nitrogen oxides from fossil fuel combustion ("acid rain") was debated many years ago. Terrestrial emissions have decreased considerably so that $CO_2$ generated acidification is higher by a factor of ten. Shipping contributes to acidification in coastal waters and along shipping lanes. In the effort to clean air emissions to fulfill regulations, the combustion air is technically stripped of sulphur and nitrogen oxides by scrubbers and concentrated in sea water, which is subsequently released to the sea. So the problem is effectively transferred from the air to the water. Although the contribution of this process to acidification is expected to be rather low in general, shipping contributions are increasing and may aggravate the problem.

**Knowledge gaps**

There are many unknowns in the Baltic Sea, and it is not clear whether acidification in the Baltic Sea is a threat for the time being, or not. A major knowledge gap is the uncertainty related to the sources and regional distributions of alkalinity, which largely determines the strength of acidification in the Baltic Sea. What is responsible for the observed trends in alkalinity in the different basins? What impact does acidification of freshwaters and soils through deposition of nitrogen and sulphur oxides have on coastal acidification? Further questions are:



- *Acidification – Marine Ecosystems:* How are organisms and the ecosystem functions affected by acidification?
- *Acidification – Aquaculture:* Which impact may acidification potentially have on open cage or extractive aquaculture?
- *Acidification – Fisheries:* Which impact may acidification potentially have on fisheries? There is evidence that otoliths may be affected by lower pHs in the surrounding waters, but observed effects on larval growth and survival

rates are scarce and contrasting.

- *Acidification – Chemical contamination:* Are there any impacts of acidification on chemical contaminants? Does the pH in the foreseeable changes affect the chemical speciation of organic substances and the reactivity or potential toxicity of these compounds?

### 6.5 Submarine Groundwater Discharges

**Key messages**

Submarine groundwater discharges may contribute to local inputs of nurients, organic and inorganic carbon, organic contaminants as well as trace metals, but its relevance for coastal waters of the Baltic Sea on a larger scale is unclear.  As climate change affects factors like precipitation, topography-driven flow, wave set-up, sea level rise and convection caused by salinity and temperature between the seawater and groundwater, it can be expected to affect submarine groundwater discharges.

Agriculture affects groundwater constituents to a large extent and thus contributes to submarine groundwater releases to the coastal Baltic Sea. SGDs may be a topic of concern for Coastal Management.

**Research gaps**

There are currently only very few measurements in selected areas, so a projection of the significance of submarine groundwater

discharges to the Baltic Sea as a whole is very difficult. A dedicated groundwater-monitoring network combined with a coupled groundwater-surface model simulating the transport, adsorption and mixing processes would help understand and evaluate the role of submarine groundwater discharges in chemical substances cycling in the coastal Baltic Sea. Further questions are:

- *Submarine Groundwater Discharges – Hypoxia:* Do Submarine Groundwater Discharges have any impact on the generation of coastal hypoxia, e.g. though local nutrient inputs?

- *Submarine Groundwater Discharges – Acidification:* Do Submarine Groundwater Discharges have any impact on coastal acidification of seawater?
- *Submarine Groundwater Discharges – Marine (coastal) ecosystems:* What are the effects of Submarine Groundwater Discharges on coastal ecosystems?

### 6.6 Marine ecosystems

**Key messages**

Marine ecosystems are very much affected by the various anthropogenic impacts, ranging from eutrophication, climate change, chemical contaminants to marine litter and fisheries. Excessive nutrient inputs by agriculture results in elevated phytoplankton biomass and consequently also of higher trophic levels up to harvestable fish. A large fraction of this biomass is respired and causes oxygen deficiencies in deep waters and coastal zones, leading to modifications in parts of the ecosystems, e.g. extensive

nitrogen fixing cyanobacteria blooms. Climate change affects different parts of the ecosystem in different ways, and provokes various complex interactions. Sub-ethal effects of chemical contaminants in ecosystems are very difficult to describe, so that large uncertainties exist.

**Knowledge gaps**

There are large uncertainties concerning the impact of climate change on the marine ecosystems of the Baltic Sea. This refers to temperature and acidification effects on various trophic levels, starting from the microbial communities and the microbial



loop to higher levels. It is uncertain how salinity may change in the future, and likewise are the potential consequences for pelagic and benthic communities. The impacts of shipping on ecosystems through pollution and underwater noise are also not well understood. Further questions are:

- *Marine ecosystems – Acidification:* It is known that primary production affects acidification, but do other ecosystem functions also have an effect?
- *Marine ecosystems – Aquaculture:* What are the interrelations of marine ecosystems with open cage or extractive aquaculture; the latter can be described as the exploited part of the natural ecosystem.
- *Marine ecosystems – Chemical contaminants:* What roles do aquatic ecosystems play in the transfer and
transformation of organic constituents between trophic levels up to organisms consumed by humans like mussels, fish or algae?

### 6.7 Non-indigenous species

**Key messages**

Non-indigenous species have the potential to disturb marine and coastal ecosystems as well as fisheries, by impairing
established food webs and displace established species. Shipping was identified as the primary pathway for new species in the Baltic Sea. Shipping regulations may be affected in a response to the danger of introducing species on ship hulls and ballast waters, with related economic consequences for the shipping industry. Species escaping from aquaculture may also be a factor for the introduction of non-indigenous species. Climate change may also be a factor due to facilitating north- and eastward migrations of individual species. It was shown that the non-indigenous benthic species *Marezellaria sp.* may increase the
release of organic substances to the overlying sea water by bioturbation (together with indigenous bioturbators).

**Knowledge gaps**

The number of non-indigenous species in the Baltic Sea is an estimation based on monitoring, for which there is not a common strategy in all Baltic Sea basins. The impacts of single or multiple non-indigenous species on the ecosystem, food webs and
food production (e.g. though the introduction of toxic algae) are complex and hard to distinguish from other factors. Further questions are:
- *Non-indigenous species – Coastal management:* What is the coastal management strategy to cope with non-indigenous species and their potential impacts?

### 6.8 Land use and land cover

**Key messages**

Land use and land cover is a main factor in the Earth system of the Baltic Sea region, with multiple impacts on other factors. Due to the drainage of terrestrial organic matter to the aquatic system, land cover has a big influence on the Baltic Sea. Furthermore, it may have an effect on the climate through albedo and latent heat flux. Forests and water surfaces have a low albedo (i.e. low reflectivity of solar energy) and may thus contribute to warming more than they can contribute to cooling by
$CO_2$ assimilation, but it is not clear which effect is stronger. Crop farming and industrial forestry with all their impacts (e.g. release of nutrients and contaminants, hypoxia, acidification, SGDs) are the dominant types of land use in the Baltic Sea region. Aquaculture in land-based factories may use much space and energy. River regulations are a type of land use change and may also indirectly affect fisheries by affecting the migratory behaviour and spawning success of salmons and other migrating fish. The increasing scarcity of available land spaces for wind farms may accelerate the development of offshore wind farms.
Tourism is a strong factor for coastal land use and is thus a topic for coastal management.

**Knowledge gaps**



There are major knowledge gaps concerning the speed and direction of terrestrial land cover change, and what the effects on atmospheric $CO_2$ concentration and dissolved organic matter transport could be. Cross-system effects and feedbacks may impede afforestation efforts to enhance $CO_2$ drawdown. There is still insufficient knowledge of the possible environmental, system and cross-system impacts and feedbacks to facilitate continental scale decisions on large scale land cover changes. Terrestrial land cover is a slowly changing system with long-term implications, so both short and long-term effects need to be evaluated. Further questions are:

- *Land cover – Marine ecosystems:* What are the impacts of land cover and land use on marine ecosystems? They are spatially separated but may be connected through various land-sea interlinkages; which could they be and how do they interact?
- *Land cover – Offshore Wind Farms:* What is the relationship between wind power generation on land, where there is an intense competition between different types of use and conflicts in regions where people live, and the designation of new offshore wind parks?

## 6.9 Agriculture and nutrient loads

**Key messages**

Agriculture is the dominant land use type in the southern Baltic Sea region, and is responsible for the bulk of nutrient release into the Baltic Sea and hence for eutrophication and hypoxia. Although nutrient releases have been reduced considerably in the past two decades, legacy nutrients in sediments (most notably phosphorus) still affect the Baltic Sea biogeochemistry and ecosystem, e.g. through hypoxia and cyanobacterial blooms. Agriculture also contributes to the distribution of organic and other contaminants to the soils, groundwater and eventually to the coastal sea, also by submarine groundwater discharges. Agriculture strongly also affects the maritime food production sectors fisheries and aquaculture through the provisioning of excess nutrients resulting in increased yields since the onset of eutrophication. Coastal management, in its capacity to manage the competing interests in the coastal zone, also needs to be concerned with agriculture and its effects.

**Knowledge gaps**

The effects of climate change on nutrient loads are highly uncertain. Nutrient loads depend on runoff and riverine discharge. Climate models indicate the north to become wetter and the south to be drier, or at least not get wetter, so that the agricultural south may experience a reduced runoff, hence reduced nutrient loads to the sea. Currently, there is no consistent, catchment-wide model of nutrient source apportionment, so it is difficult to assess where which sources predominate in different sea sub-basins. Furthermore, it is unknown how fertilization practices, crops grown, and land use will change in response to climate change. Also unknown is the relative contribution of nitrogen from accumulated, legacy sources to current riverine loads to the sea, and how the accumulation and release of legacy nutrients will be impacted by climate change. Nutrient loads are strongly influenced by runoff and discharge, so the eutrophication status of the sea is strongly influenced by riverine nutrient loads. Diffuse losses of nutrients from agriculture are one of the core drivers of nutrient loads to the Baltic Sea. Further questions are:

- *Agriculture – Coastal management:* What is the relation between agriculture and coastal management? Is coastal agriculture of any relevance for coastal management and how does it look like?

## 6.10 Aquaculture

**Key messages**

Aquaculture is a coastal or land-based branch of food production and thus affects the environment through the release of nutrients and contaminants, with its related consequences for ecosystems like eutrophication and hypoxia. Land-based aquaculture factories use considerable space and energy. It has been shown that cultured non-indigenous species can escape





from their cages, with unknown consequences in the Baltic Sea. A potential synergy between the food and energy producing

sectors is the installation of cultivation infrastructure in the wind farms where it is protected from shipping and fishing and where there are fewer problems with water quality in open waters. However, a successful implementation has not been proven yet. As a coastal activity, aquaculture must be on the coastal management plan.

**Knowledge gaps**

A climate-related risk that is poorly understood is the changed microbial biota in fish guts as water temperatures rise. Changes in salinity might also affect the biota, with possible physiological consequences for the fish. A further uncertainty is the impact of parasites in future aquaculture, and the impacts of pharmaceuticals on the environment. Further questions are:

- *Aquaculture – Fisheries:* What is the economic connection between aquaculture and fisheries? Also, there may be a competition for coastal spaces.

- *Aquaculture – Chemical contaminants:* What is the impact of pharmaceuticals to fight fish diseases and parasites on the environment? What is the impact on the environment and other organisms?

- *Aquaculture – Marine litter:* There are considerable knowledge gaps concerning aquaculture as a source for marine litter of any kind.

### 6.11 Fisheries

**Key messages**

Fisheries have a strong impact on ecosystems by extracting fish biomass, and the effects may cascade through to lower trophic levels and even to nutrient concentrations. Fisheries have a considerable impact on litter and ghost nets in the oceans, but the share in the Baltic Sea is unknown and may be much smaller than in other parts of the oceans. There is a particular danger for fishery related to unexploded and dumped military material, and fishermen (like beach strollers) have been injured and killed.

There is a competition between fishing and other coastal activities like shipping and wind farms as they compete for space, which is also a subject for coastal management.

**Knowledge gaps**

A major challenge is to quantify fishing impacts on ecosystems relative to those of other human or ecosystem factors. An

example here is cod in the Baltic Sea, where fishing for its prey species potentially influences cod growth and condition. However, as a number of other factors influence cod growth and condition at the same time (e.g., oxygen conditions, parasites from grey seals), possible effects of fisheries management actions are difficult to determine. Furthermore, little is known about fishing impacts on the seabed and the sensitivity of different organisms and communities to fishing gear disturbances. Further research and data to parameterize models are needed, as well as establishing better quantitative links to other factors (e.g.

anoxia). Monitoring systems with respect to other ecosystem components or factors need to be implemented. Further questions are:

- *Fisheries – Non-indigenous species.* What is the impact of fisheries on the success and distribution of non-indigenous species? Some may be a commercial alternative to indigenous species (e.g. Round Goby), but how do fisheries respond to non-indigenous species?

- *Fisheries – agriculture.* What is the influence of fisheries on agriculture, i.e. what is the potential feedback on terrestrial food production?

- *Fisheries – Aquaculture.* What is the connection between fisheries and aquaculture (economically though markets or directly)?

- *Fisheries – Tourism.* Is there any impact of fisheries on tourism, in addition to fishermen switching from fishing to

organizing touristic fishing tours in fishing and touristic hotspots, or selling market fish directly from fishing vessels?



### 6.12 River regulations

**Key messages**

River regulations affect the coastal areas though the modification of runoff, sediment and nutrient amounts and ratios, hence potentially changing the biogeochemistry of coastal waters. In changing the biogeochemistry, they also potentially affect

alkalinity and acidification. They also affect fisheries to a certain extent as anadromous fish migrate upstream for reproduction, and artificial fish passes may not be as effective as desired.

**Knowledge gaps**

Concerning stream restoration, there is a need for more extensive observations on the water quality effects of stream restoration

actions and a comparison of the effectiveness of such efforts compared to other land-based remediation plans. Furthermore, there is a need to be able to project the effects of remediation actions on nitrogen removal and other water quality objectives, especially regarding feedbacks between hydrological changes and biogeochemistry of the hyporheic zone under different conditions. Further questions are:

- *River regulations – Hypoxia.* Are river regulations in any way correlated with coastal and deep-water hypoxia?

- *River regulations – Submarine Groundwater Discharges.* Are river regulations in any way correlated with submarine groundwater discharges, or do they affect groundwater in coastal regions in any way?

- *River regulations – Marine Ecosystems.* What impact do river regulations have on marine (coastal) ecosystems?

- *River regulations – Aquaculture.* To what extent is coastal aquaculture affected by river regulations?

- *River regulations – Chemical contaminants.* Is there a connection between river regulations and the release or

transport of chemical contaminants?

- *River regulations – Marine litter.* Is there a connection between river regulations and the release or transport of marine litter?

### 6.13 Offshore wind farms

**Key messages**

Offshore wind farms have expanded in the recent past and are expected to expand further in the future as carbon neutral energy production methods are politically requested in the effort to reduce carbon emissions and mitigate climate change. Climate change has an impact on atmospheric circulation but a general increase in wind speed is not expected so an increase in energy yield can only be achieved through larger and more wind farms. However, there is a certain impact by offshore wind farms on the local climate, which may be considerable in the future as more and more wind farms are constructed. Also, the efficiency

of large wind farms may be compromised locally through decreasing wind speeds in the farms themselves.

Evidence from North Sea wind farms indicates an impact on the local biogeochemistry and ecosystems as they have an impact on downstream stratification of the downstream water column. Noise during piling can drive marine mammals (harbor porpoises) away from the arms, but may attract them during normal operation. Losses of birds due to collisions with blades is minor for single turbines and small wind farm, but large arrays may affect migration routes of birds. There is a possible synergy

between wind farms and aquaculture but plans have not been implemented as of now. Fisheries are banned from wind farms, so there is a competition for space, but wind farms may act as protected spawning grounds, thus indirectly having a positive effect on fisheries. Also for shipping, there is a competition for maritime space. Offshore wind farms may have a negative impact on coastal touristic resorts as they obstruct a free view of the sea. Wind farm planning and operation can be considered as a part of coastal management.


**Knowledge gaps**



The impact of wind farms on the regional climate has not been assessed in the Baltic Sea. With increasingly extensive wind parks, the impacts on the regional weather and climate as well as on currents, stratification and marine ecosystems (incl. birds) need to be further investigated. It will be important to project the future contribution of offshore wind farms to energy

production in the Baltic Sea region and Europe in relation to land-based structures and other producers of regenerative energy. Further questions are:

- *Offshore Windfarms – Chemical contaminants:* Do offshore wind farms release any organic contaminants to the water?

- *Offshore Windfarms – Dumped military material:* Is there a connection between offshore wind farm sites and
dumpsites of ammunition?

- *Offshore Windfarms – Marine litter*: Do offshore wind farms release considerable amounts of litter to the water?

### 6.14 Shipping

**Key messages**

Shipping is a considerable source of greenhouse gases and other harmful airborne emissions like sulphur and nitrous oxides.
Scrubbers used to clean exhaust air result in cleaner air but add to the acidification of the waters. Shipping itself is also much dependent on the climate as shipping routes in the North are subject to closure in severe ice winters, which is expected to get less frequent. Shipping is much dependent on weather conditions at sea, which is related to the climate. Shipping may affect coastal processes like erosion through the impact of ship-induced waves, and shipping routes may be affected by sedimentation and currents. Ecosystems may be affected by nutrient releases, pollution (e.g. antifouling agents) and underwater noise.
Shipping is the primary carrier of non-indigenous species to new ecosystems, through hull attachments and ballast water. There is a competition between shipping and fisheries for marine space, similarly as with offshore wind farms, which represent potentially dangerous obstacles for shipping. Tourism is a strong factor for shipping due to leisure boating and the increasing cruise ship sector. Coastal management is an essential regulatory instrument with regard to coastal shipping.

**Knowledge gaps**

There are large unknowns concerning underwater noise generated by ships and its impacts on marine life. A monitoring network for underwater noise would help to get reliable data. Furthermore, more emphasis should be put on the construction of quiet ship hulls and propulsion systems, and the topic of marine noise should be further implemented in IMO measures. The effects of emission abatement techniques and their waste streams ($SO_x$ scrubbers, catalytic converters, use of gas or hydrogen
as a marine fuel) should be investigated and evaluated as to their benefit or harm to the ecosystem. The climate impacts of shipping, emissions of black carbon, methane and $CO_2$ should be quantified and the use of biofuels should be evaluated. Further questions are:

- *Shipping – aquaculture:* What impacts could shipping have on aquaculture?

### 6.15 Chemical contaminants

**Key messages**

Concentrations of chemical contaminants in the Baltic Sea are affected by climate change and excessive nutrient inputs, and subsequent changes in meteorological conditions, hydrology and organic carbon transport/cycling. Contaminant inputs and releases from environmental reservoirs are also influenced by regional human activites such as land use, dredging, marine traffic and offshore constructions. However, the interactions are complex. Counteracting direct and indirect impacts of climate
change and other factors on emissions, distribution, degradation and toxicity are likely to occur. Combined effects are difficult to predict and are expected to be specific to the region and and the chemical in question.





**Knowledge gaps**

The types and amounts of organic contaminants entering the Baltic Sea are not well characterized, and major sources and pathways through the environment and food webs are not known, even for many legacy pollutants. Methods to estimate or project use patterns and emissions lack for the majority of organic substances used. It is not known what the combined effect of the thousands of chemicals in the Baltic Sea is, and what type of chemicals are the main drivers for the mixture toxicity. It is a great challenge to separate the effects of organic contaminants from natural variability and other stressors in the Baltic Sea, and to predict effects at population or ecosystem level from observations of sublethal effects in individuals. Severe detrimental effects at the population level have been linked to specific pollutants in the past, but many ecotoxicological effects on the Baltic Sea ecosystem due to the cocktail of chemicals are largely unknown. Regularly monitored organic contaminants such as dioxins and dioxin-like PCBs are currently present in Baltic Sea fish at too high levels, making sales restrictions and recommendations of maximum fish intake necessary to protect human health. Further questions are:

- *Chemical Contaminants – Submarine Groundwater Discharges.* What is the contribution of organic contaminants released through submarine groundwater discharges, and how strong is this effect?
- *Chemical Contaminants – Marine ecosystems.* The toxicity of organic contaminants and metals to many marine organisms is well documented in laboratory and field studies, yet there are many unknowns concerning how and to what extent organic contaminants have an impact on certain compartments of the marine ecosystem, in particular the microbial communities and thereby the potential implications for critical biogeochemical processes.

### 6.16 Dumped or unexploded military material

**Key messages**

Although dumped warfare agents in marine environments have been studied only recently, several projects in the Baltic Sea have produced some knowledge. The degradation processes and transport mechanisms of chemical warfare agents and explosives are almost fully recognized although the list of breakdown products is still incomplete. Dumped and unexploded military material poses a largely unknown danger to Baltic Sea waters, its ecosystems and food webs up to humans. The monitored sites show advanced corrosion stages, so that the danger of leaking toxic substances to the surrounding waters is very likely to increase. Warmer waters may accelerate corrosion rates, but if dumpsites are in hyp- or anoxic regions, corrosion rates can be expected to decelerate. A re-oxygenation of these areas would then increase corrosion rates. Fishing may be affected by endangering fishermen, and by poisoning commercially exploitable fish. It will be a challenge for coastal management to cope with this unknown danger.

**Knowledge gaps**

Not much is known about the metabolic pathways of munition related compounds in biota. This may lead to the underestimation of sublethal effects of those compounds. Further studies are needed to identify all the degradation products, their lifetime in the marine environment, and toxicity thresholds of their metabolites. Many dumpsites are unknown and may be in accessible coastal regions. Due to the scattered distribution of munitions in the Baltic Sea, further surveys and identification are needed to quantify the number of munitions that are not fully corroded and could create a source of contaminants for the Baltic Sea. How substantial is the danger of released toxic substances for ecosystems, food production and humans? Further questions are:

- *Dumped Miltary Material – Marine Ecosystems:* It is much unknown how and to what extent, and when, an impact on marine ecosystems can be expected.
- *Dumped Miltary Material - Coastal management:* Is the fate and potential danger of unexploded ordnances in the dumpsites an issue for coastal management? How is the management organized in dealing with this danger? How can modeling efforts help estimate the danger?



### 6.17 Marine litter and microplastics

**Key messages**

Marine litter is a well-recognized issue for the marine environment. The number of litter and microplastic items in the water column, sediments, and at the beaches are variable and challenging to compare. Generally, the highest concentrations can be found in areas with high anthropogenic activity, and coastal regions including beaches are considered to accumulate microplastics. In the densely populated catchment area of the Baltic Sea, microplastic discharge via wastewater treatment plants and stormwater runoffs play a major role. The relevance of interactions between microplastics and microorganisms seems to be rather low, e.g. there is so far no evidence of microbial biodegradation of plastic polymers in the marine system. Ingestion of microplastics by higher organisms, e.g. fish, has been observed frequently. It has also been discussed whether plastic particles can be transferred up the food chain, as well as whether organic pollutants associated with microplastics can enter the food web via uptake of microplastics, but so far, there is no sufficient evidence. Cigarette stubs at beaches represent a large group of litter, hence recreational activities/tourism play an important part as producer of litter, but are also affected when tourists are repelled by litter at beaches. Lost or abandoned ("ghost") fishing nets contribute to the plastic load, but their distribution in the Baltic Sea is largely unknown. Distribution and abatement measures are issues for coastal management.

**Knowledge gaps**

Concentrations of plastic particles in the Baltic Sea are scarce and highly variable, due to challenging and not harmonized methodologies and data on ecotoxicological effects of microplastics on Baltic Sea biota are rare. Methodological challenges exist, in particular for experimental studies targeting the small microplastic fractions, and due to environmental contaminants masking microplastic-related effects. Also, data on PAH (polycyclic aromatic hydrocarbon) accumulation on plastic and subsequent degradation are insufficient for the Baltic Sea. While considerable plastic emissions occur in the Baltic Sea, it is not clear yet how urgent and eologically damaging the plastic problem in the Baltic Sea is compared to other environmental problems. Further questions are:

- *Marine Litter – Chemical Contaminants:* What is the impact of the different types of marine litter, e.g. how are microplastic particles degraded down to macromolecules, and what is the relevance for the release of organic contaminants of different types? What are these types of contaminants and what impacts could they have on ecosystems and harvestable food organisms for human use? Which substances are the most dangerous?
- *Marine Litter – Marine Ecosystems:* There are many uncertainties about how and to what extent marine litter affects marine organisms, ecosystems and food webs.
- *Marine Litter – Coastal management:* How far are marine litter, its discharge and distribution patterns a subject for coastal management?

### 6.18 Tourism

**Key messages**

The Baltic Sea is an important destination for tourists, with a substantial impact on the regional economy and on ecosystems. Through mobility (travel) and shipping (leisue and cruise tourism), tourism has a strong impact on the region´s climate greenhouse gas emissions. On the other hand, tourism is very much dependent on the climate, and an increasingly warmer climate is expected to be beneficial for the tourism sector. Tourism is a strong user of coastal land, and has considerable impacts on the region´s nutrient loads. Offshore wind farms may be a disturbing element in the coastal sea for some tourists, and as coastal tourism and yachting are expected to grow, conflicts may increase in the future. This and the general conflict between different coastal uses like tourism, coastal protection, nature conservation and protection, is dealt with by coastal management.



**Knowledge gaps**

In the future, it is important to monitor and control pollution loads allocated to touristic activities, including the release of nutrients, litter, chemicals and oil. Multi-stakeholder governance and coordinated collection of sustainability management indicators are needed. Monitoring data is required both for the sustainable development of the tourism sector, consisting of a large number of enterprises of different sizes, and for planning mitigation and adaptation policies in the overall catchment area of the Baltic Sea. Further questions are:

- *Tourism – Marine ecosystems:* How does tourism in all its variations affect the different compartments of marine ecosystems?

**6.19 Coastal Management**

**Key messages**

The task of coastal management is to provide for the sustainable use of the coastal zone, and to balance the often competing interests of the different stakeholders in the coastal zone. Thus, there are many connections between natural and anthropogenic factors. Coastal management decisions and mechanisms regulate activity and therefore play (or can play) a significant role in all factors that are influenced by human activity. Infrastructure such as ports, coastal and offshore constructions (wind farms, oil and gas infrastructure), as well as erosion protection infrastructure (groynes, revetments and seawalls, renourishment) change natural coastal processes and sediment budgets, and their regulation through planning and approval processes is of utmost importance. Coastal management needs to balance competing interests but it can also explore synergistic uses in the coastal zone, such as wind farms and certain types of aquaculture. Living and other natural resources like fisheries, agriculture, aquaculture and nutrient releases need to be managed. Problems related to litter and microplastics, pollution by contaminants released from dumped or unexploded military material and the ubiquitous and manifold impacts of tourism activities at the coast all require regulation and management.

**Knowledge gaps**

Coastal management uses available mechanisms (plans, laws, regulations, engagement activities etc) to provide for the sustainable use of the coastal and marine environment. The knowledge gaps that apply to all natural and anthropogenic factors can therefore be summarized as follows:

- What data and analyses are needed to enable coastal management to be more effective, and how can the required information be obtained?
- What tools and systems (such as decision support tools are needed to enable decision makers to provide the best possible management for a sustainable Baltic Sea?
- Recognizing that the Baltic Sea does not respect borders, how can management mechanisms be integrated across jurisdictions?

**7 Summary and discussion**

The industrial revolution and the subsequent developments have dramatically changed the world, with massive benefits for humans and concomitant detrimental effects for the environment. A by-product of the industrial revolution was the industrial production of nitrogen fertilizers with the Haber-Bosch process (Smil, 1999). With the eradication of hunger, population numbers have since exploded, with various environmental consequences like anthropogenic climate change, eutrophication, overfishing and pollution. There is a growing understanding that tools are required to address the effects of these multiple intertwined factors. The detection and isolation of single factors from others is crucial for attributing detrimental effects and managing a healthy environment.



We analyzed the connections between different factors affecting the Earth system of the Baltic Sea region, by simply describing what is known from the scientific literature to assess how strong and relevant the connections are. Many publications have dealt with this complex issue on a general scale (differently termed multiple or cumulative effects, stressors, pressures, drivers),

mostly using statistical analysis or modeling approaches to better describe the problem, or management procedures to cope with it (e.g. Crain et al., 2008; Halpern et al., 2015; Gunderson et al., 2016; Liess et al., 2016; Elliott et al., 2020; Gissi et al., 2021). Many investigations have focused on the harvestable upper part of the food chain, e.g. fish and their food resources (Bolt et al., 2014; Andersen et al., 2017; Stelzenmüller et al., 2018). The Baltic Sea region or parts of it have been treated by various investigations (Jutterström et al., 2014; Andersen et al., 2015; Reusch et al., 2018; Andersen et al., 2020). Indices have

been calculated to quantify the different effects and interrelations (e.g. HELCOM, 2018a; Korpinen et al., 2012; Blenckner et al., 2021) and the work has also been incorporated into decision support systems or general advice for decision makers (Meier et al., 2012; Meier et al., 2014; Hyytiäinen et al., 2021). In the Baltic Sea, the detrimental factors with the most substantial impact have been identified as being eutrophication, hazardous substances, non-indigenous species and fisheries (Korpinen et al., 2012; Andersen et al., 2015; HELCOM 2018a; Andersen et al., 2020; Blenckner et al., 2021), but also acidification and

climate change (Jutterström et al., 2014). Looking at our DSPIR analysis of the different drivers (Table 1), it becomes evident that industry, transport, energy production and other economic activities, together with food production, are the most evident driving forces. Thus, from our analysis, it seems that socio-economic factors, i.e. those directly connected to human activities, exert the strongest impacts on the environment, i.e. the food production sector with agriculture, aquaculture and fisheries, the energy production sector (wind), shipping and tourism (Figure 8). Climate change plays a particular role as it has an integrating

effect on other factors. The main message from this analysis is that climate change represents the overarching factor, affecting almost all of the other natural and human-induced factors. For some we know the effects reasonably well (e.g. coastal processes, agriculture, aquaculture, fisheries); for others, we know little (distribution and fate of contaminants, microplastics and dumped ammunition).

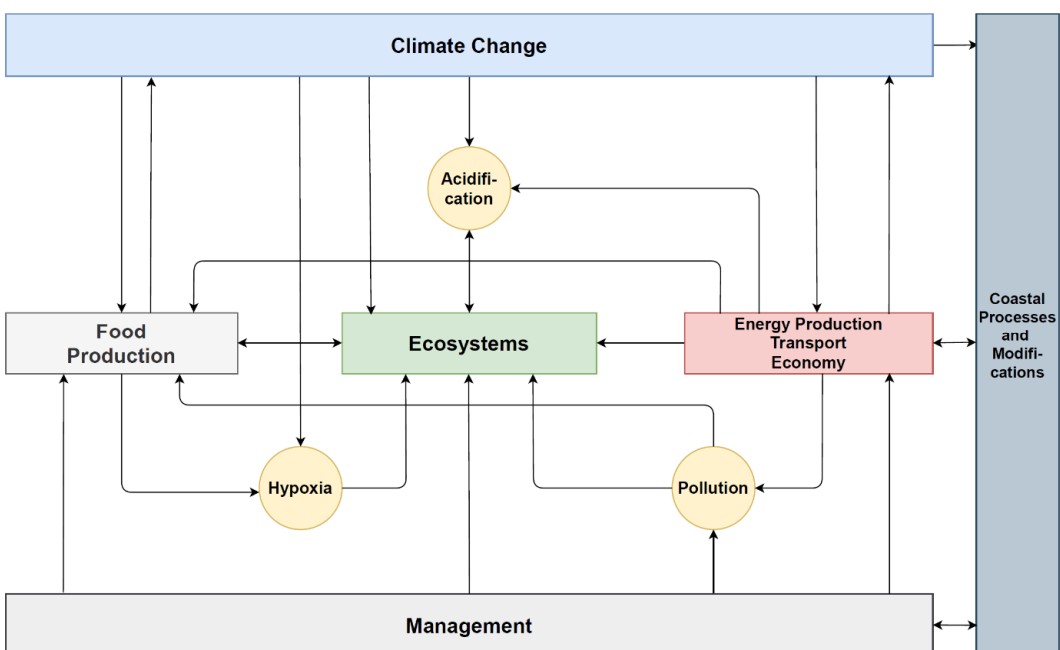

**Figure 8: Schematic view of the interrelations between the different factors. *Food production* includes land use, agriculture and nutrient loads, aquaculture and fisheries; *Energy production, transport and economy* includes offshore wind farms, shipping and tourism; *Coastal processes and modifications* includes coastal processes, SGDs and river**



**regulations; *Ecosystems* includes non-invasive species; *Pollution* includes contaminants, dumped military material and marine litter (Marcus Reckermann).**


Food production on land and in the sea has severe consequences on the ecosystems, through nutrient loads and resulting eutrophication and hypoxia, and also on climate, due to the conversion of forests to agricultural fields. More nutrients are distributed on the fields than crops can take up and convert into biomass. Artificial fertilizers are used in excess and incentives to adjust fertilizer inputs to plant needs are few. Livestock, i.e. pigs and cattle excrete highly useable fertilizers, but this manure

is mostly not used for fertilization, and use efficiencies differ largely between countries. The excess nutrients are largely washed to the sea where they have resulted not only in detrimental implications like excessive algal growth, decay and oxygen consumption, but also in increased fish stocks and catches. Fish stocks and fisheries have profited from the fertilization of the Baltic Sea in the mid-20th century (Adjers et al., 2006; Eero et al., 2016), and catches increased dramatically 8-fold over the course of the 20th century (Elmgren, 1989; Elmgren, 2021), mainly because of enhanced fishery methods and infrastructure

but also because of a heavily fertilized Baltic Sea. Fishing pressure and climate-related changes are presumably the main reason that some commercially used fish stocks broke down in the late 1980s (Dippner et al., 2008), but we do not know the effect of reduced nutrient loads for this breakdown, of which a reduction also goes in parallel. The efforts of nutrient abatements and regulations in the Baltic Sea, mainly though HELCOM, have been a success story (Reusch et al., 2018). Nutrient loads have been reduced since the 1980s (Gustafsson et al., 2012) but legacy nutrients (those that are hidden in the sediments and

are released under certain circumstances) remain a problem, especially phosphorus (McCrackin et al., 2018a).

Offshore wind power production will presumably increase considerably in the future, as the demand for renewable energy is growing in the effort to mitigate climate change. Extensive sea areas must be dedicated to wind power generation. A quick back-of-the-envelope calculation demonstrates this: to replace an average fossil fuel power plant of 2 GW, an area of about 15 x 15 km must be assigned to the wind farm, assuming the currently largest wind turbines of 10 MW with a rotor diameter of

100 m, and an estimated distance of 1 km between the individual turbines. This will be a challenge for marine spatial planning, increasing the political pressure to establish dual or multiple uses, e.g. with food production (aquaculture). Still, there are potential problems with efficiencies as large wind farms may experience self-shadowing effects, affect the micro-climate and potentially downstream ecosystems (von Berkel et al., 2020). Impacts on ecosystems may also be positive as the artificial reef effect may be beneficial for fish and eventually also for fisheries even though it is banned from the wind farm areas. Underwater

noise may be a problem for marine mammals or fish mainly during the construction phase. There is very little knowledge on marine noise impacts on marine mammals and other organisms (also through shipping). Offshore energy production also affects terrestrial land use as the energy produced offshore needs to be transferred to consumers over large distances (large powerlines over land or in soils).

Gas and oil extraction is rather minor in the Baltic Sea, as compared to other marginal seas like the neighboring North Sea.

Oil and gas production and transportation is not treated in the main part of this paper but it may have severe implications in case of permanent spills and accidents (Hassler, 2011). Precautions include the use of double hull tankers, safer shipping lanes and navigational aids (Moldanová et al., 2018) and possibly smart ship routing (Soomere and Quak, 2013). There are known oil and gas fields only in Polish and Russian (Kaliningrad region) territory, however, oil terminals exist in the Russian part of the Gulf of Finland and in the Baltic States, so the Baltic Sea is a transfer region for oil and gas tankers to the markets of the

world. Gas pipelines also cause environmental and political concerns (Nordstream 1 and 2; Lidskrog and Elander, 2012; Heinrich, 2018), transferring gas from Russia to Germany. The Baltic Pipe Project (2021) planned for 2022 intends to provide the infrastructure for transporting gas from Norway to Denmark and Poland (Górski, 2020). Overall, the impact of oil pollution in the Baltic Sea so far is considered to be relatively low (Kostianoy and Lavrova, 2014; HELCOM, 2018a) and the number of oil spills has decreased over the past 20 years, presumably through monitoring and airplane and satellite surveys (HELCOM

2018a) but the increasing volumes of surface oil transport gradually increase the relevant risk.



Maritime transport of goods affects not only the climate through the combustion of fossil fuel but has many direct consequences for marine life and water quality. It was shown in the past that regulations work slowly but efficiently (e.g. double hull tankers, ballast tank regulations, antifouling regulations, different others), so a transfer to more sustainable shipping can be expected in the future. The release of chemical contaminants is also highly regulated but the vast diversity of the different substances
makes a thorough monitoring difficult, let alone the many unknown effects and reactions between the different substances, how they affect organisms and how a warmer and wetter environment affects concentrations and pathways, transfers and transformations between land, atmosphere and the sea. Impacts on food webs and consequences for human marine food resources need to be constantly assessed and monitored as many new substances are released into the environment.

Marine litter originates from all human activities: offshore platforms, shipping, lost containers, fisheries, aquaculture,
agriculture, municipal waste, tourism. A large fraction is carried to the sea by riverine runoff. The effects on the environment at least for the large size fractions are visible and well documented but the effects of the micro- and nanosized particles down to colloid and molecular scales are largely unknown, as is the effect by their constituents on biota and the food chain. The toxicity to biota along the food chain up to humans remains unclear, and how inert microplastic particles remain over a long time horizon. Degradation rates for some polymers are very slow, so that microbial degradation will presumably not be a big
help in removing microplastics from the waters.

A legacy of human activity for which we do not know how serious potential impacts may be are the ammunitions of chemical warfare or dumped or unexploded ammunitions from the second world war. These hazardous materials were deliberately dumped into the sea during or after the war. This shows the low level of understanding and concern, and the notion of "out of sight, out of mind" in the past. It is just not known how dangerous these materials are on scales of more than just the very
vicinity of the dumpsites, e.g. is the dilution effect of any leaking toxic substances sufficient or are ecosystems and people living at the coasts and upper food levels including consumable fish seriously affected in the near future? Many containments are expected to corrode in the coming years and decades, so this is a large unknown threat with high damage potential. Models to simulate dispersion and dilution scenarios are needed. Whether we look at marine litter, microplastics, warfare agents or chemical contaminants, we see a low interference with other factors (including climate change) but a strong and direct impact
on ecosystems (which are largely unknown), and on humans as the top consumer of marine products. As this is a major health concern, there are fewer unknowns.

Climate change directly affects atmospheric and marine properties such as air and water temperature, precipitation, runoff, salinity, sea ice, sea level, and acidification. It affects the food production sector as the growth of crops on land is temperature and water dependent, and in the sea, temperature is a significant growth factor for cultured fish and shellfish. Fish stocks are
dependent on climate-sensitive availabilities of food organisms and stratification (Möllmann, 2019). In some cases, climate change affects different factors, which work antagonistically, and the net effect is not apparent. Sea level rise, for example, is expected to have a strong impact on the southern coasts and ecosystems of the Baltic Sea. Salinity (Lehmann et al., 2021) is an essential factor for marine life in the Baltic Sea, and many species live in narrow tolerance bands. Hence, it is vital to know how salinity will change in the future. It may increase through intensified inflows (as sea level rise would widen the passages
in the Danish Belts and Sounds), but conversely, it may be reduced by increased runoff (Meier et al., 2021c). Currently it is unclear which effect is prevailing.

The energy production, transport sector and tourism are strongly affected by climate change. It is the initial driver for the expanded use of wind power, and changes in wind speeds and frequencies affect the efficiency of wind turbines. However, current projections do not indicate increasing wind speed in the area (Christensen et al., 2021), so an increased yield of wind
power can only be achieved through more and larger turbines and larger wind farms, with all the consequences for other uses and of course ecosystems. Shipping in the north may be facilitated by dramatically reduced ice covers in the future, and fairways in the south may be safer in shallow passages with rising sea levels. The tourist industry in the Baltic Sea region can



be expected to profit from warmer summers, as its coasts are ideal for multiple summer leisure activities. Both leisure boating and cruise shipping can be expected to profit from warmer temperatures.

Climate change may aggravate many environmental problems, which have their causes in other anthropogenic factors, reducing the resilience of marine ecosystems. Climate change may affect the distribution of chemical contaminants in the environment through warming and changed precipitation patterns. As organic pollutants can travel through air, water and in organisms up the food chain, and they experience transformations along the way, which depend on physical and biogeochemical factors, we see a complex array of possible modifications, some of which may be deleterious. Nutrient loads

per se are not affected by climate change directly, but their pathways, turnover and regeneration rates, as well as oxygen consumption rates and hence hypoxia, are.

Socio-economic factors such as population growth, urbanization, technological development, life style and values play a significant role in developing the Baltic Sea and its coastal regions, more than the direct effects of the changing climate, but they are closely interwoven with it. A decision on land use may be primarily a socio-economic one, but it will be influenced

by climate change and its impacts. Land cover also includes urbanization and sealed surfaces in urban areas. These have a high relevance for flooding as they exacerbate the flood risks in urban areas. The same meteorological event may have a low impact where soils can absorb water and distribute it downstream and via the natural aquifers, but it may turn into a catastrophic event for human infrastructure if the drainage systems below the sealed surfaces are too weak. Hence, extreme events may be exacerbated by human infrastructure. This is similar in the case of regulated rivers, where storm floods may run up higher, and

natural flooding areas are either embanked or sealed.

Certain regional factors may exert a feedback on the climate. Anthropogenic greenhouse gas emissions have been convicted of being be responsible for the most part of the warming of the past decades (Bhend, 2015), but apparently it cannot explain the warming completely (Bakhordarian et al., 2016). Other regional factors have been discussed, such as the decreased concentrations of aerosols in the atmosphere (Hansson and Bhend, 2015) and land cover changes (Gaillard et al., 2015).

Regulations to reduce air pollution in the 1980s led to reduced aerosol concentrations in the air, and thus to a lower cooling effect through blocking of incoming insolation (Hansson and Bhend, 2015, Bakhordarian et al., 2016). Aerosol reductions are a strong candidate to contribute to the observed warming (or more precisely, to a reduced cooling effect on the regional scale, von Storch et al., 2021). This would mean that a successful mitigation effort to reduce dangerous human impacts would act to increase another one (climate warming). This once again shows how interwoven natural and human-induced impacts are and

how they can work antagonistically on different factors.

There are considerable knowledge gaps on acidification in the Baltic Sea, and many aspects remain rather uncertain, e.g. the source of variability of alkalinity and its trends in the various basins. As alkalinity in the Baltic Sea is generally lower than in the ocean (e.g. Hjalmarsson et al., 2008), Baltic Sea waters at first approximation are more sensitive to acidification. However, the recently observed alkalinity increase in the Baltic Sea is compensating for most of the present-day acidification (Müller et

al., 2016), especially in the north. In the long term, the effect of acidification is expected to be stronger than that of alkalinity increase and pH will most likely drop (Gustafsson and Gustafsson, 2020), but the scale of this change is rather uncertain and may differ between the sub-basins. Also, the effects of acidification on organisms are not clear, and most investigations found negligible or inconsistent effects under *in situ* conditions. Potential impacts on the food web, effects for human food and resources (fish, shellfish, macroalgae, seaweed etc.) remain unclear. Although the contribution of ship- and land based nitrogen

and sulphur oxide deposition (e.g. role of scrubbers) was shown to be rather low as compared to $CO_2$-driven acidification, it may become important in the land-locked Baltic Sea as shipping increases.

All these different factors and interactions create the need for management and policies, at their different facets. While the goal of management is sustainability, it follows the needs of society (e.g. food production, energy and other resources from the sea, transport, coastal infrastructure and protection, tourism, inspiration), and at the same time avoiding environmentally

detrimental consequences. Coastal management needs to balance different opposing stakeholder interests, and is a political





rather than a scientific process in which scientists actively participate. Although management and regulations have shown some success in the Baltic Sea and worldwide (e.g. monitoring surveys, reductions of nutrients, pollutants and oil spills, HELCOM, 2018a), considerable challenges are still ahead, e.g. related to fisheries and ecosystem management. In the end, the measures need to balance between different competing interests. Human benefit has been at the centre of most efforts, which

is evident in pervasive concepts like "blue growth" and "ecosystem services" which perceive nature as a "service provider" for human welfare and development, rather than protect it for its own sake (e.g. Omstedt, 2020). Put bluntly, it may be concluded that only factors relevant for the human benefit are considered in these concepts, and species or biomes that do not fulfil this criterium, or are considered a nuisance, are not worth saving and may perish. Thus, the human perspective is and will remain ubiquitous.

Marine protection can be justified by the need to secure the future provision of marine ecosystem services to benefit future generations. In this light, "Ecosystem-based Management" and "Marine Protected Areas" also represent anthropocentric concepts and the ultimate goal of management can be described as to provide a fair intergenerational distribution of ecosystem services (and avoid spoiling the resource). The UN Decade (2021-2030) of Ocean Science for Sustainable Development (e.g. Pendleton et al., 2020) could be an essential inspiration for future coastal management, also for the Baltic Sea (see also United

Nations, 2021)

**8 Conclusions**

The large sectors that provide goods and services to humans (food, energy, transport, recreation) have the most direct impact on the Baltic Sea environment. From a scientific point of view, it is not possible to render a verdict on the most harmful impact on the environment, as this is largely not a scientific but a social construction. Different stakeholders, e.g. scientists, coastal

dwellers, people who live from the sea, local policy makers or environmental activists may all have different conceptions. This is also the case between the different riparian countries (Lundberg, 2013; Martinez et al. 2014; von Storch, 2021). Science can provide the facts as far as possible to help establish and support management options.

Climate change has an overarching impact on all other factors and affects their interactions. Most of the environmental challenges in the Baltic Sea (including nutrient loads, hazardous substances, safe recovery of dangerous dumped military

material) are considered and managed at the regional scale (e.g. by HELCOM). Regional solutions are at hand through domestic political decisions and good management, and its efforts and results can be seen on the regional scale. Climate change will continue over the coming centuries, with its impacts of warming, changing precipitation patterns, less ice cover, rising sea level and acidification. Regional climate change is embedded into the global one, and it cannot be mitigated without mitigating the global one. Hazardous substances and microplastics resemble greenhouse gas emissions in their global dimension and local

appearance. They are produced and distributed all over the world, wherever humans live. It is virtually impossible to recover the material already released to the environment, and we need to live with the consequences of this persistent type of pollution for the time being.

By driving the enormous technological and socio-economic changes of the past 150 years, the inventive human spirit has caused the environmental problems we are facing; it is logical to assume that it will be able to come up with the solutions as

well. Information and open discussion, stakeholder involvement, locally tailored best practices, management and technological innovation will be keys for achieving this goal.




**Author contributions**

**MR** elaborated the concept, wrote the abstract and sections 1-4, 6-8, authored Sections 5.3. and 5.6, and contributed to different parts in section 5. **AO** co-elaborated the concept and authored Section 5.4. **TS** authored 5.2 and provided many useful

comments to all sections in an early stage of the manuscript. **JA** authored Section 5.7.; **NA** contributed to Section 5.13.; **MB** contributed to Section 5.15; **JB** authored Section 5.16.; **CPSB** contributed to Section 5.13; **TC** contributed to Section 5.13; **MC** contributed to Section 5.16.; **ME** authored Section 5.11.; **KH** contributed to sections 1 and 7; **JPJ** authored Section 5.14.; **AK** authored Section 5.10.; **EK** co-authored Section 5.1.; **KK** contributed to Section 5.4.; **XGL** authored Section 5.13.; **MMcC** authored Section 5.9.; **HEMM** co-authored Section 5.1., **SO** authored Section 5.17.; **KP** authored Section 5.19. and provided

useful comments to Section 7; **AP** authored Section 5.8.; **JS** authored Section 5.18.; **BS** authored Section 5.5.; **EU** authored Section 5.15.; **AW** authored Section 5.12.; **EZ** co-authored Section 5.1. All co-authors contributed to the bullet lists in Section 5, according to their expertise.

**Competing interests:** The authors declare that they have no conflict of interests.


**Special Issue statement:** This paper is part of the Special Issue "Baltic Earth Assessment Reports (BEAR)" in Earth System Dynamics.

**Acknowledgments**

**TS** wa supported by the European Economic Area (EEA) Financial Mechanism 2014–2021 Baltic Research Programme, project SolidShore (EMP480), and the Estonian Research Council Grant PRG1129. The research work by **JB** was co-funded by the European Union (European Regional Development Fund) under the Interreg Baltic Sea Region Programme 2014–2020, project #X005 DAIMON 2. It was conducted as a part of an international project co-financed from the funds of the Polish Ministry of Science and Higher Education programme entitled "International Co-financed Projects" in the years 2019–2021;

agreement no. 5051/INTERREG BSRBSR/2019/2. **KK** was supported by the Polish National Science Centre (grants no. 2015/19/B/ST10/02120; 2019/34/E/ST10/00167), and BONUS INTEGRAL project funded by BONUS (Art 185), jointly from the European Union's Seventh Framework Programme for Research, Technological Development and Demonstration and the Polish National Centre for Research and Development. **XGL** thanks the Danish ForskEL/EUDP (grant no. PSO-12521/EUDP 64017-0017) for support. **KP** was supported by the European Regional Development Fund program Mobilitas Plus, Estonian

Research Council Top Researcher Grant MOBTT72, reg. no. 2014-2020.4.01.16-0024, and the European Economic Area (EEA) Financial Mechanism 2014–2021 Baltic Research Programme, project SolidShore (EMP480), and the Estonian Research Council Grant PRG1129. **BS** was supported by the Norway Grants 2014-2021 operated by National Science Centre under Project Contract: 2019/34/H/ST10/00645 and the National Science Center under Project Contract: 2019/34/E/ST10/00217. **EU** was supported by The Swedish Agency for Marine and Water Management through the grant

1:11 - Measures for marine and water environment.






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
