# Peer review of "Human impacts and their interactions in the Baltic Sea region"

_Earth System Dynamics, 2021_

## Author Comment (AC1)

**Reviewer 1**

Comment: Human impacts …

**General:**

This is a fine overview about a very complex field, and the authors are to be applauded for a good overview. In principle it could be accepted as is, but I take the opportunity to list a few issues, which the authors may want to think about, plus as series of minor points.

*Thank you, we appreciate the thorough review and suggested changes.*

It would be helpful, If all aspects disused would be introduced with a similar sketch-diagram showing the major and minor influences upon each other.

*We see the reviewer´s point, and we agree that this would be very good. In fact, we have been going through many iterations and attempts to visualize the different influences on each other. Actually, it became clear that with the many interdependencies, either a sketch would be too chaotic und unreadable, or we would have a proper sketch for each factor, which would mean that the page number would increase even more. We may opt to have these 19 single graphs as supplement files, which may be downloaded from the ESD website.*

The societal dimension could be covered more seriously. For instance, for tourism, it may matter more the perception of climate change than climate change itself. Also societal preferences may change preferred tourist sites.

*We will add these aspects but we see that there are few references on these effects. A paragraph is added in section 5.1.1*

Figure 8 places ecosystems in the center- but there should also be "societal system", which is not really part of the ecosystem (or the rest of the manuscript had to be written differently. In the summary the relationship of society/ecosystem and the role of societal value systems is dealt with in a superfluous manner (characteristic of ecologists)

*We will revise Fig 8, adding societal/human aspects, and add a paragraph in the summary section as well as in other sections where appropriate*

In general, the historical dimension is a bit weak. What about "pristine"/"undisturbed" conditions, earlier "strange" events; so far it is almost entirely on conditions since 1950 or 1990. (e.g., 6.4, 6.12)

*We will include historical/background values/conditions as far as possible. Note: section 6 key messages and knowledge gaps are now incorporated into section 5 to eliminate repetitions*

In the section of shipping, the issue of using different fuels should be mentioned, also the effect of phasing out crude oil (also 6.14)

*The use of crude oil and LNG are discussed in section 5.14; we will make it clearer and shortly add the aspect of phasing out crude oil.*

In general: reduce the list of linkages wot significant ones, and mention: "the issues X, y, and z likely are insignificant."

*Not sure we understand this comment. We have stated in the bullet lists in section 5 only the connections which were indicated as significant (+); and we have also discussed the*

*ones with a question mark (?) because we believe that these linkages may be interesting. The insignificant ones are already omitted from the bullet lists. They are there in Table 2 but we think they should remain there for the complete picture.*

In Sec 6.1, the issues: effect of aerosol emissions and of land use need to be listed aa major issues in the knowledge gaps.

*These effects will be added to the knowledge gaps. Thank you!*

In Sec 6.8 – the issue of cities should be addressed.

*Of course, cities should be addressed in this section. Thank you!*

Why have big constructions not been addressed – this was a big issue some years ago, and partly still so (Fehmarn belt tunnel).

*The large infrastructure projects (oil and gas pipelines) Nordstream1 and 2 as well as the Baltic Pipe project are discussed but the Fehmarn Belt tunnel was not, as ecological impacts are probably very much eliminated as compared to a bridge, which was previously planned.*

**Minor**

- Both tables are hard to read.
  *We will try to increase readibility and explain better*
- line 295: authors'
  *OK*
- 328: I guess this is not so much an issue of the latitudinal position but of the fact that most of Earth is covered by the ocean, but that the sa is only a minor part of the Baltic Sea region.
  *We believe it is related to the northerly position of the region and the proximity to the polar region. The feedback processes related to snow/sea-ice in the winter half of the year are responsible for the large temperature increase. Looking not only at winter it is also clear that the region is to a strong degree continental and the small size of the Baltic Sea implies that it can't reduce the warming in the full region in the summer. So for the sake of brevity we prefer to keep this phrase.*
- 355: warming of the … why so?
  *Thermal expansion is the largest contributor to global sea level rise. Sea level rise in the Baltic Sea basically follows global mean sea level rise (with some modification due to the melting of the Antarctic ice sheet). We will change "most obvious" to "profound".*
- 363 – the numerical estimate of sa level rise – still valid in view of the new IPCC report; also this number MUST be associated with a time horizon – when?
  *… until 2100.*
- 545 – emissions of chemicals …
  *OK*
- 604 – "unfortunate" for whom?
  *The term will be deleted*
- 623 – "dangerous" – for whom?
  *The sentence will be re-phrased with neutral term to eliminate subjective terminology*
- 630 ff – needs better structuring. Why "as a consequence"?
  *The sentence will be re-phrased for better understanding*

- 643 – What is the "running out of sand"-concept?
  *Sand extractions and potential consequences will be better explained*
- 695ff – unclear; what does the sentence "nt a total load …" tells the reader?
  *The concentrations of heavy metals in the eroding soft cliffs are low, but due to the total volume of eroding material, the introduction of heavy metals into the coastal sea in this way is considerable.*
- 801 – a repetition
  *We cannot spot the repetition*
- 823 – are these significant effects or just something tny? How much "may speed up corrosion"?
  *Corrosion rates in Baltic Sea conditions were studied in field experiments in different anoxic/oxic transition areas, i.e. the Bornholm Deep and the Gdansk Deep western fringe. Metal coupons placed in near bottom water in fully oxic, oxic/anoxic and fully anoxic sites showed almost a doubling of corrosion rates between anoxic and transitional conditions, and ca. 20% increase in transitional conditions as compared to oxic. Therefore, the following sentence was added: "Therefore, hypoxic events and the resulting changes in oxic conditions may have a considerable impact on corrosion rates, e.g. a doubling in transitional as compared to fully anoxic conditions (Fabisiak et al. 2018)"*
- Figure 2 – needed?
  *Will consider to delete*
- 892 – "importance of short-term variability" – consisting of what?
  *For flux calculations of surface $CO_2$ in coastal areas, short term variability on hourly scales needs to be considered*
- 1067 – BSAP?
  *Will be explained*
- 1095 – refer to climate change section
  *This paragraph will be shortened considerably*
- 1105ff – us fold face key words.
  *Key words will be printed in bold*
- 1181 – "driving a regime shift through" sound a bit Germanic.
  *The language will be adjusted*
- 1255 – "projected decrease in the South" was earlier declared uncartein.
  *The wording will be adjusted*
- 1285 – "migration" – of whom?
  *Will be adjusted; "migration of species"*
- 1323ff something missing
  *We do not understand the comment*
- 1335 – "loss of social welfare" - hm, this may be a value-based assertion by a friend of pristine ecosystems
  *This will be either discussed or deleted*
- 1355 – I remember that kay Emeis had an early key paper on this.
  *Will be investigated and cited if applicable*
- 1448 – what does blue and blue-green stand for?
  *blue: sea-based, blue-green: sea-land based; will be explained*
- 1475 – More probable – why?
  *Will be edited*
- 1480 – add number for usage of wind energy
  *We will see if this can be calculated and provided*

- 1601 – delete – trivial and vague; same with 1608
  *Will be deleted*
- 1658 – "An engineering" – much too general an assessment.
  *The sentence can be deleted*
- 1664 – "periodicity of climate" – what shall that be: annual cycle?
  *replace with "climate variability"*
- 1728 – deepening
  *should be "depending"*
- 1774 – "gentle" is not an adequate word
  *replace with "suggesting a small increase in the Northern part…"*
- 1822 – There are --- which?
  *Yes, the associated risks and obstacles need to be briefly discussed*
- 1836 – Anholt is in the Baltic Sea?
  *In a sense, yes… but will be changed to "…in the Kattegat"*
- 1856 – discussed before
  *Yes, but not as a bullet in this section; whole text will still be revised to eliminate too much repetition*
- 1870 – what are black waters, what grey waters?
  *This will be defined in the text*
- 1877 – Kadet
  *will be corrected*
- 2935 – one scenario or an ensemble of scenarios?
  *Wrong line number, unable to find the correct one*
- 2308 – something I was wondering while reading: what about military ships and military exercises?
  *Information on this aspect may not be available; military ships and exercises may be too scarce to make an impact compared to other shipping; however, we will try to take this up in section 5.14…*
- Figure 6 carries no informational value
  *Correct, purely decorative, may be deleted*
- 2378 – very general statement
  *We think that this is an important definition of coastal management and how people understand the term*
- 2394- detrimental – for whom? This is related to values
  *The term will be deleted*
- Figure 7 – significant for the paper?
  *We are considering deletion*
- 2447 – so what?
  *Should remain as a question mark bullet*
- 2813 – farms
  *?*
- 2982 "eradication of hunger" – hu? This is a key point in the UN millenniums goals.
  *Sorry!!! Of course, hunger is not eradicated. It is the first author´s view that this is not because of principal shortcomings in food production procedures or technologies, but due to dramatic political and social inequalities in different parts of the world. This discussion is beyond the scope of this paper, so we will re-phrase the sentence, thank you for pointing this out.*

- 3145 – what is "blue growth"?
  *will be briefly explained*
  *https://www.sciencedirect.com/science/article/pii/S0308597X17306905*

---

## Author Comment (AC2)

**Reviewer 2**

This manuscript is an ambitious endeavour, covering very much fild. The topic is very important and I congratulate the authors for taking up the challenge.

However I think some major improvements need to be done with the manuscript.

***Thank you indeed for your appraisal and suggestions, which we will try to follow as far as possible.***

First of all, it is a review paper, yet the methods of the review have not been outlined at all. While a full systematic review might be too much to carry out, the methods of article finding and selection should be clearly explained and - to the extent possible - repeatable.

***We will add a paragraph in the introduction, shortly explaining the process of writing this paper. From installing the Baltic Earth Grand Challenge on this topic, to recruiting a core writing team and topic list at a dedicated workshop in Tallinn 20218, to establishing a basic framework and structure for the paper, to collecting and integrating individual contributions by co-authors, to using the "matrix" to visualize connections in a table, and describing the connections in bullet lists with references. We hope that is what the reviewer has in mind.***

Secondly, the criteria for the #+# and "?" categories should be clearly defined and constant throughout the article. This is not currently the case. For example, on r. 1106 -> and 1118, it says there is a + effect, but there are no references to back up these claims. On row 1120 the text says "may be" but it is marked as +, not as ?, and the given reference doesn't relate to tourism at all.

***We will go through all the connections, check and revise the + and ? connections so that there are references for a + connection in any case; and we will try to better explain the criteria used.***

Secondly, the paper is uncecessarily long. It covers a lot of ground, which makes it long by necessity, but that being the case, the authors should be extra careful to include only relevant information. At the moment, the introductory sections for each pressure are way too detailed. It would serve the paper better to just give the necessary amount of backgroud that is needed to understand the bullet points below, not give a general review of the pressure.

***Thank you for this suggestion, which we will absolutely agree with. We will eliminate unnecessary repititions (section 6 will be altogether deleted, and the key messages will be inserted into the section 5 subsections. Also, the introductory paragraphs in section 5 will be shortened considerably, we agree that these are less crucial to the paper than the bullet lists. Also the introduction and summary will be shortened, so that we hope to arrive at a much more concise and denser manuscript.***

I recommend thet the authors

- shorten the manuscript considerably
  ***Will do***
- check the consistency of the criteria of teh + and ? classes across all of the work (also considering the joint effects that are implicitly considered as missing!)
  ***Will do***
- outline the review process
  ***Will do***

---

## Author Response (AR1)

**List of changes, reviewer´s comments and responses to Manuscript ESD-2021-54**
Reckermann et al.: Human impacts and their interactions in the Baltic Sea region

**A. List of changes**

**1. Restructuring**

The previous section 6 "Key messages and knowledge gaps" is now incorporated into the main section 5, and a short section "knowledge gaps" is now added to each topic; "key messages" are deleted because they are mere repetitions of the already short text in section 5. These changes are not seen in the track changes.

**2. Shortenings**

Each topic description was shortened to approx. 750 words each. The introduction as well as the discussion section was shortened considerably, as can be followed by the track changes.

**3. Figure deletions**

Previous Figures 2, 6 and 7 were deleted because they do not contribute to the scope of the paper, are too detailed considering the broad scope of the paper, or just of decorative nature. That had been suggested by reviewers.

**4. Table 2 (the matrix) and visualization of the interconnections**

Table 2 is now updated, and references in the text are updated accordingly. Table 2 is now colored for a better visualization. In Table 2a, factors are sorted according to the sequence in the text. In Table 2b, factors are sorted according to the number of acknowledged connections (+ signs). This allows a visual representation of factors which are important as affecting other factors, or as being affected by others. We chose this representation over a single dedicated graph although that had been suggested by reviewer 1 and editor, for the following reason:

We have tried various ways to visually show the interconnections described in the text and in Table 2 in a single graph. All attempts showed chaotic plots with no real additional interpretative or insight value. Therefore, we think that our new Table 2b is a good way to visualize the important connections. Actually, we may opt to delete Figure 2a but we are uncertain as it retains the sequence in the text and may be helpful to intercompare connections while reading.

An alternative would be to consider single plots for each factor, with arrows how it affects others and is being affected by others, but that would add 19 single plots to the paper. We leave this to the editor´s appraisal. If the editor suggests that, we are happy to add these plots to the paper, or possibly as supplementary material.

**5. Other changes**

Various small changes and additions were made to the text, please see the responses to reviewer´s comments, and track changes in the text.

**B. Reviewer 1 comments and responses**

Comment: Human impacts …

**General:**

This is a fine overview about a very complex field, and the authors are to be applauded for a good overview. In principle it could be accepted as is, but I take the opportunity to list a few issues, which the authors may want to think about, plus as series of minor points.

> *Thank you, we appreciate the thorough review and suggested changes.*

It would be helpful, If all aspects disused would be introduced with a similar sketch-diagram showing the major and minor influences upon each other.

> *We see the reviewer´s point, and we agree that this would be very good. In fact, we have been going through many iterations and attempts to visualize the different influences on each other. Actually, it became clear that with the many interdependencies, either a sketch would be too chaotic und unreadable, or we would have a proper sketch for each factor, which would mean that the page number would increase even more.*
>
> *We have tried various ways to visually show the interconnections described in the text and in Table 2 in a single graph. All attempts showed chaotic plots with no real additional interpretative or insight value. **Therefore, we think that our new Table 2b is a good way to visualize the important connections.** Also, we would like to refer to our revised Figure 5 (formerly Figure 8) where we have tried to group the factors.*
>
> *An alternative would be to consider single plots for each factor, with arrows how it affects others and is being affected by others, but that would add 19 single plots to the paper. We leave this to the editor´s appraisal. If the editor suggests that, we are happy to add these plots to the paper, or possibly as supplementary material.*

The societal dimension could be covered more seriously. For instance, for tourism, it may matter more the perception of climate change than climate change itself. Also societal preferences may change preferred tourist sites.

> *We have added paragraphs on the societal dimension in sections 3, 5.1.1 and 5.18 and 6. We consider a more thorough consideration of this aspect to be outside the scope of this paper; it would be worth a dedicated paper on its own, but the literature on this topic is very scarce.*

Figure 8 places ecosystems in the center- but there should also be "societal system", which is not really part of the ecosystem (or the rest of the manuscript had to be written differently. In the summary the relationship of society/ecosystem and the role of societal value systems is dealt with in a superfluous manner (characteristic of ecologists)

> *We have revised Figure 5 (formerly Fig 8), and added "human needs and values", and add a paragraph in section 6.*

In general, the historical dimension is a bit weak. What about "pristine"/"undisturbed" conditions, earlier "strange" events; so far it is almost entirely on conditions since 1950 or 1990. (e.g., 6.4, 6.12)

> *We consider this to be outside the scope of this paper, considering the amount of material covered and shortenings that had to be made; also there is a good number of references considering the historical developments of the different topics. So we decided not to add this aspect.*

In the section of shipping, the issue of using different fuels should be mentioned, also the effect of phasing out crude oil (also 6.14)

*The use of crude oil and LNG are and the phasing out of crude oil are discussed in section 5.14.*

In general: reduce the list of linkages wot significant ones, and mention: "the issues X, y, and z likely are insignificant."

*We have stated in the bullet lists in section 5 only the connections which were indicated as significant (+); and we have also discussed the ones with a question mark (?) because we believe that these linkages may be interesting. The insignificant ones are omitted from the bullet lists. They are there in Table 2 and we think they should remain there for the complete picture. This is now better explained in section 4.*

In Sec 6.1, the issues: effect of aerosol emissions and of land use need to be listed aa major issues in the knowledge gaps.

*These knowledge gaps were added to section 5.1.2. Thank you for indicating this important aspect.*

In Sec 6.8 – the issue of cities should be addressed.

*The aspect of urban complexes was added to section 5.8. Thank you for indicating this important aspect.*

Why have big constructions not been addressed – this was a big issue some years ago, and partly still so (Fehmarn belt tunnel).

*The large infrastructure projects (oil and gas pipelines) Nordstream1 and 2 as well as the Baltic Pipe project are discussed in section 6 but the Fehmarn Belt tunnel was not, as ecological impacts are probably very much eliminated as compared to a bridge, which was previously planned.*

**Minor**

- Both tables are hard to read.
  *We have tried to explain both tables better, in particular Table 2a and b (sections 3 and 4)*
- line 295: authors'
  *Corrected*
- 328: I guess this is not so much an issue of the latitudinal position but of the fact that most of Earth is covered by the ocean, but that the sa is only a minor part of the Baltic Sea region.
  *We believe it is related to the northerly position of the region and the proximity to the polar region. The feedback processes related to snow/sea-ice in the winter half of the year are responsible for the large temperature increase. Looking not only at winter it is also clear that the region is to a strong degree continental and the small size of the Baltic Sea implies that it can't reduce the warming in the full region in the summer. So for the sake of brevity we prefer to keep this phrase.*
- 355: warming of the … why so?
  *Thermal expansion is the largest contributor to global sea level rise. Sea level rise in the Baltic Sea basically follows global mean sea level rise (with some modification due to the melting of the Antarctic ice sheet). We have rephrased this paragraph.*

- 363 – the numerical estimate of sa level rise – still valid in view of the new IPCC report; also this number MUST be associated with a time horizon – when?
  *… until 2100.*
- 545 – emissions of chemicals …
  *Corrected*
- 604 – "unfortunate" for whom?
  *The term was be deleted, paragraph rephrased*
- 623 – "dangerous" – for whom?
  *The term was deleted and the sentence was re-phrased to eliminate subjective terminology*
- 630 ff – needs better structuring. Why "as a consequence"?
  *The paragraph was be re-phrased*
- 643 – What is the "running out of sand"-concept"?
  *Sand extractions and potential consequences; paragraph was re-phrased*
- 695ff – unclear; what does the sentence "nt a total load …" tells the reader?
  *The concentrations of heavy metals in the eroding soft cliffs are low, but due to the total volume of eroding material, the introduction of heavy metals into the coastal sea in this way is considerable; rephrased for better understanding.*
- 801 – a repetition
  *Repetition with previous section was eliminated*
- 823 – are these significant effects or just something tny? How much "may speed up corrosion"?
  *Corrosion rates in Baltic Sea conditions were studied in field experiments in different anoxic/oxic transition areas, i.e. the Bornholm Deep and the Gdansk Deep western fringe. Metal coupons placed in near bottom water in fully oxic, oxic/anoxic and fully anoxic sites showed almost a doubling of corrosion rates between anoxic and transitional conditions, and ca. 20% increase in transitional conditions as compared to oxic. Therefore, the following sentence was added: "Therefore, hypoxic events and the resulting changes in oxic conditions may have a considerable impact on corrosion rates, e.g. a doubling in transitional as compared to fully anoxic conditions (Fabisiak et al. 2018)"*
- Figure 2 – needed?
  *Deleted*
- 892 – "importance of short-term variability" – consisting of what?
  *For flux calculations of surface $CO_2$ in coastal areas, short term variability on hourly scales needs to be considered; rephrased*
- 1067 – BSAP?
  *Abbreviation was explained*
- 1095 – refer to climate change section
  *Paragraph was shortened considerably, referred to climate change section*
- 1105ff – us fold face key words.
  *Corrected*
- 1181 – "driving a regime shift through" sound a bit Germanic.
  *The language was be corrected*
- 1255 – "projected decrease in the South" was earlier declared uncartein.
  *The wording was adjusted*
- 1285 – "migration" – of whom?
  *"migration of species", corrected*
- 1323ff something missing
  *We do not understand the comment*

- 1335 – "loss of social welfare" - hm, this may be a value-based assertion by a friend of pristine ecosystems
  *Paragraph was rephrased*
- 1355 – I remember that kay Emeis had an early key paper on this.
  *This section was shortened considerably and we believe it is sufficiently referenced*
- 1448 – what does blue and blue-green stand for?
  *These catchwords were deleted but the concept explained*
- 1475 – More probable – why?
  *Paragraph was rephrased*
- 1480 – add number for usage of wind energy
  *We added a back-of the envelope calculation with a reference for the basic number (energy demand per volume of fish production), using freely available energy production numbers for solar and wind power; depending on the source used these numbers vary a bit; the reader is free to reproduce this calculation with their own figures; the message is that a massive energy production of renewable energy is required if renewable sources are to be used for this type of fish production*
- 1601 – delete – trivial and vague; same with 1608
  *Maybe trivial, but we prefer to keep this as this analysis does not exclude trivial connections, and we think it may be not that trivial, added reference*
- 1658 – "An engineering" – much too general an assessment.
  *Deleted, paragraph re-phrased*
- 1664 – "periodicity of climate" – what shall that be: annual cycle?
  *Deleted, paragraph re-phrased*
- 1728 – deepening
  *Corrected, should be "depending"*
- 1774 – "gentle" is not an adequate word
  *Replaced with "suggesting a small increase in the Northern part…"*
- 1822 – There are --- which?
  *Was specified more*
- 1836 – Anholt is in the Baltic Sea?
  *In a sense, yes… but changed to "…in the Kattegat"*
- 1856 – discussed before
  *Yes, but not as a bullet in this section; whole text was revised to eliminate too much repetition*
- 1870 – what are black waters, what grey waters?
  *Defined in the text*
- 1877 – Kadet
  *Corrected*
- 2935 – one scenario or an ensemble of scenarios?
  *Wrong line number, unable to find the correct one*
- 2308 – something I was wondering while reading: what about military ships and military exercises?
  *It was shortly mentioned that there is no information available on this*
- Figure 6 carries no informational value
  *Deleted*
- 2378 – very general statement
  *We think that this is an important definition of coastal management and how people understand the term; still, section was rephrased*

- 2394- detrimental – for whom? This is related to values
  *Deleted, rephrased*
- Figure 7 – significant for the paper?
  *Deleted*
- 2447 – so what?
  *Should remain as a question mark bullet*
- 2813 – farms
  *?*
- 2982 "eradication of hunger" – hu? This is a key point in the UN millenniums goals.
  *Sorry, hunger is of course not eradicated on the global scale. Section was rephrased.*
- 3145 – what is "blue growth"?
  *Now defined in the text*

**C. Reviewer 2 comments and responses**

This manuscript is an ambitious endeavour, covering very much fild. The topic is very important and I congratulate the authors for taking up the challenge.

However I think some major improvements need to be done with the manuscript.

*Thank you indeed for your appraisal and suggestions, which we will try to follow as far as possible.*

First of all, it is a review paper, yet the methods of the review have not been outlined at all. While a full systematic review might be too much to carry out, the methods of article finding and selection should be clearly explained and - to the extent possible - repeatable.

*We have updated section 4 in which the methodology of the analysis is explained in more detail; we hope that is what the reviewer has in mind.*

Secondly, the criteria for the #+# and "?" categories should be clearly defined and constant throughout the article. This is not currently the case. For example, on r. 1106 -> and 1118, it says there is a + effect, but there are no references to back up these claims. On row 1120 the text says "may be" but it is marked as +, not as ?, and the given reference doesn't relate to tourism at all.

*Thank you indeed for pointing out these inconsistencies. The whole text was scanned and revised so that connections given in the table follow these in the text "bullet list" sections. Wherever there were no references for certain claims, these are now provided, or inappropriate ones corrected.*

Secondly, the paper is uncecessarily long. It covers a lot of ground, which makes it long by necessity, but that being the case, the authors should be extra careful to include only relevant information. At the moment, the introductory sections for each pressure are way too detailed. It would serve the paper better to just give the necessary amount of backgroud that is needed to understand the bullet points below, not give a general review of the pressure.

*We absolutely agree. We have shortened the manuscripts in various ways:*

*1. Former Section 6 was deleted altogether, "knowledge gaps" were inserted as subsections in section 5, "key messages" were deleted as repetitions.*

*2. All subsections in section 5 were shortened considerably to approx. 750 words each.*

*3. Introduction and Discussion sections were shortened considerably*

*4. Former Figures 2, 6 and 7 were deleted, as they do not contribute to the scope of the paper, are too detailed considering the broad scope of the paper, or just of decorative nature.*

*We hope to have arrive at a much more concise and denser manuscript.*

I recommend thet the authors

- shorten the manuscript considerably
  *Done*
- check the consistency of the criteria of teh + and ? classes across all of the work (also considering the joint effects that are implicitly considered as missing!)
  *Done*
- outline the review process
  *Done*